**Cite this article:** van der Hoek Y, Gaona GV, Ciach M, Martin K. 2020 Global relationships between tree-cavity excavators and forest bird richness. *R. Soc. Open Sci.* **7**: 192177.

ecology

facilitator species, indicator species, management surrogates, secondary cavity-nesting birds, Picidae, species interactions

**Author for correspondence:**
Yntze van der Hoek
e-mail: yntzevanderhoek@gmail.com

# Global relationships between tree-cavity excavators and forest bird richness

Yntze van der Hoek[1,2], Gabriel V. Gaona[1], Michał Ciach[3] and Kathy Martin[4,5]

[1]Universidad Regional Amazónica Ikiam, Vía Muyuna, Kilómetro 7, Tena, Ecuador
[2]The Dian Fossey Gorilla Fund International, Musanze, Rwanda
[3]Department of Forest Biodiversity, University of Agriculture, al. 29 Listopada 46, 31-425 Kraków, Poland
[4]Department of Forest and Conservation Sciences, University of British Columbia, 2424 Main Mall, Vancouver, British Columbia, Canada V6T 1Z4
[5]Environment and Climate Change Canada, 5421 Robertson Road, R.R. 1, Delta, British Columbia, Canada V4 K 3N2

YvdH, 0000-0003-3979-2896

Global monitoring of biodiversity and ecosystem change can be aided by the effective use of indicators. Tree-cavity excavators, the majority of which are woodpeckers (Picidae), are known to be useful indicators of the health or naturalness of forest ecosystems and the diversity of forest birds. They are indicators of the latter due to shared associations with particular forest elements and because of their role in facilitating the occurrence of other species through the provision of nesting cavities. Here, we investigated whether these positive correlations between excavators and other forest birds are also found at broad geographical scales. We used global distribution maps to extract richness estimates of tree-cavity nesting and forest-associated birds, which we grouped by zoogeographic regions. We then created generalized least-squares models to assess the relationships between these groups of birds. We show that richness of tree-cavity excavating birds correlates positively with that of secondary cavity nesters and other forest birds (generalists and specialists) at global scales, but with variation across zoogeographic regions. As many excavators are relatively easy to detect, play keystone roles at local scales and are effective management targets, we propose that excavators are useful for biodiversity monitoring across multiple spatial scales and geographical regions, especially in the tropics.

## 1. Introduction

Forests worldwide are facing increasing anthropogenic pressures, with both a rapid decline in the area of natural forests and a

decrease in the naturalness of forest remnants as a result [1,2]. In turn, these reductions of high-quality forest habitats have led to a loss of forest-associated biodiversity [3–5]. To monitor changes in forest ecosystems and their denizens, we often look at specialized animals that require the availability of specific habitat structures and processes running across long temporal and large spatial scales for long-term persistence [6], such as birds that respond not only to a loss in overall forest cover but also to changes in forest health, quality and integrity [7–10]. Excavators such as woodpeckers, barbets, nuthatches, trogons and certain parrot species—among which woodpeckers are the most numerous group [11]—may be especially effective indicators due to their associations with particular forest elements and their role in facilitating the presence of other species through the provision of nest cavities. Indeed, there is empirical evidence that the presence of excavators can be indicative of the state of both the forest (e.g. the presence of large trees [12], heterogeneous forest structure [13] or a high level of naturalness [14–18]) as well as the richness and abundance of other forest-associated species [19–21]. However, we may question whether excavators are universally effective indicators, especially of overall levels of biodiversity, in all geographical regions of the globe or for all forest types [15].

If relationships between excavators and other forest-associated biota are universally positive, and are common not only at stand and landscape scales but hold across large geographical regions, then excavators could form a unique group of indicators that may be effective across multiple locations and spatial scales [22]. To find species or guilds that are effective as both indicator and management surrogates across multiple regions would be especially useful in the largely understudied tropics where comprehensive biodiversity assessments are costly and logistically challenging [23]. The first signs are promising, as the overwhelming majority of tree-cavity excavators are forest or woodland birds and general patterns of woodpecker richness correlate positively with amounts of forest cover at the global scale [24]. Moreover, many excavators are regarded as habitat specialists and tend to have highly specific habitat requirements which are usually only met in forests with high degrees of naturalness and low levels of human-related disturbances (e.g. logging). Indeed, restoration efforts have effectively used the presence of woodpeckers to indicate that restored habitats have reached a certain level of naturalness [25,26]. Excavators often use large, live or decaying trees, or coarse woody debris, for nesting, roosting and foraging, and the availability of large amounts of dead wood is found to be important for their long-term persistence in a given area [27–30]. As a result, we find that excavators are highly responsive to local changes in forest habitat quantity, quality and the loss of certain forest elements, and may, therefore, be effective indicators of the naturalness of ecosystems (e.g. [31]).

Excavators are reliable indicators of the presence of other forest birds through their mutual associations with forest characteristics, but correlations between both groups of birds are also determined partially by species interactions. Taken together, these 'nest-web' networks [32] of interacting species are structured by the availability and acquisition of tree cavities for nesting and roosting. These cavities are formed by two major processes: 'fungal formation', in which fungi decompose wood over an extended period, or animal excavation [33,34]. Since most excavators—at least the majority of woodpeckers—excavate a new nest cavity each year, this mode of cavity creation provides a steady supply of potential roosting and nesting sites available to secondary users, which includes vertebrates [34], invertebrates [35,36] and diverse fungal assemblages [37]. As a result, cavity formation by excavation is an important source of nest holes for numerous species in many regions of the world, especially in North America [34]. However, decay-formed cavities are a more important source of nest sites in other regions, especially in the tropics [33]. Thus, it is very likely that the relative importance of both processes of cavity formation—and thus the strength of the relationship between richness of excavators and secondary cavity nesters—varies globally [34].

It is worth noting that excavators play additional roles (beyond the direct provision of cavities) that may facilitate the presence of other species (both cavity nesters and birds using other types of nest sites). For example, excavators can aid in the dispersal of fungi, which in turn enhances wood softening processes and the formation of decay-formed cavities [38]. In addition, certain excavators that perforate the bark of trees exposing insects or sap provide foraging opportunities for other species [39,40].

Given the aforementioned indirect associations (mutually shared habitat preferences) and direct interactions (provision of cavities and foraging opportunities), excavators are considered reliable indicators of forest bird diversity and richness at the stand or landscape levels [9,19]. However, it is unknown whether this relationship scales up across larger regions or holds across the globe, as interactions between cavity-nesting birds and other forest birds are likely to vary spatially in strength and complexity (e.g. [41]). Moreover, certain dominant groups of excavators (woodpeckers, in particular) vary in richness across zoogeographic regions [11], which may induce variation among relationships between excavators and other forest birds.

Here, we aim to investigate whether there are consistent patterns in the relationships between avian tree-cavity excavators (hereafter: excavators) and species richness of non-excavating tree-cavity-nesting birds (hereafter: secondary cavity nesters), or non-cavity-nesting forest birds (forest specialist and forest generalist species), across the globe. If previously observed relationships between cavity excavators and other forest birds scale up from local forest stands [17] or ecosystems [19] to zoogeographic or global levels, then this strengthens the potential value of tree-cavity excavators as both indicators and management surrogates.

Given the important potential indicator role of excavators in forest ecosystems across the world, we predict a global tendency for richness of secondary cavity nesters, forest specialists and forest generalists to increase in correlation with the number of excavators present in the ecosystem, regardless of the ecosystem or geographical location. For a large part, these relationships between excavators and other forest birds reflect whether species have joint or overlapping distributions, something largely shaped by factors such as broad spatial patterns of forest cover. However, we also predict that the nature (e.g. the strength and slope of correlations) of the relationships between excavator richness and richness of these other groups of forest-associated birds will vary across zoogeographic regions, reflecting global differences in dominant forest characteristics (e.g. broad-leaved versus coniferous forests), forest management practices (e.g. forests being more or less intensively managed) and bird communities (e.g. the relative richness of different groups of forest-associated birds), some aspects of which were previously discussed by, for example, Cockle *et al.* [34]. In particular, we predict that correlations between excavators and secondary cavity nesters will be strong in regions with slow processes of decay-related tree-cavity formation. Thus, in predominantly temperate and boreal regions such as the Nearctic and Palaearctic, relationships between excavators and secondary cavity nesters driven by mutual associations with particular forest elements will be enhanced by direct interactions through the provision of nesting and roosting cavities as a lack of available decay-formed cavities may increase the dependence of secondary cavity nesters on excavated cavities. By contrast, for regions that largely span the tropics, such as the Neotropical, Afrotropical and Oriental regions, we predict weaker, but still positive, correlations between excavators and secondary cavity nesters. These relationships will be driven mainly by mutual associations with particular forest elements and less by direct interactions through the provision of nest cavities—decay-formed cavities being more frequently used by secondary cavity nesters in these regions [33,41].

We also predict that relationships between excavators and secondary cavity nesters are stronger than those between excavators and non-cavity-nesting forest birds (specialists or generalists) in the Nearctic and Palaearctic, but that this pattern will be less clear for the predominantly tropical regions. For Australasia, where the largest group of excavators (woodpeckers) is absent, we predict that relationships between the present excavators, often weaker excavators that require softer or rotting wood (see [11] for species) and other groups of birds will be relatively weak and based mainly on shared habitat preferences. For all regions, we predict that relationships between excavators and secondary cavity nesters and between excavators and forest specialists will be stronger than those between excavators and forest generalists, as the latter group includes species that are less likely to have either direct interactions or mutually shared habitat (forest) preferences with excavators. The relationships between excavators and forest generalists may even be negative, as many forest generalists may also be found in regions with sparse tree cover and have habitat requirements opposite from those required by excavators due to traits such as ground-nesting behaviour or preferences for early-successional forests with small trees [41].

Finally, we assessed the strength of the relationships between excavators and all non-cavity-nesting forest-associated birds (i.e. forest specialists and forest generalists pooled) in each region. If these relationships turn out to be strong and uniform in direction, then that would strengthen the notion of excavators as a pragmatic and effective indicator of forest birds via mutual associations with habitat elements and characteristics.

## 2. Material and methods

We used published global distribution maps of tree-cavity-nesting birds [11] and forest birds [42] to extract richness estimates per $10 \times 10$ km grid cell (table 1). Maps of the distribution of tree-cavity nesters and forest birds were based on the species range maps provided by BirdLife International and NatureServe [43]. We selected and classified focal bird species (following [11,42]) as (i) excavators (species that are known to often or always excavate their nesting cavities), (ii) secondary cavity

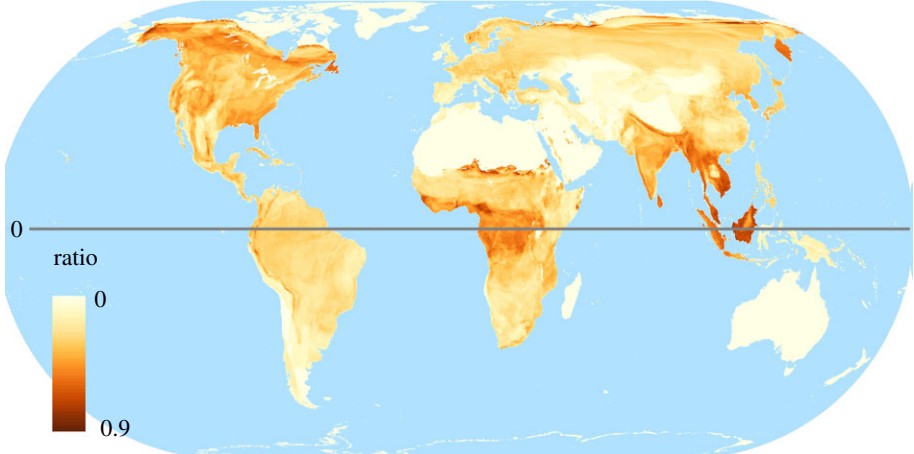

**Figure 1.** Global map of the relative richness of tree-cavity excavators versus secondary cavity-nesting birds (expressed as the ratio excavator/secondary cavity-nester) in 10 × 10 km grid cells. Low values (light colours) indicate relatively low numbers of excavators when compared with secondary cavity nesters, while high values (dark colours) indicate a relatively higher proportion of excavators to secondary cavity nesters.

**Table 1.** Species richness for forest-associated bird groups found in each of six zoogeographic regions. In parenthesis, the maximum number of species of each group found in a single 10 × 10 km grid cell in each region.

| zoographic region | excavators | secondary cavity nesters | forest specialists[a] | forest generalists[a] | all forest birds (specialists + generalists)[a] |
|---|---|---|---|---|---|
| Nearctic | 42 (16) | 111 (41) | 93 (36) | 365 (111) | 458 (140) |
| Palaearctic | 67 (23) | 181 (58) | 155 (65) | 725 (221) | 880 (276) |
| Oriental | 120 (35) | 241 (67) | 463 (79) | 958 (238) | 1421 (300) |
| Neotropical | 173 (36) | 373 (114) | 1078 (200) | 1764 (277) | 2842 (436) |
| Afrotropical | 78 (33) | 218 (68) | 220 (60) | 911 (287) | 1131 (331) |
| Australasia | 14 (7) | 178 (70) | 442 (67) | 1014 (190) | 1456 (226) |

[a]These numbers do not include secondary cavity nesters.

nesters: birds which are known to nest in tree cavities but which never or rarely excavate their cavity, (iii) forest specialists: birds that use forest habitat exclusively, excluding cavity nesters, and (iv) forest generalists: habitat generalist birds that use forest habitat but also at least one other habitat type for nesting, again excluding cavity nesters.

We used only range polygons where species presence was classified as 'Extant' or 'Probably extant', and assigned a bird's 'presence' to all grid cells that overlapped with range polygons. Next, we created generalized least-squares (gls) models to assess linear relationships between excavators and secondary cavity nesters, forest specialists, forest generalists and non-cavity-nesting forest birds grouped (specialists + generalists). For this, we aimed to take biogeographic differences into account and thus created gls models for the six zoographic regions separately: Nearctic, Palaearctic, Neotropical, Afrotropical, Oriental and Australasia. We did not restrict our analyses to forested regions as we wanted to avoid making subjective decisions on the classification of a grid cell to a specific vegetation/habitat type. Yet, to ensure we looked for possible relationships in regions where these forest-associated species occur, we removed cells with zero excavators, and cells where the numbers of secondary cavity nesters, forest specialists or forest generalists were lower than that of excavators, from further analyses.

Our data were spatially structured (i.e. the richness of birds in different categories (excavators, etc.) showed spatial clustering) across and within zoogeographic regions, as we demonstrated by mapping the global distribution of the relative number of cavity excavators to non-excavating cavity-nesting birds (expressed as the ratio of excavator/non-excavating cavity-nester; figure 1). Thus, we followed protocols to adjust for spatial autocorrelation as per Storch *et al.* [44]. This implied the use of gls

models with excavator richness as a predictor variable, the richness of one of the other groups of birds as a response variable and a variable exponential spatial covariance structure, which reduced—rather than eliminated—the influence of spatial autocorrelation on our models. We used exponential covariance structures, because the Akaike's information criterion (AIC) of models with such structures was consistently lower than that of least-squares models with no spatial structure, or of gls models with Gaussian, linear, rational quadratics and spherical spatial covariance structures.

Although gls models work well with large datasets, we found that gls models fitted on all data failed to converge (we used the R package 'nlme' [45]). To circumvent this computational problem, we first grouped grid cells by zoogeographic region. We thereafter ran 1000 gls models on bootstrap permutations of 1000 randomly selected grid cells (see [24,46]). Of these 1000 model iterations, we subsequently calculated the mean (±s.d.), and minimum and maximum coefficients of the regression slope.

As gls models do not provide a measure of the strength of relationships, we next proceeded with a measure of the strength of each correlation: we fitted each exponential gls model to a different, non-overlapping, test dataset of 1000 randomly selected grid cells. We then proceeded to calculate the Spearman $\rho$ coefficient of the correlation between predicted and observed data, for all 1000 models per zoogeographic region. Finally, we calculated the mean Spearman $\rho$ of all significant models per zoogeographic region and the percentage of iterations that generated significant Spearman correlations at an $\alpha$ of 0.05. It is both statistically problematic to calculate an average for these probability values and computationally problematic to generate a plot of all gls models. Thus, we opted to visualize the relationships for each region using scatterplots of the species richness values in all $10 \times 10$ km grid cells, with a fitted line that we generated using a linear regression model to show the general trends. We repeated all of these analyses, except the visualizations, for woodpeckers as a subset of the excavators (excluding the Australasian region where woodpeckers are absent). All data, R scripts and initial results needed to reproduce our analyses presented here are deposited online (electronic supplementary material, appendix S1).

# 3. Results

Using gls modelling to assess the global patterns, we found positive relationships for all correlations between excavator richness and that of secondary cavity nesters, forest specialists and forest generalists, except between excavators and both secondary cavity nesters and forest generalists in the Australasian region. However, these relationships differed in relative strength and characteristics (e.g. steepness of regression slope) depending on the group of birds and zoogeographic region analysed (figure 2 and table 2). For clarity, we reiterate that the mean slope indicates the general increase in number of species for a given bird group (i.e. non-excavating cavity nesters, forest specialists, forest generalists) per increase in excavator species richness, whereas the mean Spearman $\rho$ is a measure of the strength of these relationships.

Taken broadly, we found stronger relationships (higher Spearman $\rho$) in the predominantly tropical Afrotropical, Oriental and Neotropical regions than in the comparably more temperate/boreal Nearctic and Palaearctic regions, and particularly weak relationships in Australasia. Moreover, the relationships between excavators and the other three groups of birds were generally similar (positive), without clear support for our prediction that the relationship between excavators and secondary cavity nesters is stronger than that between excavators and non-cavity nesters (both forest specialists and forest generalists). Instead, we found that the relative strength of these relationships (i.e. between excavators and the other groups of birds) differed across zoogeographic regions. For example, excavator richness was a particularly strong predictor of secondary cavity-nester richness in the Oriental region ($\rho = 0.88$), and slightly less so for forest specialists ($\rho = 0.72$), whereas in the Afrotropics, excavator richness showed a strong relationship with forest specialists ($\rho = 0.83$) and forest generalists ($\rho = 0.91$), but a somewhat weaker association with secondary cavity nesters ($\rho = 0.79$). We note that excavators in Australasia (which include no woodpeckers) showed a moderately strong ($\rho = 0.73$) and significant ($p < 0.05$ in 100% of model iterations) positive relationship with forest specialists, and none with secondary cavity nesters or forest generalist ($\rho \leq |0.19|$ for both). Moreover, for Australasia, we found that the slopes of gls models for the relationship between excavator and secondary cavity-nester richness were positive, but that Spearman $\rho$ indicated a weak to negligible negative relationship ($\rho = -0.14$). This seemingly contradictory result potentially stemmed from the small sample of excavators for this region. Finally, for all regions, we found particularly strong relationships between excavators and forest specialists and generalists grouped, often stronger than for forest specialists or generalists alone.

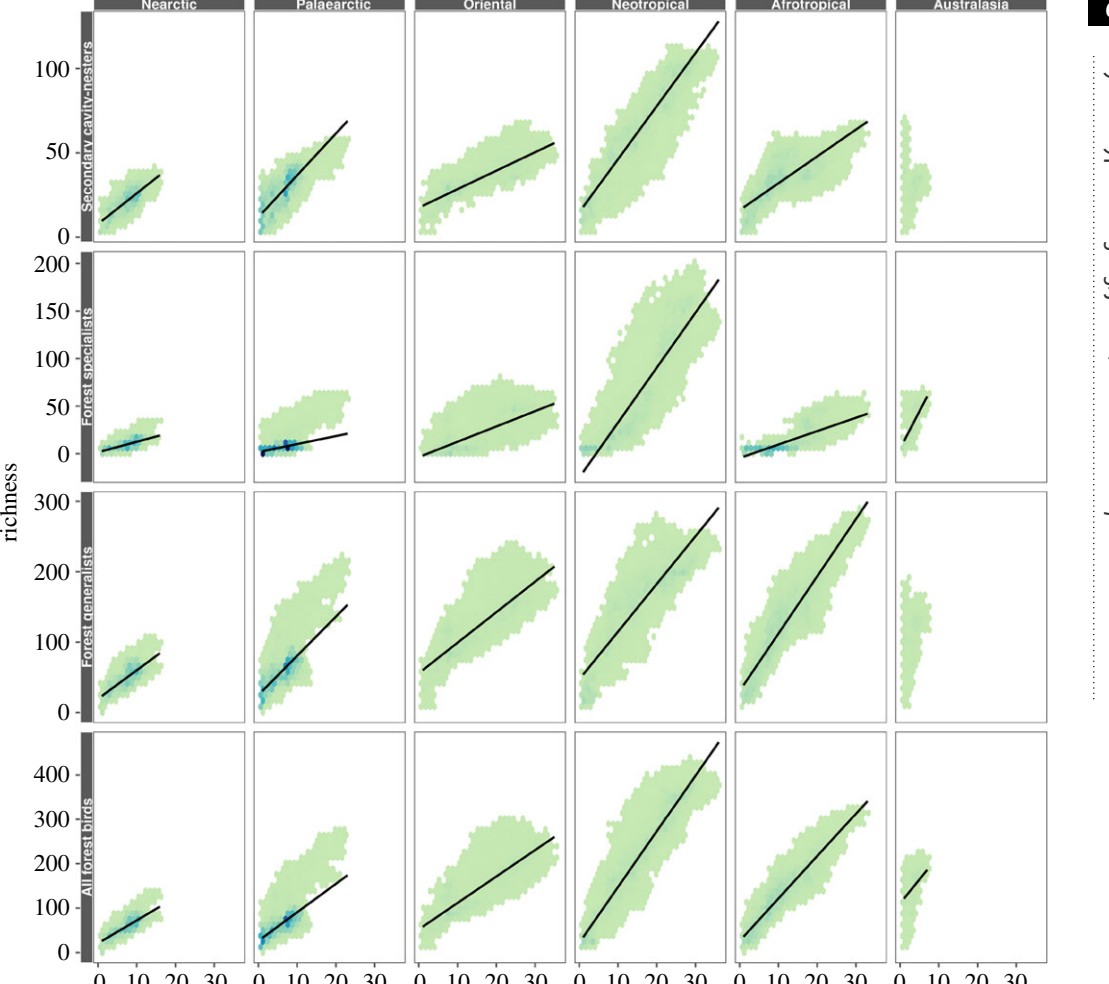

**Figure 2.** Scatter plots of the correlations between the richness (number of species) of excavators and secondary cavity nesters (upper row), forest specialists (second row), forest generalists (third row) and all non-cavity-nesting forest birds (specialists and generalists; bottom row) in six zoogeographic regions. Points represent species richness values in $10 \times 10$ km grid cells, with darker colours representing an overlap of multiple points and, therefore, higher richness counts. The solid lines represent linear regression models, created using all grid cell values. We omitted regression lines for the relationships between excavators and secondary cavity nesters as well as between excavators and forest specialists for the Australasian region, as these relationships were negligibly weak (Spearman $\rho \leq |0.19|$ for both relationships). Note that nearly all secondary cavity nesters are also forest-associated species, but were not included as forest specialists or generalists.

In terms of variation in the regression slopes, we found that the relationships between excavators and other groups of birds were typified by particularly steep regression slopes in the Neotropics. For example, the presence of one additional excavator species in a grid cell was related to an increase of more than two additional secondary cavity nesters. By contrast, regression slopes were less steep in the predominantly temperate and boreal regions such as the Nearctic and Palaearctic, where an increase of one excavator species signified an increase of approximately one secondary cavity-nester species.

We repeated all the above-mentioned analyses for woodpeckers as a subset of excavators and found nearly identical results. We list the results of these analyses in electronic supplementary material, table S1.

## 4. Discussion

We found strong positive relationships between excavators and secondary cavity nesters, forest specialists, and forest generalists at all zoogeographic scales except for secondary cavity nesters in Australasia.

**Table 2.** Results of generalized least square models of possible correlations between (1) excavator and secondary cavity-nester richness, (2) excavator and forest specialist richness, (3) excavator and forest generalist richness and (4) excavator and forest specialists and generalist richness. Each result reflects the mean outcome of 1000 models build with 1000 randomly selected grid cell data points that include an exponential spatial covariance structure. The mean slope indicates by how many species a given group generally increases per unit increase in excavator richness and the mean Spearman $\rho$ is a measure of the strength of the relationships.

| zoogeographic region | correlation | mean slope (min–max; s.d.) | % of models with $p < 0.05$ | mean Spearman $\rho$ |
|---|---|---|---|---|
| Nearctic | 1 | 0.8 (0.4–1.3; 0.1) | 60 | 0.73 |
| | 2 | 1.0 (0.7–1.3; 0.1) | 97 | 0.71 |
| | 3 | 3.1 (2.0–4.5; 0.4) | 89 | 0.76 |
| | 4 | 4.2 (2.7–5.7; 0.5) | 94 | 0.81 |
| Palaearctic | 1 | 1.1 (0.7–1.4; 0.1) | 84 | 0.78 |
| | 2 | 0.6 (0.3–1.1; 0.3) | 83 | 0.68 |
| | 3 | 2.9 (2.1–4.1; 0.3) | 82 | 0.80 |
| | 4 | 3.5 (2.6–5.4; 0.4) | 88 | 0.80 |
| Oriental | 1 | 0.7 (0.6–0.9; 0.0) | 100 | 0.88 |
| | 2 | 1.1 (0.9–1.3; 0.1) | 100 | 0.72 |
| | 3 | 3.5 (3.0–4.0; 0.2) | 100 | 0.86 |
| | 4 | 4.6 (3.9–5.5; 0.2) | 100 | 0.83 |
| Neotropical | 1 | 2.4 (1.9–3.0; 0.2) | 83 | 0.95 |
| | 2 | 3.5 (9.1–14.0; 0.2) | 73 | 0.95 |
| | 3 | 5.2 (3.9–6.7; 0.5) | 77 | 0.91 |
| | 4 | 8.8 (7.1–11.0; 0.6) | 84 | 0.96 |
| Afrotropical | 1 | 1.1 (0.7–1.9; 0.2) | 76 | 0.79 |
| | 2 | 0.6 (0.3–1.1; 0.1) | 78 | 0.83 |
| | 3 | 5.8 (3.7–8.1; 0.9) | 84 | 0.93 |
| | 4 | 6.4 (4.3–9.2; 0.9) | 87 | 0.93 |
| Australasian | 1 | 1.8 (1.4–2.2; 0.1) | 88 | −0.14 |
| | 2 | 1.7 (0.9–2.5; 0.3) | 100 | 0.73 |
| | 3 | 9.2 (7.2–11.2; 0.6) | 96 | 0.19 |
| | 4 | 11.0 (8.3–13.5; 0.8) | 99 | 0.50 |

Although these relationships differed in both strength and characteristics (regression slopes) across zoogeographic regions, they presented comparable patterns concerning the potential role of excavators to serve as indicators for avian biodiversity. For example, relationships between excavators and non-cavity-nesting forest birds were particularly strong in most regions, often stronger than those between excavators and secondary cavity nesters. This indicates that these relationships are probably shaped more by indirect associations with forest elements and shared habitat preferences, thus reflecting correlated distribution patterns of these groups of birds shaped by spatial patterns in forest cover, rather than direct facilitation of nest cavities, although the relative importance of both factors depends on the region under consideration. Our results imply that excavators and woodpeckers (a predominant subset of excavators) hold strong potential as indicator and management surrogates across most of the world's forests [47], with the possible exception of the Australasian region, but also that we need to take differences among zoogeographic regions into account (e.g. the composition of local and regional nest webs and the relative numbers of excavators and secondary cavity nesters as seen in figure 1) if we are to adopt excavators as surrogates for conservation or management. By considering the relative strengths of the relationships, we can indicate where and for which species excavators would make the best surrogates.

Variation across biogeographic regions in the relationships between excavators and other groups of forest birds is firstly the result of macro-scale processes that determine species distributions (e.g. as outlined by

Gaston [48]). The data used for our analyses reflect patterns of overlapping ranges of excavators and other forest birds, which indicate not so much the presence of causative relationships but rather of correlations in distributions caused by broad-scale patterns in biological processes, land cover and climate. Biological processes, such as speciation events, drive spatial patterns in avian richness [49], including cavity-nesting birds [11], and trees as potential substrates for cavities [50]. Among zoogeographical regions, differences in richness and related factors such as niche specialization may contribute to patterns such as the stronger relationships between excavators and other groups of forest birds in predominantly tropical regions when compared with the mainly temperate/boreal regions. For example, nest webs in a subtropical forest in South America have a higher diversity and evenness of interactions compared to other nest webs in temperate zones [33,34]. A similar pattern can be seen in plant–pollinator networks, where networks in the tropics show relatively low connectance but high modularity when compared with networks in temperate zones [51]. For plant–pollinator networks, there are efforts to link these differences in tropical and temperate zones to larger biogeographic (e.g. latitudinal) patterns and we may wish to explore whether spatial patterns in the relative strength of direct facilitation of nest cavities could be governed by similar rules. These macro-scale spatial patterns may also influence the relative strength of indirect associations with forest elements and shared habitat preferences. We could hypothesize that because excavators in the tropics have particularly strong associations with dead trees [52], they are particularly strong indicators of mature forest (but see [33] for a contrasting observation in a subtropical forest). This could create strong links with other species that require mature forests, explaining the particularly strong relationships observed for the largely tropical zoogeographic regions.

Variation in the strength of these relationships could also stem partially from macro-scale spatial variation in patterns of land cover or land-use change (e.g. in patterns of historical deforestation, discussed below), which may explain why regions with the largest areas of mature forest (i.e. the Neotropics and Afrotropics) show the strongest relationships. Similarly, and in addition to biological processes and land cover, climatic factors could be a driver of the observed differences in relationships across regions. For example, climatic factors influence the availability of decay-formed cavities [34], and may thus influence the dependency of secondary cavity nesters on excavators for cavities. In general, excavated tree cavities are less likely to be used by secondary cavity nesters in the tropics, as high rates of precipitation and high temperatures lead to a low persistence (i.e. longevity) of excavated cavities [34] and the relatively rapid formation of cavities through fungal activity [53]. This, in turn, leads to relatively higher use of decay-formed cavities by secondary cavity nesters in the tropics [34,54], and less dependence on excavators for the supply of nesting cavities. For example, even woodpeckers will occasionally use decay-formed cavities in Neotropical temperate rainforests where tree decay is the key driver of nest-web structure [54]. Thus, the fact that the relationships between excavators and secondary cavity nesters are particularly strong in all three largely tropical regions (Oriental, Neotropical and Afrotropical) may not reflect a larger dependency of secondary cavity nesters on excavators for cavities in the tropics, but is more likely to be linked to an overall high species richness in these regions as well as to the strong indirect associations that both excavators and secondary cavity nesters have with shared forest elements and habitat preferences.

Although the relationships assessed in this study are correlative rather than causative, we observed that relationships varied in strength among different groups of birds, potentially reflecting that the broad-scale distribution of certain species was determined by more than the macro-scale factors described above. For example, we found that relationships between excavators and secondary cavity nesters were stronger than those between excavators and forest specialists within the Nearctic and Palaearctic regions, which may reflect the additional importance of cavity facilitation. Or, to put this differently, secondary cavity nesters are more often found coexisting with excavators beyond what could be predicted from broad-scale patterns in shared forest cover preferences. We find something similar in the largely tropical Oriental region, which may be the result of the specific characteristics of the forests or avifaunal communities in this region (e.g. it is a hotspot for woodpeckers [55]) but may also be influenced by the fact that there is less mature forest left in this region than in other tropical regions (see [56] for brief synopsis). The loss of mature forest reduces the availability of substrates (e.g. large dead trees) for excavators as well as the opportunity for cavities to form via decay [57], potentially making secondary cavity nesters more dependent on excavated cavities. In contrast with the Nearctic, Palaearctic and Oriental regions, we found weaker relationships between excavators and secondary cavity nesters than between excavators and forest specialists in both the Neotropical and Afrotropical regions, possibly because at least some secondary cavity nesters do not share a preference for habitats and forest elements with excavators in this region and because a high reliance on decay-formed cavities releases secondary cavity nesters from dependency on excavators for cavities [34].

Interestingly, and in contrast with our predictions, relationships between both excavators and secondary cavity nesters and between excavators and forest specialists were often weaker than those

between excavators and forest generalists. We do not have a clear explanation for this finding, although we argue that forest generalists and excavators might share landscapes where forest specialists are hard to find, for example, in open landscapes with scattered trees [58]. Not all excavators require mature forest and, in fact, some species require certain elements in more open landscapes (e.g. large scattered dead trees used by species from the woodpecker genus *Melanerpes* [52]) that are also important for forest generalist bird species (e.g. as perching substrates). Finally, we acknowledge that our use of range map data allows for assessments of broad overlaps in species distributions, via indirect associations with broad-scale topographies, climates and vegetation or forest cover types, but may not always allow for assessments of the fine-grained species associations and interactions between excavators and secondary cavity nesters or forest specialists.

Relationships between excavators and other groups of birds were less evident in Australasia, a region that differs from all others in both the composition of avifauna (e.g. there are no woodpeckers there, but the richness of cavity nesters is relatively high [11]) and forest structure and ecological processes (e.g. cavity densities in Australasian forests are among the highest globally due to rapid rates of decay [53]). In Australasia, we found a moderately strong relationship between excavators and non-cavity-nesting forest specialists, but not between excavators and secondary cavity nesters or forest generalists. This could indicate that the excavators in this region show a preference for forest conditions (e.g. the provision of trees that are in advanced stages of decay, given that many of these excavators are smaller weaker excavators that require softer or rotting wood; see [11] for species), habitat conditions that are selected by many forest specialists. By contrast, secondary cavity nesters in Australasia may depend on elements not particularly required by excavators (e.g. large old trees [58] that contain numerous dead branches and have accumulated multiple decay-formed cavities [59]).

Beyond variation in the strength of relationships among regions, we also found that regressions differed in slopes, both across regions and across the different relationships tested. For example, the steep slopes we found for correlations between excavators and non-cavity-nesting forest birds in the predominantly tropical Neotropical region contrast with more shallow slopes in the temperate/boreal Palaearctic and Nearctic regions. Within the Neotropical region, the relationship between excavators and secondary cavity nesters showed shallower slopes than that between excavators and non-cavity-nesting forest birds. These differences in slopes mainly reflect differences in species richness across regions and groups of forest birds, with relatively steeper slopes representing a relatively larger increase in richness of a particular group of birds per increase in the number of excavators.

In conclusion, we found strong and positive relationships between excavator and forest bird richness around most of the world, not only for specialized species such as secondary cavity nesters or forest specialists but also for forest generalists and the overall forest-associated bird communities (specialists + generalists). As such, we deem excavators to be useful indicator species for forest bird diversity at broad spatial scales, as was previously demonstrated at stand and landscape scales. Excavators, especially woodpeckers, form a small and fairly easy to identify (even by citizen scientists) subset of a larger group of species (forest birds) often studied to understand the effects of anthropogenic disturbances. Moreover, many woodpeckers have relatively high detectability and respond to playbacks, a potentially effective survey tool that allows researchers to survey large areas fairly readily (e.g. [60]). Studies of excavator presence and abundance, especially in high biodiversity tropical regions, might provide quick measures of forest quality and biodiversity, which may in turn guide forest management decisions (e.g. improving the retention of large trees). A comprehensive study of all forest birds in some regions like the Amazon or Congo Basin (where, notably, there are hotspots of woodpecker richness [55]) would require considerable local expertise and training, which is often difficult to achieve due to social and monetary limitations. Thus, we conclude, excavators have excellent potential as study subjects for conservation monitoring and planning initiatives (see also [61] with regard to woodpecker conservation itself) and to guide the establishment of region-wide forest conservation strategies, especially in the largely understudied tropical regions of the world.

Data accessibility. Supporting data as well as R scripts can be found on the GitLab repository https://gitlab.com/gavg712/CB_cavity_nesters, the details of which are outlined in electronic supplementary material, appendix S1.

Authors' contributions. Y.v.d.H. and G.V.G. carried out the data analysis, and led the design of the study and writing of the manuscript; M.C. and K.M. critically revised the manuscript. All authors contributed to the writing of the final manuscript, gave final approval for publication and agreed to be held accountable for the work performed therein.

Competing interests. We declare we have no competing interests.

Funding. This research was not externally funded.

Acknowledgements. We would like to thank Dr Lisa Manne, Dr Kristina Cockle and three anonymous reviewers for their critical remarks on drafts of this article.

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
