## [Reviewer comments · Royal Society Open Science]

Review History

RSOS-190962.R0 (Original submission)

Review form: Reviewer 1

Is the manuscript scientifically sound in its present form?

Yes

Are the interpretations and conclusions justified by the results?

Yes

Is the language acceptable?

Yes

Do you have any ethical concerns with this paper?

No

Have you any concerns about statistical analyses in this paper?

No

Recommendation?

Major revision is needed (please make suggestions in comments)

Comments to the Author(s)

General comment:

The manuscript entitled "Global and universal relationships between tree-cavity excavators and forest bird richness." reports on an interesting exploratory study linking woodpeckers/other tree cavity excavators, secondary cavity users and other forest tree species at the global and bioregional levels. The results are useful and may inspire new studies exploring these relationships more in-depth. My main criticism concerns the quality of reasoning and how and to what extent different issues are introduced and discussed. Proposing the use of certain group of species as indicators or other type of indicator species shall rest on solid grounds as well as need of such indicators should be clearly put forward. I simply miss the delineation of the conservation/management problem why we need indicators. There are many articles published on the worldwide decline of primary/intact forests, their characteristics and, in my opinion, you do not use the opportunity to discuss the potential role of woodpeckers/other tree cavity excavators (with very intimate linkages to natural forest characteristics) to support your case. Also, even if you mention their usefulness in management, there is lacking information on the intensively managed forests as areas that lose both important characteristics linked to old-growth forests and forests with high level of naturalness and thereby lose most specialized forest species often being woodpeckers. Also use of woodpeckers/other excavators as indicators of successful forest restoration could be mentioned. There is large body of literature on both above topics and I suggest that you should add paragraphs pertaining on these issues in both Introduction and Discussion sections. To limit the length of the article, you may get rid of several unnecessary repetitions. More suggestions below:

Detailed comments:

Line 99: Several species use smaller trees or snags too. Possibly add "often" or "usually".

Line 109: Woodpeckers (cavity excavators) are most often resident species what could additionally strengthen your reasoning on strong linkages to forest environments and their quality.

Lines 127-133: Please consult Wesolowski and Martin (2018) and extend the discussion on mechanism a bit more. I am lacking a clear statement on two possible, non-excluding each other mechanisms of expected relationships as very nicely stated way down in Discussion (Lines 312-314).

RESULTS: Could you add a summary table with numbers of species in different groups in different bioregions? It would provide some nice background for further discussions of results. You have those numbers already!

Lines 224-231: I do not understand how you can model the relationship between woodpeckers and the three other groups in Australasia, where, as you state below, there is no woodpeckers or even other strong excavators are largely lacking.

DISCUSSION: a bit confusing use of words "excavators/woodpeckers" (Line 244) and "excavators" (Line 275). Please check if necessary to use both of them!

Lines 239-243: Worthy to mention that forests in some bioregions supply decay-formed cavities in high numbers (particularly non-managed forests) and some other not (e.g. boreal forest in Europe - see Andersson et al. (2018): Andersson J., Domingo Gomez E., Michon S., Roberge J.-M. (2018) Tree cavity densities and characteristics in managed and unmanaged Swedish boreal forest. *Scandinavian Journal of Forest Research*, 33 (3), pp. 233-244. It would add the management angle in discussion about the role of woodpeckers/excavators for other species.

Lines 272-273: Please refer to studies on the global species richness of woodpeckers e.g. Mikusinski, G. 2006. Woodpeckers (Picidae) - distribution, conservation and research in a global perspective. *Annales Zoologici Fennici* 43: 86-95.

Lines 289-294: Refer also to the "primeval" Bialowieza Forest case in temperate zone being in stark contrast to managed forests (i.e. higher predation risk and subsequently preference for decay-formed cavities among secondary users. See: Wesolowski and Martin (2018) and references therein.

Lines 299-304: Please refer to papers by Wesolowski here: Wesolowski T. 2011. "Lifespan" of woodpecker-made holes in a primeval temperate forest: A thirty year study. Forest Ecology and Management, 262 (9) , pp. 1846-1852.

Review form: Reviewer 2

Is the manuscript scientifically sound in its present form?

No

Are the interpretations and conclusions justified by the results?

No

Is the language acceptable?

Yes

Do you have any ethical concerns with this paper?

No

Have you any concerns about statistical analyses in this paper?

Yes

Recommendation?

Major revision is needed (please make suggestions in comments)

Comments to the Author(s)

Many comments are made in the word file attached (Appendix A).

Decision letter (RSOS-190962.R0)

31-Jul-2019

Dear Dr van der Hoek:

Manuscript ID RSOS-190962 entitled "Global and universal relationships between tree-cavity excavators and forest bird richness" which you submitted to Royal Society Open Science, has been reviewed. The comments from reviewers are included at the bottom of this letter.

In view of the criticisms of the reviewers, the manuscript has been rejected in its current form. However, a new manuscript may be submitted which takes into consideration these comments.

Please note that resubmitting your manuscript does not guarantee eventual acceptance, and that your resubmission will be subject to peer review before a decision is made.

Your resubmitted manuscript should be submitted by 28-Jan-2020. If you are unable to submit by this date please contact the Editorial Office.

on behalf of Kevin Padian (Subject Editor)
openscience@royalsociety.org

Subject Editor Comments to Authors:

I am recommending a "reject/resub" decision because our "revision" turnaround time may be too short for the authors' needs in revising, and because the reviewers have raised extensive concerns, despite being encouraging overall about the manuscript.

They see issues with how the main problem of the study is set out, the methods, the evaluation, and the conclusions. They also note a geographic bias that may entail ecological biases in the relationships of excavators to their ecosystems.

If you elect to re-submit, please detail your responses to reviewers' comments and make clear how you have altered your re-submitted manuscript. Best wishes with your revisions.

Reviewers' Comments to Author:

Reviewer: 1

General comment:

The manuscript entitled "Global and universal relationships between tree-cavity excavators and forest bird richness." reports on an interesting exploratory study linking woodpeckers/other tree cavity excavators, secondary cavity users and other forest tree species at the global and bioregional levels. The results are useful and may inspire new studies exploring these relationships more in-depth. My main criticism concerns the quality of reasoning and how and to what extent different issues are introduced and discussed. Proposing the use of certain group of species as indicators or other type of indicator species shall rest on solid grounds as well as need of such indicators should be clearly put forward. I simply miss the delineation of the conservation/management problem why we need indicators. There are many articles published on the worldwide decline of primary/intact forests, their characteristics and, in my opinion, you do not use the opportunity to discuss the potential role of woodpeckers/other tree cavity excavators (with very intimate linkages to natural forest characteristics) to support your case. Also, even if you mention their usefulness in management, there is lacking information on the intensively managed forests as areas that lose both important characteristics linked to old-growth forests and forests with high level of naturalness and thereby lose most specialized forest species often being woodpeckers. Also use of woodpeckers/other excavators as indicators of successful forest restoration could be mentioned. There is large body of literature on both above topics and I suggest that you should add paragraphs pertaining on these issues in both Introduction and Discussion sections. To limit the length of the article, you may get rid of several unnecessary repetitions. More suggestions below:

Detailed comments:

Line 99: Several species use smaller trees or snags too. Possibly add "often" or "usually".

Line 109: Woodpeckers (cavity excavators) are most often resident species what could additionally strengthen your reasoning on strong linkages to forest environments and their quality.

Lines 127-133: Please consult Wesolowski and Martin (2018) and extend the discussion on mechanism a bit more. I am lacking a clear statement on two possible, non-excluding each other mechanisms of expected relationships as very nicely stated way down in Discussion (Lines 312-314).

RESULTS: Could you add a summary table with numbers of species in different groups in different bioregions? It would provide some nice background for further discussions of results. You have those numbers already!

Lines 224-231: I do not understand how you can model the relationship between woodpeckers and the three other groups in Australasia, where, as you state below, there is no woodpeckers or even other strong excavators are largely lacking.

DISCUSSION: a bit confusing use of words "excavators/woodpeckers" (Line 244) and "excavators" (Line 275). Please check if necessary to use both of them!

Lines 239-243: Worthy to mention that forests in some bioregions supply decay-formed cavities in high numbers (particularly non-managed forests) and some other not (e.g. boreal forest in Europe – see Andersson et al. (2018): Andersson J., Domingo Gomez E., Michon S., Roberge J.-M. (2018) Tree cavity densities and characteristics in managed and unmanaged Swedish boreal forest. *Scandinavian Journal of Forest Research*, 33 (3), pp. 233-244. It would add the management angle in discussion about the role of woodpeckers/excavators for other species.

Lines 272-273: Please refer to studies on the global species richness of woodpeckers e.g. Mikusinski, G. 2006. Woodpeckers (Picidae) - distribution, conservation and research in a global perspective. *Annales Zoologici Fennici* 43: 86-95.

Lines 289-294: Refer also to the "primeval" Bialowieza Forest case in temperate zone being in stark contrast to managed forests (i.e. higher predation risk and subsequently preference for decay-formed cavities among secondary users. See: Wesolowski and Martin (2018) and references therein.

Lines 299-304: Please refer to papers by Wesolowski here: Wesolowski T. 2011. "Lifespan" of woodpecker-made holes in a primeval temperate forest: A thirty year study. *Forest Ecology and Management*, 262 (9) , pp. 1846-1852.

Reviewer: 2

Comments to the Author(s)

Many comments are made in the word file attached

Author's Response to Decision Letter for (RSOS-190962.R0)

See Appendix B.

RSOS-192177.R0

Review form: Reviewer 1

Is the manuscript scientifically sound in its present form?

Yes

Are the interpretations and conclusions justified by the results?

Yes

Is the language acceptable?

Yes

Do you have any ethical concerns with this paper?

No

Have you any concerns about statistical analyses in this paper?

No

Recommendation?

Accept with minor revision (please list in comments)

Comments to the Author(s)

Line 130: add a bracket at the end of the sentence

Lines 164-187: I opt for a very short explanation what do you mean by "universal" used in the title of the paper already here.

Lines 268-277: In contrast with added information about not analysing woodpeckers in Australasia (lines 236-238), you report it here and in Table 2. It is confusing to me.

Line 293: I have a hard time finding the paper by Tikkanen et al. 2006 as supporting your claim here. Possibly cite several paper based on single-species woodpecker studies or use some of those:

JM Roberge, P Angelstam, MA Villard. 2008. Specialised woodpeckers and naturalness in hemiboreal forests—deriving quantitative targets for conservation planning. *Biological conservation*, 2008

or

Angelstam, P., & Mikusiński, G. 1994: Woodpecker assemblages in natural and managed boreal and hemiboreal forest - a review. *Annales Zoologici Fennici* 31: 157-172.

Lines 392-394: I suggest adding further argument here i.e. that woodpeckers are highly responsive to playbacks (both calls and particularly drumming) that, if applied in right season are very effective survey tool and cover relatively large areas. See e.g. Kumar, R. and Singh, P. (2010), Determining woodpecker diversity in the sub-Himalayan forests of northern India using call playbacks. *Journal of Field Ornithology*, 81: 215-222. doi:10.1111/j.1557-9263.2009.00267.x or Jeremy A. Baumgardt, Joel D. Sauder, and Kerry L. Nicholson (2014) Occupancy Modeling of Woodpeckers: Maximizing Detections for Multiple Species With Multiple Spatial Scales. *Journal of Fish and Wildlife Management*: December 2014, Vol. 5, No. 2, pp. 198-207.

Table 2: remove data concerning the woodpeckers (see my comment above)

There is one important paper dealing with woodpeckers at the global scale that is relevant to your study but omitted namely: Vergara-Tabares, D. L., M.Lammertink, E. G.Verga, A.Schaaf, and J.Nori (2018). Gone with the forest: Assessing global woodpecker conservation from land use patterns. *Diversity and Distributions* 24:640–651.

Please consider mentioning the results of this paper in the Introduction or Discussion. It is particularly interesting from the management perspective and the human influence on forest qualities.

Review form: Reviewer 2

Is the manuscript scientifically sound in its present form?

No

Are the interpretations and conclusions justified by the results?

No

Is the language acceptable?

Yes

Do you have any ethical concerns with this paper?

No

Have you any concerns about statistical analyses in this paper?

Yes

Recommendation?

Major revision is needed (please make suggestions in comments)

Comments to the Author(s)

For the article to have the potential to be accepted, the authors must work, in my opinion, mainly on the following points:

1) work on the predictions, making them more precise, with more ecological language, and that revolve around the two main causes that the authors propose as the main drivers determining the relationships of excavators and forest birds: the sharing of habitat requirements, and the provision of excavated cavities. How these causes spatially vary based on geographic, climatic and historical characteristics in each region, will determine the predictions that will be tested.

2) rethink the overlapping of the bird categories, assessing which is the best option to determine which of the aforementioned factors (i.e. sharing of habitat requirements and cavity provisions), are the most likely drivers of the proposed relationships. As I comment in the text, the definition of the categories is inconsistent with the methodology used. And, if I understood well the methodology, I also have the impression that the overlapping of categories used may be influencing the results, inflating the relationships between excavators and forest birds, since in this last category are also included many SCNs and forest specialists. I have seen your response to a related comment I made in the previous version. I guess that a good possibility is that the only categories that overlap are 2 (SCN) and 3 (forest specialists). That way, you can see relationships that supposedly may be due more exclusively to provision of cavities (relationships of excavator with 2 would be stronger than with 3), or more related to sharing habitat requirements (relationships of excavator with 3 would be stronger than with 2). Also, leaving category 4 without overlapping can lead to different results and interpretations I think could be more interesting.

3) The discussion needs a lot of work. I would suggest that you focus on the differences between the zoogeographic regions, which is the main objective of the paper and what has been evaluated. The differences that may exist within each region should be minimized to the maximum, and only mentioned when strictly necessary. The discussion still has a great bias to try to explain the relationships between excavators and SCN, and there is a considerably smaller space in trying to explain the relationships with forest birds that do not nest in cavities. Finally, relationships in the Australasia region, which showed a different pattern from the rest (even in sign), were not explained, and these results are practically ignored when proposed, from the title, and the first and last paragraph of the discussion, that relationships are common to all zoogeographic regions.

Many other comments have been made in the attached file (Appendix C).

Decision letter (RSOS-192177.R0)

08-Jan-2020

Dear Dr van der Hoek,

The Subject Editor assigned to your paper ("Global and universal relationships between tree-cavity excavators and forest bird richness") has now received comments from reviewers. We would like you to revise your paper in accordance with the referee and Associate Editor suggestions which can be found below (not including confidential reports to the Editor). Please note this decision does not guarantee eventual acceptance.

Please submit a copy of your revised paper before 31-Jan-2020. Please note that the revision deadline will expire at 00.00am on this date. If we do not hear from you within this time then it will be assumed that the paper has been withdrawn. In exceptional circumstances, extensions may be possible if agreed with the Editorial Office in advance. We do not allow multiple rounds of revision so we urge you to make every effort to fully address all of the comments at this stage. If deemed necessary by the Editors, your manuscript will be sent back to one or more of the original reviewers for assessment. If the original reviewers are not available we may invite new reviewers.

When submitting your revised manuscript, you must respond to the comments made by the referees and upload a file "Response to Referees" in "Section 6 - File Upload". Please use this to document how you have responded to each of the comments, and the adjustments you have made. In order to expedite the processing of the revised manuscript, please be as specific as possible in your response.

- Ethics statement

- Data accessibility

It is a condition of publication that all supporting data are made available either as supplementary information or preferably in a suitable permanent repository. The data accessibility section should state where the article's supporting data can be accessed. This section should also include details, where possible of where to access other relevant research materials such as statistical tools, protocols, software etc can be accessed. If the data has been deposited in an external repository this section should list the database, accession number and link to the DOI for all data from the article that has been made publicly available. Data sets that have been

deposited in an external repository and have a DOI should also be appropriately cited in the manuscript and included in the reference list.

If you wish to submit your supporting data or code to Dryad (<http://datadryad.org/>), or modify your current submission to dryad, please use the following link:
<http://datadryad.org/submit?journalID=RSOS&manu=RSOS-192177>

- **Competing interests**

- **Authors' contributions**

- **Acknowledgements**

- **Funding statement**

Kind regards,

Andrew Dunn

on behalf of Prof Kevin Padian (Subject Editor)

Associate Editor Comments to Author:

The reviewers have provided further substantial feedback, but are broadly of the view that not only have you made efforts to improve the manuscript but that there is certainly merit in the general 'story' the manuscript is telling.

With this in mind, we'd like to invite you to submit a revised manuscript for further consideration. Bear in mind that we do not generally permit multiple rounds of revision, so

please do ensure that you incorporate the requested changes into the manuscript and provide an indication of how you've done so (using a tracked-changes version of the paper and a point-by-point response document).

Good luck!

Reviewer comments to Author:

Reviewer: 2

Comments to the Author(s)

For the article to have the potential to be accepted, the authors must work, in my opinion, mainly on the following points:

1) work on the predictions, making them more precise, with more ecological language, and that revolve around the two main causes that the authors propose as the main drivers determining the relationships of excavators and forest birds: the sharing of habitat requirements, and the provision of excavated cavities. How these causes spatially vary based on geographic, climatic and historical characteristics in each region, will determine the predictions that will be tested.

2) rethink the overlapping of the bird categories, assessing which is the best option to determine which of the aforementioned factors (i.e. sharing of habitat requirements and cavity provisions), are the most likely drivers of the proposed relationships. As I comment in the text, the definition of the categories is inconsistent with the methodology used. And, if I understood well the methodology, I also have the impression that the overlapping of categories used may be influencing the results, inflating the relationships between excavators and forest birds, since in this last category are also included many SCNs and forest specialists. I have seen your response to a related comment I made in the previous version. I guess that a good possibility is that the only categories that overlap are 2 (SCN) and 3 (forest specialists). That way, you can see relationships that supposedly may be due more exclusively to provision of cavities (relationships of excavator with 2 would be stronger than with 3), or more related to sharing habitat requirements (relationships of excavator with 3 would be stronger than with 2). Also, leaving category 4 without overlapping can lead to different results and interpretations I think could be more interesting.

3) The discussion needs a lot of work. I would suggest that you focus on the differences between the zoogeographic regions, which is the main objective of the paper and what has been evaluated. The differences that may exist within each region should be minimized to the maximum, and only mentioned when strictly necessary. The discussion still has a great bias to try to explain the relationships between excavators and SCN, and there is a considerably smaller space in trying to explain the relationships with forest birds that do not nest in cavities. Finally, relationships in the Australasia region, which showed a different pattern from the rest (even in sign), were not explained, and these results are practically ignored when proposed, from the title, and the first and last paragraph of the discussion, that relationships are common to all zoogeographic regions.

Many other comments have been made in the attached file

Reviewer: 1

Comments to the Author(s)

Line 130: add a bracket at the end of the sentence

Lines 164-187: I opt for a very short explanation what do you mean by "universal" used in the title of the paper already here.

Lines 268-277: In contrast with added information about not analysing woodpeckers in Australasia (lines 236-238), you report it here and in Table 2. It is confusing to me.

Line 293: I have a hard time finding the paper by Tikkanen et al. 2006 as supporting your claim here. Possibly cite several paper based on single-species woodpecker studies or use some of those:

JM Roberge, P Angelstam, MA Villard. 2008. Specialised woodpeckers and naturalness in hemiboreal forests—deriving quantitative targets for conservation planning. *Biological conservation*, 2008

or

Angelstam, P., & Mikusiński, G. 1994: Woodpecker assemblages in natural and managed boreal and hemiboreal forest - a review. *Annales Zoologici Fennici* 31: 157-172.

Lines 392-394: I suggest adding further argument here i.e. that woodpeckers are highly responsive to playbacks (both calls and particularly drumming) that, if applied in right season are very effective survey tool and cover relatively large areas. See e.g. Kumar, R. and Singh, P. (2010), Determining woodpecker diversity in the sub-Himalayan forests of northern India using call playbacks. *Journal of Field Ornithology*, 81: 215-222. doi:10.1111/j.1557-9263.2009.00267.x or Jeremy A. Baumgardt, Joel D. Sauder, and Kerry L. Nicholson (2014) Occupancy Modeling of Woodpeckers: Maximizing Detections for Multiple Species With Multiple Spatial Scales. *Journal of Fish and Wildlife Management*: December 2014, Vol. 5, No. 2, pp. 198-207.

Table 2: remove data concerning the woodpeckers (see my comment above)

There is one important paper dealing with woodpeckers at the global scale that is relevant to your study but omitted namely: Vergara-Tabares, D. L., M.Lammertink, E. G.Verga, A.Schaaf, and J.Nori (2018). Gone with the forest: Assessing global woodpecker conservation from land use patterns. *Diversity and Distributions* 24:640–651.

Please consider mentioning the results of this paper in the Introduction or Discussion. It is particularly interesting from the management perspective and the human influence on forest qualities.

Author's Response to Decision Letter for (RSOS-192177.R0)

See Appendix D.

RSOS-192177.R1 (Revision)

Review form: Reviewer 1

Is the manuscript scientifically sound in its present form?

Yes

Are the interpretations and conclusions justified by the results?

Yes

Is the language acceptable?

No

Do you have any ethical concerns with this paper?

No

Have you any concerns about statistical analyses in this paper?

No

Recommendation?

Accept with minor revision (please list in comments)

Comments to the Author(s)

Dear Authors,

Thanks for the new version of the manuscript that I find generally improved. I have some further suggestions (see below) and also think that the entire text should be thoroughly read by the native speaker; it is "sloppy" in several places.

Line 172: remove "comma" after "indicator"

Line 175: add "with" after "correlation"

Line 179: possibly, in global perspective, better to use broad-leaved since several coniferous trees may be "deciduous". Or change the classification to temperate, boreal, tropical etc.

Lines 185-190: poor English - check and re-write

Line 218: using "same species" is confusing here. What do you mean by "same species"?

Lines 219-220: provide names of the authors in the text

Line 292: add "i.e." before "between excavators..."

Line 295: was rho really 0.88 in both cases?

Line 355: higher temperature?

Line 363: "largely tropical" instead of "more tropical"?

Line 367: provide valid reference for this claim i.e. on high logging rates in Oriental Region

Line 373-377: it seems to be against your prediction. Whole sentence is a bit confusing.

Line 386: what dynamics do you mean? Fire dynamics?

Line 514-516: I would explicitly mention high detectability here

Review form: Reviewer 2

Is the manuscript scientifically sound in its present form?

No

Are the interpretations and conclusions justified by the results?

No

Is the language acceptable?

Yes

Do you have any ethical concerns with this paper?

No

Have you any concerns about statistical analyses in this paper?

Yes

Recommendation?

Major revision is needed (please make suggestions in comments)

Comments to the Author(s)

See attached (Appendix E).

Decision letter (RSOS-192177.R1)

04-Mar-2020

Dear Dr van der Hoek:

Manuscript ID RSOS-192177.R1 entitled "Global relationships between tree-cavity excavators and forest bird richness" which you submitted to Royal Society Open Science, has been reviewed. The comments of the reviewer(s) are included at the bottom of this letter.

Please submit a copy of your revised paper before 27-Mar-2020. Please note that the revision deadline will expire at 00.00am on this date. If we do not hear from you within this time then it will be assumed that the paper has been withdrawn. In exceptional circumstances, extensions may be possible if agreed with the Editorial Office in advance. We do not allow multiple rounds of revision so we urge you to make every effort to fully address all of the comments at this stage. If deemed necessary by the Editors, your manuscript will be sent back to one or more of the original reviewers for assessment. If the original reviewers are not available we may invite new reviewers.

- Ethics statement

- Data accessibility

- Competing interests

- Authors' contributions

- Acknowledgements

- Funding statement

on behalf of Kevin Padian (Subject Editor)
openscience@royalsociety.org

Associate Editor Comments to Author:

Associate Editor: 1

Comments to the Author:

As the reviewers acknowledge you've made 'good faith' efforts to improve your paper in response to their concerns, we'd like to offer you a final opportunity to revise your manuscript - this will be your last chance to persuade the reviewers that your manuscript is ready for acceptance: if you are unable to do so after revision, we may not be able to consider the paper further. We wish you every success in this revision, and look forward to receiving a new version of the paper in due course.

Reviewer comments to Author:

Reviewer: 2

Comments to the Author(s)

See attached. (RSOS-192177.R1_Proof_hi.pdf)

Reviewer: 1

Comments to the Author(s)

Dear Authors,

Thanks for the new version of the manuscript that I find generally improved. I have some further suggestions (see below) and also think that the entire text should be thoroughly read by the native speaker; it is "sloppy" in several places.

Line 172: remove "comma" after "indicator"

Line 175: add "with" after "correlation"

Line 179: possibly, in global perspective, better to use broad-leaved since several coniferous trees may be "deciduous". Or change the classification to temperate, boreal, tropical etc.

Lines 185-190: poor English – check and re-write

Line 218: using "same species" is confusing here. What do you mean by "same species"?

Lines 219-220: provide names of the authors in the text

Line 292: add "i.e." before "between excavators..."

Line 295: was rho really 0.88 in both cases?

Line 355: higher temperature?

Line 363: "largely tropical" instead of "more tropical"?

Line 367: provide valid reference for this claim i.e. on high logging rates in Oriental Region

Line 373-377: it seems to be against your prediction. Whole sentence is a bit confusing.

Line 386: what dynamics do you mean? Fire dynamics?

Line 514-516: I would explicitly mention high detectability here

Author's Response to Decision Letter for (RSOS-192177.R1)

See Appendix F.

RSOS-192177.R2 (Revision)

Review form: Reviewer 1

Is the manuscript scientifically sound in its present form?

Yes

Are the interpretations and conclusions justified by the results?

Yes

Is the language acceptable?

Yes

Do you have any ethical concerns with this paper?

No

Have you any concerns about statistical analyses in this paper?

No

Recommendation?

Accept with minor revision (please list in comments)

Comments to the Author(s)

I am generally pleased with changes included in this version of the manuscript. However, it would be great to carefully read some of the new sentences and improve them linguistically. For example check lines 312-319 and 363-368 (clean version) for missing words.

Decision letter (RSOS-192177.R2)

Dear Dr van der Hoek:

On behalf of the Editors, I am pleased to inform you that your Manuscript RSOS-192177.R2 entitled "Global relationships between tree-cavity excavators and forest bird richness" has been accepted for publication in Royal Society Open Science subject to minor revision in accordance with the referee suggestions. Please find the referees' comments at the end of this email.

The reviewers and Subject Editor have recommended publication, but also suggest some minor revisions to your manuscript. Therefore, I invite you to respond to the comments and revise your manuscript.

- Ethics statement

- Data accessibility

<http://datadryad.org/submit?journalID=RSOS&manu=RSOS-192177.R2>

- Competing interests

- Authors' contributions

All submissions, other than those with a single author, must include an Authors' Contributions section which individually lists the specific contribution of each author. The list of Authors

should meet all of the following criteria; 1) substantial contributions to conception and design, or acquisition of data, or analysis and interpretation of data; 2) drafting the article or revising it critically for important intellectual content; and 3) final approval of the version to be published.

- Acknowledgements

- Funding statement

Because the schedule for publication is very tight, it is a condition of publication that you submit the revised version of your manuscript before 19-Jun-2020. Please note that the revision deadline will expire at 00.00am on this date. If you do not think you will be able to meet this date please let me know immediately.

- 1) A text file of the manuscript (tex, txt, rtf, docx or doc), references, tables (including captions) and figure captions. Do not upload a PDF as your "Main Document".
- 2) A separate electronic file of each figure (EPS or print-quality PDF preferred (either format should be produced directly from original creation package), or original software format)
- 3) Included a 100 word media summary of your paper when requested at submission. Please ensure you have entered correct contact details (email, institution and telephone) in your user account
- 4) Included the raw data to support the claims made in your paper. You can either include your data as electronic supplementary material or upload to a repository and include the relevant doi within your manuscript

5) All supplementary materials accompanying an accepted article will be treated as in their final form. Note that the Royal Society will neither edit nor typeset supplementary material and it will be hosted as provided. Please ensure that the supplementary material includes the paper details where possible (authors, article title, journal name).

on behalf of Prof Kevin Padian (Subject Editor)
openscience@royalsociety.org

Associate Editor Comments to Author :

The reviewer is broadly satisfied with the scientific content of the paper, though they would prefer that you seek further English language support. Examples of professional services providing such advice may be found at <https://royalsociety.org/journals/authors/benefits/language-editing/>.

Reviewer comments to Author:
Reviewer: 1

Comments to the Author(s)

I am generally pleased with changes included in this version of the manuscript. However, it would be great to carefully read some of the new sentences and improve them linguistically. For example check lines 312-319 and 363-368 (clean version) for missing words.

Author's Response to Decision Letter for (RSOS-192177.R2)

See Appendix G.

Decision letter (RSOS-192177.R3)

Dear Dr van der Hoek,

It is a pleasure to accept your manuscript entitled "Global relationships between tree-cavity excavators and forest bird richness" in its current form for publication in Royal Society Open Science.

on behalf of Prof Kevin Padian (Subject Editor)
openscience@royalsociety.org

Appendix A

Global and universal relationships between tree-cavity excavators and forest bird richness

The overwhelming majority of tree-cavity excavators, among which woodpeckers are the most numerous group (van der Hoek, Gaona, & Martin, 2017), are forest or woodland birds. Indeed, general patterns of woodpecker richness correlate positively with amount of forest cover at the global scale (Ilsøe et al., 2017). However, many excavators are regarded as habitat specialists and have highly specific habitat requirements, among which the availability of dead wood is especially important (Hoyt & Hannon, 2002; Bütler, Angelstam, Ekelund, & Schlaepfer, 2004; Tikkanen et al., 2006). This group of birds utilizes large, live or decaying trees, or coarse woody debris, for nesting, roosting, and feeding. As a result, the prevalence of such attributes in a forest ecosystem may influence local richness and abundance of excavators (Martin & Eadie, 1999; Drever, Aitken, Norris, & Martin, 2008; Drever & Martin, 2010).

The strong attachment to specific forest elements, such as dead wood and large trees, makes excavators good indicators of the naturalness of ecosystems (Marchetti, 2004; Roberge & Angelstam, 2006; Virkkala, 2006; Drever et al., 2008; Drever & Martin, 2010), whereby their presence correlates strongly with other aspects of biodiversity (Mikusiński, Gromadzki, & Chylarecki, 2001; Nilsson, Hedin, & Niklasson, 2001; Angelstam et al. 2003). For example, studies in northern boreal and temperate regions revealed that many other forest birds respond similarly to habitat components required by woodpeckers, which leads to a strong correlation between woodpecker and forest bird richness and abundance at stand and landscape scales (Mikusiński et al., 2001; Roberge & Angelstam 2006; Drever et al., 2008). As a result, the responses of woodpeckers, and likely that of other tree-cavity-excavating birds (e.g.,

Commented [MOU1]: what is the difference between 'global' and 'universal'. There is nothing in the introduction, nor in the discussion, justifying the usage of these two words. I suggest to delete 'universal'

Commented [MOU2]: I see this paragraph as mostly unnecessary, with some of this information that can be fused with the second paragraph

Commented [MOU3]: you cited this article in the previous paragraph with the four authors, and here it is as 'et al.'. Please, correct where needed

Commented [MOU4]: you previously cited this article with the three authors, and here it is as 'et al.'. Please, correct where needed

barbets, trogons, nuthatches, and certain parrot species), to changes in forest habitats
may be indicative of responses of forest bird communities in general (Drever & Martin,
2010). Thus, excavators can be effective and efficient indicators at the forest ecosystem
scale, and thus useful as conservation and management targets (Roberge, Mikusiński, &
Svensson, 2008).

Excavators ~~are not only reliable indicators of other forest birds by means of their~~
~~associations with forest characteristics, but correlations between both groups of birds~~
~~are also partially determined by species interactions. Taken together, these and non-~~
~~excavator cavity nesters are linked into 'nest-webs' (sensu Martin and Eadie, 1999) of~~
~~interacting species, which~~ are structured by the availability and acquisition of tree-
cavities formed by two major processes: 'natural formation', in which fungi decompose
wood by fungi over an extended period, or animal excavation (Cockle, Martin, &
Wesolowski, 2011; Cockle, Martin, & Robledo, 2012; Ruggera et al. 2016). Since most
excavators—at least the majority of woodpeckers— excavate a new nest cavity each
40 year, this mode of cavity creation steadily provides a large number of potential sheltered
roosting and nesting sites for secondary-users: both for vertebrates (Cockle et al., 2012)
and invertebrates (Tylianakis, Klein, Lozada, & Tschamtko, 2006; Powell, Costa,
Lopes, & Vasconcelos, 2011). As a result, cavity formation by excavation is the main
source of nest holes for species in many regions of the world (Cockle et al., 2011), and
richness of cavity excavators is likely to be indicative of the richness of a broad range of
non-excavating tree-cavity nesters. However, decay-formed cavities might be very
common in other regions, especially the tropics, and it is very likely that the relative
importance of both processes of cavity formation—and thus the strength of the
relationship between excavator richness and non-excavator richness—varies globally
(Cockle et al., 2011).

Commented [MOU5]: I am not aware of this. I've seen some parrot species burrowing their own cavities in termite mounds and cliffs, or 'arranging' cavities already done by woodpeckers or fungi. But I have not seen/heard about parrots excavating 'brand new' cavities in trees. Please, could you add a reference for that?

Commented [MOU6]: I see this extrapolation from responses of a very specific bird group (i.e. cavity nesters) to responses of the whole forest bird community, a bit pretentious. Drever & Martin did not find that association either, as can be seen in their summary: "Recent studies indicated that species richness of woodpeckers was correlated with richness of all forest birds, thus suggesting potential exists for management practices that can address needs of woodpeckers in particular and other forest birds in general. ... These results, combined with previously identified positive correlations between woodpecker and forest bird richness, indicate woodpeckers can be managed as a suite for the purpose of managing avian biodiversity as a whole."

Besides, most of these extrapolations were done in north temperate forests (NA and Europe), which are structurally simpler than tropical forests. For example, excavators and understory birds might have similar responses to logging in north temperate forests, but different responses in tropical and subtropical forests. If the article is focused at a global scale, these considerations need to be taken into account.

So, unless you have a better argumentation for this association at a global scale, I suggest to delete this part

Commented [MOU7]: This is not what Cockle et al. (2011) found. According to the map in the Figure 2a, you can see that the pattern that you mention (excavated cavities as main source for non-excavator birds) is only strong in North America, very weak in Europe (only 1 out of 4 studies showed that pattern), and opposite in South America and Australia, this is, non-excavator birds mainly used decay-formed cavities (there is no study in Africa). After Cockle et al. (2011), Ruggera et al. (2016) quoted "Therefore, in the PF woodpeckers do not seem to play a key role in provisioning cavities to the nest-web, a pattern also shown for other South American and some European forests, and contrary to the findings in North America (Cockle et al., 2011a,b; Pereira et al., 2009; Wesolowski, 2007)."

Please, be careful when citing other authors, and put strictly what they have found

Finally, it is worth noting that ~~the diversity of excavators can be indicative of other~~
~~groups of species as~~ excavators can ~~be habitat engineers which~~ facilitate foraging of
other bird species by perforating the bark of trees to expose insects or sap (Bull &
Jackson, 1995; Montellano, Blendinger, & Macchi, 2013) and ~~which~~ play a role in the
dispersal of fungi which leads to wood softening, thus further aiding in the provision of
potential nest-cavities (Jusino et al., 2016).

Commented [MOU8]:

Local studies of cavity-nesting birds and their ~~community~~ interactions are largely
biased towards temperate zones, and only a few authors have explored global variation
in nest-web characteristics or addressed how such variation should be taken into
account when considering cavity-nester communities in forest conservation or
management (e.g., Cockle et al., 2011; ~~Ruggera et al. 2016~~). Given the significant role

Commented [MOU9]: I see this unnecessary at this point

that woodpeckers and other excavators play in forest ecosystems across most of the
world, we might predict that these birds could be reliable indicators of avian diversity
and richness not only at local and regional scales (Mikusiński et al., 2001; Martin,
Ibarra, & Drever, 2015), but also at a global scale. Therefore, the aim of the present

Commented [MOU10]: Please, see the previous comment on this pattern. Actually, it seems that woodpeckers play a considerably smaller role than that you stated, as cavity-providers in many non-disturbed ecosystems

study is to investigate whether there are consistent patterns in the relationships between
avian tree-cavity excavators (hereafter: excavators) and species richness of ~~non-~~
~~excavating tree-cavity nesters (hereafter: non-excavators)~~, forest specialist species, and
forest birds in general, across the globe. Given the important role of excavators in forest
ecosystems, we expect that non-excavator, forest specialist, and forest bird richness will
generally increase with the number of excavators present in the ecosystem, regardless of
the ecosystem or location. However, we also predict that the strength of the relationship
between excavators and richness of other forest birds will vary across zoogeographic
regions, following indications that the richness of different groups of tree-cavity nesting
birds (excavators, non-excavators) varies substantially across the world (van der Hoek

Commented [MOU11]: If my previous comment is correct, how is this prediction held?

Commented [MOU12]: I see necessary to call to this group with something that recalls that are cavity users, because both forest specialist species and forest birds in general include also non-excavator bird species. An option can be 'secondary cavity-nesters'

Commented [MOU13]: Here or in Methods: it's necessary to clarify if these categories are mutually exclusive. For example, some secondary cavity-nesters are forest specialists, and of course, are also forest birds in general. These species are considered in the three categories or only in the 'non-excavator' category? How this nested data set would eventually influence your results?

et al., 2017). If previously observed relationships between cavity-excavators and other
forest birds scale up from local forest stands or ecosystems to zoogeographic or global
levels, then this strengthens the potential of tree cavity-excavators as both indicators and
management surrogates.

Commented [MOU14]: what about the other two bird groups?

81 **Material and Methods**

We used published global distribution maps of tree-cavity nesting birds (van der Hoek
et al., 2017) and forest birds (Betts et al., 2017) to extract richness estimates per 10×10
84 km grid cell. Maps for both tree-cavity nesters and forest birds were based on the same
species range maps provided by BirdLife International and NatureServe (2015). Bird
species were classified in: 1) excavators: XXXXXX; 2) non-excavators: all birds known
to nest in tree cavities but which do not excavate their cavities, most of which are forest
birds; 3) forest specialists: birds that make exclusive use of forest; and 4) forest birds in
general: birds that use forest habitat and at least one other type of habitat. We used only
range polygons where a species' presence was classified as 'Extant' or 'Probably
extant', and assigned 'presence' to all grid cells that overlapped with range polygons.

Commented [MOU15]: I suggest to deeply restructure the introduction: 1) a first paragraph that talks in general about the relationship of excavators with other birds (fusion of the first two paragraphs); 2) Exclusive paragraphs for each excavator relationship with each of the other 3 groups of birds. In each paragraph express what is known and what is the relationship predicted by the authors, and how they think that these relationships may vary in relation to the biogeographic regions considered. Another option is that the geographical variations of the relationships are expressed together in a separate paragraph.

On the other hand, I think that it must necessarily be emphasized and made clear, what is the novelty provided by this study, in comparison with other studies cited in the same introduction and that address comparisons very similar to those of this study (eg Drever et al. 2008, Mikusinski et al 2001, Virkkala 2006).

Next, we created generalized least squares (gls) models to assess the linear relationships
between ~~excavators and the other bird categories~~ ~~excavators and all forest birds (birds~~
~~classified as using forest exclusively and those using forest habitat and at least one other~~
~~type of habitat; hereafter: forest birds), all forest specialists (birds classified as making~~
~~exclusive use of forest by Betts et al. (2017)), or all non-excavating cavity-nesters (all~~
~~birds known to nest in tree cavities but which do not excavate their cavities, most of~~
~~which are forest birds; hereafter: non-excavators)~~. To take biogeographical differences
into account, we proceeded to separate trends across the globe, and created gls models

Commented [MOU16]: I find necessary to clearly define the categories of bird groups you use. I tried to do this by copy-and-paste some information you wrote below. However, I don't know if I did it correct, and the definition of 'excavator' group is missing. Please, check it and correct it. Yet, something is not clear to me, Can a given bird species be part of more than one group? If the answer is yes, how that influences on results?

Commented [MOU17]:

for the six zoogeographic regions separately (Nearctic, Palearctic, Neotropical,
Afrotropical, Oriental, and Australasia).

As our data were spatially structured, we followed similar protocols to adjust for
spatial autocorrelation as per Storch et al. (2006). This implied the use of GLS models with
an exponential spatial covariance structure, as these models had the best fit in
preliminary tests, which reduced—rather than eliminated—the influence of spatial
autocorrelation on our models. In these tests, exponential GLS models consistently had
the lowest Akaike's Information Criterion (AIC) value compared to a set of models,
including an ordinary least squares model with no spatial structure, and GLS models with
Gaussian, linear, rational quadratics and spherical spatial covariance structures.

Even though GLS models work well with large datasets, we found that GLS models
fitted on all data failed to converge (using the R package “nlme” (Pinheiro et al., 2016)).
To circumvent this computational problem, we first grouped grid cells by
zoogeographic region. We thereafter ran 1000 models on bootstrap permutations of
1000 randomly selected grid cells (see Kissling, Sekercioglu, & Jetz, 2012; Ilsøe et al.,
2017). Of these 1000 model iterations, we subsequently calculated the mean (\pm SD),
minimum and maximum coefficients of the regression slope. Next, we obtained a
measure of the strength of each correlation by fitting each exponential GLS model to a
different, non-overlapping, test data set of 1000 randomly selected grid cells. We then
proceeded to calculate the Spearman rho coefficient of the correlation between
predicted and observed data, for all 1000 models per zoogeographic region. Finally, we
calculated the mean Spearman rho of all significant models per zoogeographic region.
We repeated all of these analyses for woodpeckers as a subset of the excavators. All
data, R scripts, and initial results needed to reproduce our analyses presented here are
deposited online (Appendix S1).

Commented [MOU18]: I find necessary to clarify some 'ecological' issues (i.e. not statistical) in methods. 1) Did you include the whole area of each biogeographic zone? or 2) only the forest areas in each one of them?
If (1): How do you think the different amount of area/forest area of each biogeographic zone can influence your results? How non-forest excavators (as some woodpeckers) can influence your results? or did you exclude them?
Another question: given the consideration of Palearctic and Oriental regions, what does the AUSTRALASIA region includes? what part of Asia?

Commented [MOU19]: What did you specifically model? which were the variables?

Commented [MOU20]: I am not an specialist in this kind of analyses, and I could not fully understand it. Given that many readers of this kind of papers are not specialists either, I find very important to better explain this methodological part. What are the predicted and what the observed data? What does tell us the regression slopes and what the mean spearman rho?

Nevertheless, at this point, I have the impression that you modeled the richness of non-excavators and forest birds, relative the presence of excavators (explanatory variable), i.e. in a causal manner (= GLS models). However, in the introduction this relationship was mostly raised in a correlational manner, this is, that both excavators and the other forest bird groups show similar responses to forest structure and disturbances. Exceptions to this correlational relationship are, for example, the FEW cases (in terms of global areas) in which excavators are essential for non-excavators by providing new cavities or food resources (sap and insects in excavator perforations). In short, you are analysing a relationship that you mostly posted as correlational, with regression models. Please, if my reasoning is incorrect, just ignore it, but explain better the methodology.

**Results**

~~We distilled global patterns in the r~~Relationships between excavator richness and that of
 non-excavators, all forest birds, and forest specialists (Fig. 1), ~~though relationships~~
 differed in relative strength and regression slope, ~~and also according to depending on~~
 ~~the group of birds and the~~ zoogeographic region analysed (Fig. 2, Table 1). ~~For~~
 ~~example, the R~~relationships between excavators and non-excavators and forest-
 associated birds were typified by particularly steep regression slopes in the Neotropics
 ~~and the other~~ predominantly tropical regions (Neotropical Afrotropical and Oriental).

In the Neotropics, the presence of one additional excavator species in a grid cell implied
 an increase of more than two additional non-excavators, ten forest bird species, or four
 forest specialists. In contrast, regression slopes were less steep in predominantly
 temperate and boreal regions such as the Nearctic and Palearctic, where an increase of
 one excavator species signified an increase of approximately one non-excavator species,
 four to five forest birds, or one forest specialist. Similar to variation in the slopes of the
 regressions, we found differences in the strength (as represented by Spearman's rho) of
 these relationships, with again stronger relationships in the predominantly tropical
 Afrotropical, Oriental, and Neotropical regions than in the comparably more
 temperate/boreal Nearctic and Palearctic regions (Table 1).

Although the relationships between excavators and the other three ~~bird groups of cavity-~~
 ~~nesting and forest-associated birds~~ were generally similar in nature (positive), there
 were some differences in the relative strength of these relationships across
 zoogeographic regions. For example, excavator richness was a particularly strong
 predictor of non-excavator richness and of all forest birds in the Oriental region (rho =
 0.88 and 0.86, respectively), and slightly less so for forest specialists (rho = 0.77),

Commented [MOU21]: I find Figures 1 and 2 redundant, given that in Fig. 1 you detailed ellipses for each of the six zoogeographic region. I'd say that you can leave only the figure 2 and nothing will be lost.

Commented [MOU22]: what does mean this category? specialists? all forest birds? both? you should be consistent with category names

Commented [MOU23]: I wouldn't say that Afrotropical and Oriental zones have also 'particularly steep slopes'. For example, in Table 1 can be seen that Australasian slopes are all steeper than Afrotropical and Oriental slopes. Also, Nearctic and Palearctic slopes of non-excavator are equal or steeper than that of Oriental, respectively; and that of forest specialist birds are also steeper or equal than that of Afrotropical, respectively. SO, the pattern you wrote is not as clear as you suggest.

Commented [MOU24]: It's hard to follow your reasoning. It seems like sometimes your reasoning is associative/correlative (e.g. both excavators and other groups of forest birds have similar responses to disturbances), other times it is explanatory (e.g. the excavator richness determines the richness of other non-excavator cavity-users by providing new cavities), and here it is predictive: the addition of 1 excavator species implies the addition of X forest bird species, ignoring any other type of causes, such as phylogenetic, biogeographical, etc. You must decide which of these visions you are going to explore, or clearly establish that all of them will be analysed, with their respective methodologies and theoretical support

Commented [MOU25]: What exactly does the Rho (=strength) mean in this context? low values of Rho mean that the relationships are not very reliable? for example in australasian region. And more important, what would be the joint interpretation of the slopes, the percentage of models with p greater than 0.5 and the rho?

whereas excavator richness in the Afrotropics showed a strong relationship with all
forest birds ($\rho = 0.94$) and with forest specialists ($\rho = 0.88$), but a somewhat weaker
association with non-excavators ($\rho = 0.79$).

We repeated all analyses for woodpeckers as a subset of excavators and found nearly
identical results, with again significant positive relationships between woodpeckers and
other groups of cavity-nesting or forest-associated birds in all regions but Australasia,
and with steeper regression slopes and stronger relationships in the predominantly
tropical Neotropical, Oriental and Afrotropical regions as compared to more shallow
slopes and weaker associations in the Nearctic and Palearctic (Table 2). The relative
strength of the relationships between woodpeckers and the three other groups of cavity-
nesting and forest-associated birds did again depend on the zoogeographic region (in
similar ways as for excavators), and woodpeckers were equally strong predictors of the
richness of other birds as excavators in general, with the notable exception of
Australasia. In this region, excavators showed a weak ($\rho = 0.69$) but often significant
($p < 0.05$ in 97 percent of model iterations) positive relationship with forest specialists,
whereas no clear relationship emerged between woodpeckers and forest specialists (ρ
$= -0.08$).

**Discussion**

We found strong positive relationships between excavators/woodpeckers and non-
excavators, forest birds and forest specialists at global and zoogeographic scales,
although these relationships differed in both strength and characteristics (regression
slopes) across zoogeographic regions. Our results imply that excavators and
woodpeckers (a predominant subset of excavators) hold potential as indicators and
management surrogates across most of the world (Hunter Jr et al., 2016), but also that

Commented [MOU26]: I did not understand the difference between the global and the zoogeographic scale of analysis. That is why I suggested that figures 1 and 2 are redundant. Furthermore, there are not numerical results for the global scale of analysis. So, I'd suggest to better clarify these results

we have to take differences among zoogeographic regions into account if we are going
to adopt these surrogates for conservation or management. Furthermore, the
zoogeographic variation among nest-web assemblages hints at underlying differences in
factors ranging from broad-scale patterns in evolutionary history (e.g., speciation) to
more local-scale variation in ecosystem characteristics and species traits (e.g., tree
cavity availability and a species' level of habitat specialization).

Commented [MOU27]: I would like to read what are those differences to take into account from a practical point of view

Many excavators, woodpeckers included (~~hereafter excavators/woodpeckers~~), have
strong preferences for specific forest elements, such as dead and decaying trees or
particularly large trees (e.g., Tikkanen et al. 2006). In turn, these elements, and the
processes that drive their presence and abundance, are of great importance to habitat
formation and create additional niche space to support many groups of organisms
(Stokland, Siitonen, & Jonsson, 2012). Therefore, we can expect that forests that meet
the habitat requirements of excavators (e.g., certain levels of availability of decaying
and dead wood) provide high quality habitat for many other organisms. This may
explain higher richness of forest birds (especially forest specialists) in areas occupied by
high numbers of excavators/woodpeckers. For non-excavating species that utilize tree-
cavities for their nests, this relationship may be enhanced by the fact that
excavators/woodpeckers create or facilitate potential nest substrates (Martin & Eadie,
1999). In other words, the strong habitat preferences of many excavators/woodpeckers,
as well as the facilitatory role excavators play in providing nest substrates, make these
birds potential indicators of many forest-associated species and, by extension, of high-
quality forests (i.e., natural, old growth; Drever et al. 2008).

Commented [MOU28]: The comment on nest-webs here, in the first paragraph of Discussion, is out of context. Besides, variation among nestwebs... in relation to what? Are you trying to relate the association of excavators and other forest birds with variations in nestweb characteristics? it's far from clear, and you are leaving out a key aspect: the diversity, persistence and characteristics of trees with cavities

The steep slopes we found for correlations between excavators/woodpeckers and
other forest birds in the predominantly tropical regions (Neotropics, Oriental,
Afrotropical) contrast to more shallow slopes in the temperate/boreal regions

Commented [MOU29]: However, in several other nestwebs (for example in South America [Cockle et al. 2012, Ruggera et al. 2016] and India [Manikandan & Balasubramanian 2018]), most non excavator birds mainly use decay-formed cavities in living trees. Only 10-25% of interactions (including excavator interactions) are made with snags. So, the evidence to date (even in the Neotropics, where your results showed the strongest association between excavators and non-excavators), would not be explained by the number of snags or the amount of woodpecker-excavated cavities. Also, you need to better explain why forest specialist birds would benefit from the presence of 'certain levels (how much?) of availability of decaying and dead wood'

(Palearctic, Nearctic). Macro-scale processes (e.g., as outlined by Gaston, 2000), such
 as speciation events, are likely to account for some of these differences among
 zoogeographic regions. First, overall species richness of birds (Blackburn & Gaston,
 1996), including cavity-nesting birds (van der Hoek et al., 2017), and also of trees as
 potential substrates for cavities (Currie & Paquin, 1987), differs both among and within
 zoogeographic regions, likely inducing differences in nest-web assemblages (see e.g.
 Cockle et al. (2012) for global differences in nest-web characteristics). However,
 variation in species composition and associated variation in functional richness (i.e.
 variety of functional traits), more than richness per se, is likely to contribute most to
 spatial variation in nest-web assemblages across global scales. For example, the
 Australasian region lacks the presence of Piciformes, the Order to which nearly 75% of
 excavators belong (van der Hoek et al., 2017), whereas speciation processes have led to
 a relatively high diversity of Piciformes in southeast Asia (Benz, Robbins, & Peterson,
 2006). Given that we found nearly similar results for woodpeckers and excavators as a
 larger group that includes e.g., barbets, except for in Australasia, we hereafter discuss
 these jointly as ‘excavators’.

We hypothesize that observed spatial variation in the relationships between
 excavators and other forest birds within zoogeographic regions stems more from local
 differences in habitat and vegetation characteristics (e.g. tree species composition and
 climatic variables (Huston, 1999), as well as tree cavity availability and their
 formation agents (Remm & Löhms 2011)) rather than from speciation processes. Such
 differences would for example explain how in the Neotropics, cavity-nesting
 assemblages studied in the temperate mountain forests of Chile had a 1:6 ratio of
 excavators (4 species) to non-excavators (25 species; Altamirano, Ibarra, Martin, &
 Bonacic, 2017) whereas a subtropical Atlantic moist forest supported relatively higher

Commented [MOU30]: differences in relation to what of the nestwebs? you need to be more specific

Commented [MOU31]: You need to better explain this: how speciation and other macro-scale processes across the globe influence on the results you found. Explanations generally are biased to nestwebs. However, you have other bird groups that not necessarily are cavity-nesters. Also, you don't necessarily have to invoke networks (of which there are relatively few reports worldwide) to talk about cavity-nesters (of which there are comparatively many more reports around the world)

Commented [MOU32]: Again, 1) variation in relation to what of the nestwebs? you need to be more specific. And 2) Explanations generally are biased to nestwebs, you don't need to invoke networks to talk about cavity-nesters

Commented [MOU33]: Australasia region doesn't lack Piciformes. As you say a few words later, SE Asia has a high diversity of Piciformes. So, be careful when you explain Australasia results as a unique region (see comment also about AUSTRALASIA in methods). In fact, I would like to find a better explanation of the results of the Australasian region, since it had the most contrasting results with the rest of the regions.

Commented [MOU34]: I honestly lost the point of the paragraph. I do not see a concrete explanation or discussion of some aspect of the reported results. I did not understand why the tropical regions had deeper slopes (although, see comment on results on the criticism of this pattern), I did not understand how to explain the flatter slopes of the temperate / cold regions of the north, nor did I find a convincing explanation of the contrasting results of the Australasian region

Commented [MOU35]: What are these 'within' variations? In results, you have always emphasized differences 'between' zoogeographic regions. Please, explain better

numbers of excavators with a 1:3 ratio of excavators (9) to non-excavators (25) (Cockle
et al., 2012). Similarly, we can hypothesize that at least some variation at regional
scales stems from differences in the propensities for re-use of excavated cavities by
non-excavators. For example, proportions of decay-formed cavities used are particularly
high in parts of the world with high amounts of precipitation (e.g., most of the wet
tropics), a reflection of fungal growth, and lower in anthropogenically disturbed
landscapes than in primary forests (Remm & Löhmus, 2011). In turn, a high availability
of decay-formed cavities might allow for a relatively high non-excavator richness, with
fewer species dependent on excavators for the supply of nest-cavities. For example,
even woodpeckers, commonly considered primary excavators, occasionally use decay-
formed cavities in Neotropical temperate rainforests where tree decay is the key driver
of nest-web structure (Altamirano et al., 2017). Finally, we find that relatively higher
numbers of non-excavators, or cavity-nesters in general, may occur in regions where
many species are habitat generalists that utilize other nest substrates than tree-cavities
(e.g., 40% of cavity nesters in Neotropical temperate forests are facultative users of tree
cavities for nesting; Altamirano et al., 2017), as compared to regions where most cavity
nesters are obligate tree-cavity users (e.g., over 85% of cavity nesters in temperate
Nearctic forests; Wesolowski & Martin, 2018). Finally, it is worth noting that cavity
availability and use are also likely to be influenced by cavity persistence (longevity), a
characteristic that can have considerable spatial variation due to factors such as land use
patterns (e.g., logging), climatic factors (e.g., rainfall), and cavity characteristics (e.g.,
whether they are located in live or dead trees; e.g., Edworthy, Wiebe, & Martin, 2012;
Cockle, Martin, & Bodrati, 2017).

We detected relative differences in strengths among the relationships between
excavators/woodpeckers and non-excavators, forest birds, and forest specialists across

Commented [MOU36]: what differences? you have not established what are the differences along the neotropical region in terms of climatic variables, tree species composition, tree availability and their formation agents. Also, include in this comparison the work by Ruggera et al. 2016, how their results fit in the explanation your trying to develop?

Commented [MOU37]: variation of what?

Commented [MOU38]: then, if non-excavators do not 'need' excavators as cavity-forming agents, why did they show the greatest association in richness in the Neotropics?

Commented [MOU39]: Also, Lammertink et al. 2019. Helmeted Woodpeckers roost in decay-formed cavities in large living trees: a clue to an old-growth forest association. Condor 121:1-10 DOI: 10.1093/condor/duy016

Commented [MOU40]: Altamirano et al. 2017 wrote: 'Among SCNs, ten species (40%) were obligate and 15 (60%) were non obligate cavity nesters. Non obligate cavity nesters included six (24%) facultative, five (20%) marginal, and four (16%) incidental SCN'. If we include also PCN, facultative cavity nesters are 20%.

Commented [MOU41]: how are they influenced? Again, I lost the point. You began talking about 'spatial variation in the relationships between excavators and other forest birds within zoogeographic regions', but then you focused (again) only on excavators and non-excavators, especially in the Neotropics.

zoogeographic regions. These differences were most likely to stem from the same
variation among species and ecosystem characteristics discussed above (e.g., the
distribution and relative importance of cavity forming agents), and indicate that richness
of excavators/woodpeckers may be a better indicator of richness of other cavity and
forest-associated species in one region than in another. In addition, the relative strength
of relationships between excavators and other forest birds might reflect whether
excavators are indicative of the richness of other species due to shared associations with
particular forest elements (indirect association), due to their direct facilitation of nest
cavities (direct facilitation), or whether both factors play a role. For example, we found
that relationships between excavator and forest bird or forest specialist richness in the
Afrotropics are relatively strong, whereas the relationship between excavators and non-
excavators is comparatively weaker. This might indicate that, in the Afrotropics,
excavators are especially strong predictors of other groups of forest-associated birds
through their mutual habitat requirements, rather than through their direct facilitation of
tree-cavities. In contrast, excavators are relatively stronger predictors of non-excavators
in the Oriental region, which might lead to hypotheses that excavators play a more
direct, nest-web structuring role through the provision of cavities in the Oriental region.
Some non-excavators might use cavities created by multiple excavator species,
whereas others predominantly use cavities formed by processes of degradation, damage
or insect activity (Martin & Eadie 1999; Cockle et al., 2012; Ruggera et al., 2016). In
addition, certain excavators might be especially abundant or provide cavities that are
very commonly reused by several species, whereas others are scarce and leave only a
few short-lived cavities that would not be generally available for use by non-excavators.
Given such complexities among nest-web interactions, it is important to emphasize that
we present general global patterns in the composition of tree-cavity-using assemblages

Commented [MOU42]: in which ones?

Commented [MOU43]: I believe that this is a central point, that it should be better developed in the introduction, and that the methodology should be chosen according to that. It is not the same to analyze something in a correlative way, that in an explanatory or predictive way

Commented [MOU44]: This does not make sense to me. The relationship is weak, and you argue that it is because they have similar requirements. However, if they have similar habitat requirements, it would not be logical to think that their richness should vary in a similar way?

Commented [MOU45]: however, see Manikandan & Balasubramanian 2018. Sequential Use of Tree Cavities by Birds and Nest Web in a Riparian Forest in Southwest India. Acta Ornithologica, 53(1):49-60.

Commented [MOU46]: All this is vague and inconclusive: it may be this but it may also be the other

(i.e., communities of linked species that often interact within networks), and do not
provide evidence for direct causal links between the presence of one group of species
and another or direct evidence of species interactions at the zoogeographic regional
scale.

Commented [MOU47]: I do not see an important point in this paragraph. I suggest to erase it completely

Excavators, of which woodpeckers are an important subset, appear to be useful
indicator species for forest bird diversity at multiple spatial scales, across the world.

Commented [MOU48]: which ones? you only analysed at the zoogeographic scale

They form a small and fairly easy to identify ~~(even by citizen scientists)~~ subset of the a
larger group of species (forest birds), often studied to understand the effects of
anthropogenic disturbances. Studies of excavators (or woodpeckers specifically),
especially in high biodiversity tropical regions, might provide quick measures of forest
quality and biodiversity. A comprehensive study of all forest birds in some regions like
Amazon or Congo basin would require considerable local expertise and training, which
is often difficult to achieve due to social and monetary limitations. Thus, we conclude,
excavators have excellent potential as study subjects for conservation monitoring and
planning initiatives, in comparative research, and to guide the establishment of region-
wide forest conservation strategies, especially in the largely understudied tropical
regions of the world.

**Acknowledgments**

We would like to thank Lisa Manne, and Kristina Cockle for their critical remarks on
drafts of this article.

**References**

Altamirano, T. A., Ibarra, J. T., Martin, K., & Bonacic, C. (2017). The conservation
value of tree decay processes as a key driver structuring tree cavity nest webs in

South American temperate rainforests. *Biodiversity and Conservation*, 26(10),
2453-2472.

**Angelstam**, P. K., Bütler, R., Lazdinis, M., Mikusiński, G., & Roberge, J. M. (2003;
**January**). Habitat thresholds for focal species at multiple scales and forest
biodiversity conservation—dead wood as an example. In *Annales zoologici fennici*
(pp. 473-482). Finnish Zoological and Botanical Publishing Board.

**Benz**, B. W., Robbins, M. B., & Peterson, A. T. (2006). Evolutionary history of
woodpeckers and allies (Aves: Picidae): placing key taxa on the phylogenetic tree.
*Molecular phylogenetics and evolution*, 40(2), 389-399.

**Betts**, M. G., Wolf, C., Ripple, W. J., Phalan, B., Millers, K. A., Duarte, A., ... & Levi,
310 T. (2017). Global forest loss disproportionately erodes biodiversity in intact
landscapes. *Nature*, 547(7664), 441.

BirdLife International, & NatureServe (2015). *Bird species distribution maps of the*
*world*. Cambridge and Arlington: BirdLife International and NatureServe.

**Blackburn**, T. M., & Gaston, K. J. (1996). Spatial patterns in the species richness of
birds in the New World. *Ecography*, 19(4), 369-376.

**Bull** E.L., & Jackson, J.A. (1995). Pileated Woodpecker (*Dryocopus pileatus*). No. 148.
In A. Poole, & F. Gill (Eds.), *The birds of North America*. Washington DC: The
Academy of Natural Sciences & The American Ornithologists' Union.

**Bütler** R., Angelstam P., Ekelund P., & Schlaepfer R. (2004). Dead wood threshold
values for the three-toed woodpecker presence in boreal and sub-Alpine forest.
*Biological Conservation*, 119, 305-318.

**Cockle**, K. L., Martin, K., & Bodrati, A. (2017). Persistence and loss of tree cavities
used by birds in the subtropical Atlantic Forest. *Forest ecology and management*,
384, 200-207.

Commented [MOU49]: Sometimes, there's a -, and sometimes there's a . Please, use only one

**Cockle**, K. L., Martin, K., & Wesolowski, T. (2011). Woodpeckers, decay, and the
future of cavity-nesting vertebrate communities worldwide. *Frontiers in Ecology*
*and the Environment*, 9(7), 377-382.

**Cockle**, K. L., Martin, K., & Robledo, G. (2012). Linking fungi, trees, and hole-using
birds in a Neotropical tree-cavity network: Pathways of cavity production and
implications for conservation. *Forest Ecology and Management*, 264, 210-219.

**Currie**, D. J., & Paquin, V. (1987). Large-scale biogeographical patterns of species
richness of trees. *Nature*, 329(6137), 326.

**Drever**, M. C., & Martin, K. (2010). Response of woodpeckers to changes in forest
health and harvest: Implications for conservation of avian biodiversity. *Forest*
*ecology and management*, 259(5), 958-966.

**Drever**, M. C., Aitken, K. E., Norris, A. R., & Martin, K. (2008). Woodpeckers as
reliable indicators of bird richness, forest health and harvest. *Biological*
*conservation*, 141(3), 624-634.

**Edworthy**, A. B., Wiebe, K. L., & Martin, K. (2012). Survival analysis of a critical
resource for cavity-nesting communities: patterns of tree cavity longevity.
*Ecological Applications*, 22(6), 1733-1742.

**Gaston**, K. J. (2000). Global patterns in biodiversity. *Nature*, 405(6783), 220.

Hoyt, J. S., & Hannon, S. J. (2002). Habitat associations of black-backed and three-toed
woodpeckers in the boreal forest of Alberta. *Canadian Journal of Forest Research*,
32(10), 1881-1888.

**Hunter** Jr, M., Westgate, M., Barton, P., Calhoun, A., Pierson, J., Tulloch, A., ... &
Heino, J. (2016). Two roles for ecological surrogacy: Indicator surrogates and
management surrogates. *Ecological Indicators*, 63, 121-125.

**Huston**, M. A. (1999). Local processes and regional patterns: appropriate scales for
understanding variation in the diversity of plants and animals. *Oikos*, 393-401.

**Ilsøe**, S. K., Kissling, W. D., Fjeldså, J., Sandel, B., & Svenning, J. C. (2017). Global
variation in woodpecker species richness shaped by tree availability. *Journal of*
*biogeography*, 44(8), 1824-1835.

**Jusino**, M. A., Lindner, D. L., Banik, M. T., Rose, K. R., & Walters, J. R. (2016).
Experimental evidence of a symbiosis between red-cockaded woodpeckers and
fungi. *Proceedings of the Royal Society B: Biological Sciences*, 283(1827),
20160106.

**Kissling**, W. D., Sekercioglu, C. H., & Jetz, W. (2012). Bird dietary guild richness
across latitudes, environments and biogeographic regions. *Global Ecology and*
*Biogeography*, 21(3), 328-340.

**Marchetti**, M. (Ed.). (2004). *Monitoring and indicators of forest biodiversity in Europe*
*– from ideas to operationality*. Joensuu, Finland: European Forest Institute

**Martin**, K., & Eadie, J. M. (1999). Nest webs: a community-wide approach to the
management and conservation of cavity-nesting forest birds. *Forest Ecology and*
*Management*, 115(2-3), 243-257.

**Martin** K., Ibarra J.T., & Drever, M. (2015). Avian surrogates in terrestrial ecosystems:
theory and practice. In D. Lindenmayer, P. Barton P, & J. Pierson (Eds.), *Indicators*
*and Surrogates in Ecology, Biodiversity and Environmental Management* (pp. 33-
34). Melbourne & London: CSIRO Publishing & CRC Press.

**Mikusinski**, G., Gromadzki, M., & Chylarecki, P. (2001). Woodpeckers as indicators of
forest bird diversity. *Conservation biology*, 15(1), 208-217.

**Montellano**, M. G. N., Blendinger, P. G., & Macchi, L. (2013). Sap consumption by the
White-fronted Woodpecker and its role in avian assemblage structure in dry forests.
*The Condor*, 115(1), 93-101.

**Nilsson**, S. G., Hedin, J., & Niklasson, M. (2001). Biodiversity and its assessment in
boreal and nemoral forests. *Scandinavian Journal of Forest Research*, 16(S3), 10-
26.

**Pinheiro** J., Bates D., DebRoy S., Sarkar D., Heisterkamp S., Van Willigen B., &
Maintainer, R. (2016). Package “nlme.” Available from [https://cran.r-](https://cran.r-project.org/web/packages/nlme/nlme.pdf)
[project.org/web/packages/nlme/nlme.pdf](https://cran.r-project.org/web/packages/nlme/nlme.pdf) (accessed January 15, 2018).

**Powell**, S., Costa, A. N., Lopes, C. T., & Vasconcelos, H. L. (2011). Canopy
connectivity and the availability of diverse nesting resources affect species
coexistence in arboreal ants. *Journal of Animal Ecology*, 80(2), 352-360.

**Remm**, J., & Löhmus, A. (2011). Tree cavities in forests—the broad distribution pattern
of a keystone structure for biodiversity. *Forest Ecology and Management*, 262(4),
579-585.

**Roberge**, J. M., & Angelstam, P. (2006). Indicator species among resident forest birds—a
cross- regional evaluation in northern Europe. *Biological Conservation*, 130(1),
134-147.

**Roberge**, J. M., Mikusiński, G., & Svensson, S. (2008). The white-backed woodpecker:
umbrella species for forest conservation planning?. *Biodiversity and conservation*,
17(10), 2479-2494.

**Ruggera**, R. A., Schaaf, A. A., Vivanco, C. G., Politi, N., & Rivera, L. O. (2016).
Exploring nest webs in more detail to improve forest management. *Forest Ecology*
*and Management*, 372, 93- 100.

**Stokland**, J. N., Siitonen, J., & Jonsson, B. G. (2012). *Biodiversity in dead wood*.
Cambridge, UK: Cambridge University Press.

**Storch**, D., Davies, R. G., Zajíček, S., Orme, C. D. L., Olson, V., Thomas, G. H., ... &
Blackburn, T. M. (2006). Energy, range dynamics and global species richness
patterns: reconciling mid-domain effects and environmental determinants of avian
diversity. *Ecology Letters*, 9(12), 1308-1320.

**Tikkanen**, O. P., Martikainen, P., Hyvärinen, E., Junninen, K., & Kouki, J. (2006;
~~January~~). Red- listed boreal forest species of Finland: associations with forest
structure, tree species, and decaying wood. In *Annales Zoologici Fennici* (pp. 373-
383). Finnish Zoological and Botanical Publishing Board.

**Tylianakis**, J. M., Klein, A. M., Lozada, T., & Tschamtker, T. (2006). Spatial scale of
observation affects α , β and γ diversity of cavity-nesting bees and wasps across a
tropical land-use gradient. *Journal of Biogeography*, 33(7), 1295-1304.

**van** der Hoek, Y., Gaona, G. V., & Martin, K. (2017). The diversity, distribution and
conservation status of the tree-cavity-nesting birds of the world. *Diversity and*
*Distributions*, 23(10), 1120-1131.

**Virkkala**, R. (2006, ~~January~~). Why study woodpeckers? The significance of
woodpeckers in forest ecosystems. In *Annales Zoologici Fennici* (pp. 82-85).
Finnish Zoological and Botanical Publishing Board.

**Wesołowski**, T., & Martin, K. (2018). Tree holes and hole-nesting birds in European
and North American Forests. In G. Mikusinski, J.-M. Roberge, & R. J. Fuller
(Eds.), *Ecology and Conservation of Forest Birds* (Chapter 4, pp 79-134).
Cambridge, UK: Cambridge University Press.

**Table 1. Results of generalized least square models of possible correlations**
 **between (1) excavator and non-excavator richness, (2) excavator and forest bird**
 **richness, (3) excavator and forest specialist richness.** Each result reflects the mean
 outcome of 1000 models build with 1000 randomly selected grid cell data points that
 include an exponential spatial covariance structure. The mean Spearman rho was
 calculated over the fit of each of 1000 models to a test data set.

Zoogeographic region	Correlation	Mean Slope (Min. – Max.; SD)	% of models with $P < 0.05$	Mean Spearman rho
Nearctic	1	0.8 (0.4 - 1.3; 0.1)	60	0.73
	2	4.3 (3.3 - 5.9; 0.4)	63	0.84
	3	1.4 (1.2 - 1.6; 0.1)	71	0.71
Palearctic	1	1.1 (0.7 - 1.4; 0.1)	84	0.78
	2	4.9 (4.0 - 6.7; 0.4)	73	0.85
	3	1.1 (0.8 - 1.7; 0.1)	83	0.79
Oriental	1	0.7 (0.6 - 0.9; 0.0)	100	0.88
	2	5.7 (4.9 - 6.6; 0.3)	100	0.86
	3	1.6 (1.3 - 1.9; 0.1)	100	0.77
Neotropical	1	2.4 (1.9 - 3.0; 0.2)	83	0.95
	2	10.8 (9.1 - 14.0; 0.7)	72	0.97
	3	4.2 (3.4 - 5.2; 0.2)	72	0.96
Afrotropical	1	1.1 (0.7 - 1.9; 0.2)	76	0.79
	2	6.8 (5.0 - 9.8; 0.9)	73	0.94
	3	0.9 (0.6 - 1.5; 0.1)	79	0.88
Australasian	1	1.8 (1.4 - 2.2; 0.1)	88	-0.14
	2	7.5 (2.3 - 11.2; 1.3)	100	0.31
	3	1.0 (-0.1 - 2.0; 0.3)	97	0.69

**Table 2. Results of generalized least square models of possible correlations**
 **between (1) woodpecker and non-excavator richness, (2) woodpecker and forest**
 **bird richness, and (3) woodpecker and forest specialist richness.** Each result reflects
 the mean outcome of 1000 models build with 1000 randomly selected grid cell data
 points that include an exponential spatial covariance structure. The mean Spearman rho
 was calculated over the fit of each of 1000 models to a test data set.

Zoogeographic region	Correlation	Mean Slope (Min. – Max.; SD)	% of models with $P < 0.05$	Mean Spearman rho
Nearctic	1	1.0 (0.5 - 1.6; 0.2)	61	0.70
	2	5.3 (3.7 - 8.0; 0.7)	67	0.79
	3	1.7 (1.1 - 2.2; 0.1)	70	0.63
Palearctic	1	1.2 (0.9 - 2.3; 0.1)	81	0.79
	2	5.6 (4.3 - 8.1; 0.6)	73	0.83
	3	1.3 (0.9 - 2.0; 0.2)	82	0.73
Oriental	1	1.0 (0.8 - 1.3; 0.1)	100	0.88
	2	7.9 (6.6 - 9.8; 0.5)	100	0.88
	3	2.2 (1.7 - 2.6; 0.1)	100	0.80
Neotropical	1	3.3 (2.4 - 4.9; 0.4)	68	0.94
	2	14.3 (10.8 - 20.5; 1.6)	67	0.93
	3	5.2 (3.9 - 6.6; 0.5)	73	0.88
Afrotropical	1	1.6 (0.9 - 3.5; 0.4)	72	0.74
	2	9.9 (6.4 - 19.6; 1.9)	69	0.86
	3	1.3 (0.7 - 3.2; 0.3)	80	0.81
Australasian	1	1.9 (-0.1 - 4.4; 0.6)	81	-0.33
	2	8.6 (1.2 - 21.8; 2.4)	75	-0.30
	3	3.9 (-0.2 - 9.0; 1.2)	96	-0.08

**Figure legends**

**Figure 1. Scatterplots indicating potential correlations between excavators and**

**non-excavators (a), excavators and all forest birds (b), and excavators and ~~all~~**

**forest specialist birds (c). Circular polygons represent the distribution of 95% of grid**

**cells pertaining to each of six zoogeographic regions. Points represent richness values in**

**10x10 km grid cells.**

**Figure 1.**

Commented [MOU50]: if something is 'circular', it cannot be a 'polygon'

**Figure 2. Scatter plots of the correlation between the richness (number of species)**
**of excavators and all forest birds (upper row), all forest specialists (middle row),**
**and all non- excavating cavity-nesters (bottom row) in six zoogeographic regions.**
Points represent species richness values in 10x10 km grid cells, and the solid line
represents the fit of a linear regression model, created using all grid cell values.

**Figure 2.**

Appendix B

Authors Response letter to editor:

I am recommending a "reject/resub" decision because our "revision" turnaround time may be too short for the authors' needs in revising, and because the reviewers have raised extensive concerns, despite being encouraging overall about the manuscript.

They see issues with how the main problem of the study is set out, the methods, the evaluation, and the conclusions. They also note a geographic bias that may entail ecological biases in the relationships of excavators to their ecosystems.

If you elect to re-submit, please detail your responses to reviewers' comments and make clear how you have altered your re-submitted manuscript. Best wishes with your revisions.

Reviewers' Comments to Author:

Reviewer: 1

General comment:

The manuscript entitled "Global and universal relationships between tree-cavity excavators and forest bird richness." reports on an interesting exploratory study linking woodpeckers/other tree cavity excavators, secondary cavity users and other forest tree species at the global and bioregional levels. The results are useful and may inspire new studies exploring these relationships more in-depth.

RESPONSE: We thank the reviewer for their supportive comments.

My main criticism concerns the quality of reasoning and how and to what extent different issues are introduced and discussed. Proposing the use of certain group of species as indicators or other type of indicator species shall rest on solid grounds as well as need of such indicators should be clearly put forward. I simply miss the delineation of the conservation/management problem why we need indicators. There are many articles published on the worldwide decline of primary/intact forests, their characteristics and, in my opinion, you do not use the opportunity to discuss the potential role of woodpeckers/other tree cavity excavators (with very intimate linkages to natural forest characteristics) to support your case. Also, even if you mention their usefulness in management, there is lacking information on the intensively managed forests as areas that lose both important characteristics linked to old-growth forests and forests with high level of naturalness and thereby lose most specialized forest species often being woodpeckers.

RESPONSE: We thank the reviewer for pointing out these ways to strengthen our text. We have added arguments and citations to strengthen these points throughout the Introduction (e.g., lines 96-98).

Also use of woodpeckers/other excavators as indicators of successful forest restoration could be mentioned. There is large body of literature on both above topics and I suggest that you should add paragraphs pertaining on these issues in both Introduction and Discussion sections. To limit the length of the article, you may get rid of several unnecessary repetitions. More suggestions below:

Detailed comments:

Line 99: Several species use smaller trees or snags too. Possibly add "often" or "usually".

RESPONSE: We agree with the reviewer that there are many exceptions, for which we followed the suggestion to add 'often' to soften the generalization.

Line 109: Woodpeckers (cavity excavators) are most often resident species what could additionally strengthen your reasoning on strong linkages to forest environments and their quality.

RESPONSE: We thank the reviewer for this valid point, and have incorporated a sentence to illustrate this in the second paragraph of our introduction.

Lines 127-133: Please consult Wesolowski and Martin (2018) and extend the discussion on mechanism a bit more. I am lacking a clear statement on two possible, non-excluding each other mechanisms of expected relationships as very nicely stated way down in Discussion (Lines 312-314).

RESPONSE: We thank the reviewer for this valid point. We agree that this is a key element of our manuscript that should be clarified much earlier in the text, and have a such included lines to address this topic in the very beginning of the Introduction.

RESULTS: Could you add a summary table with numbers of species in different groups in different bioregions? It would provide some nice background for further discussions of results. You have those numbers already!

RESPONSE: We understand that this might make it much easier to follow some of the reasoning, plus it would benefit authors who would like to build on our analyses. We have therefore followed your advice and added a supplementary table (Table S1) summarizing these data.

Lines 224-231: I do not understand how you can model the relationship between woodpeckers and the three other groups in Australasia, where, as you state below, there is no woodpeckers or even other strong excavators are largely lacking.

RESPONSE: Thank you for pointing this out, this is indeed a mistake. We have removed the 'woodpecker part' for Australasia and have added text to make this clear in methods as well ('We repeated all of these analyses for woodpeckers as a

subset of the excavators, with the notable exception of Australasia where woodpeckers are absent.')

DISCUSSION: a bit confusing use of words “excavators/woodpeckers” (Line 244) and “excavators” (Line 275). Please check if necessary to use both of them!

RESPONSE: We agree that this is a bit confusing and have simplified this, by simply using excavator/woodpecker when an argument or explanation concerns both groups equally and ‘excavator’ or ‘woodpecker’ by itself if it is particular to only one of these groups (e.g., when evidence does not extend beyond woodpeckers or when discussing specific results (e.g., ratios)).

Lines 239-243: Worthy to mention that forests in some bioregions supply decay-formed cavities in high numbers (particularly non-managed forests) and some other not (e.g. boreal forest in Europe – see Andersson et al. (2018): Andersson J., Domingo Gomez E., Michon S., Roberge J.-M. (2018) Tree cavity densities and characteristics in managed and unmanaged Swedish boreal forest. *Scandinavian Journal of Forest Research*, 33 (3), pp. 233-244. It would add the management angle in discussion about the role of woodpeckers/excavators for other species.

RESPONSE: We have added more text on this (esp. lines 304-309) including the reference suggested. We absolutely agree that this is a key element of the story, yet we aimed not to add too much to an already rather lengthy manuscript.

Lines 272-273: Please refer to studies on the global species richness of woodpeckers e.g. Mikusinski, G. 2006. Woodpeckers (Picidae) - distribution, conservation and research in a global perspective. *Annales Zoologici Fennici* 43: 86–95.

RESPONSE: We have added this reference.

Lines 289-294: Refer also to the “primeval” Bialowieza Forest case in temperate zone being in stark contrast to managed forests (i.e. higher predation risk and subsequently preference for decay-formed cavities among secondary users. See: Wesolowski and Martin (2018) and references therein.

RESPONSE: We have added a section on this, including additional references.

Lines 299-304: Please refer to papers by Wesolowski here: Wesolowski T. 2011. "Lifespan" of woodpecker-made holes in a primeval temperate forest: A thirty year study. *Forest Ecology and Management*, 262 (9) , pp. 1846-1852.

RESPONSE: We added this reference.

Reviewer 2:

What is the difference between ‘global’ and ‘universal’. There is nothing in the introduction, nor in the discussion, justifying the usage of these two words. I suggest to delete ‘universal’

RESPONSE: We thank the reviewer for this remark, it made it clear to us that we needed to clarify these terms. When we use the term ‘global’ we imply that these relationships can be found in nearly every region of the world, with ‘universal’ we mean that they are of similar nature (strong and positive).

We clarify these terms in both the Introduction and the Discussion (e.g., by adding that ‘Around the world, we can find relationships of a universal nature (strong and positive) between excavator and forest bird richness.’).

I see this paragraph as mostly unnecessary, with some of this information that can be fused with the second paragraph

RESPONSE: We understand that this start of the Introduction could be stronger, highlighting both the problem (need for indicators) and the opportunity (excavators/woodpeckers could make great indicators). We have reworked the introduction to accommodate these changes.

The reviewer mentions multiple inconsistencies in citations/references, we have now adopted the ‘Open Biology/Royal Society Open Science’ style and all issues should be resolved.

I am not aware of this. I’ve seen some parrot species burrowing their own cavities in termitaria and cliffs, or ‘arranging’ cavities already done by woodpeckers or fungi. But I have not seen/heard about parrots excavating ‘brand new’ cavities in trees. Please, could you add a reference for that?

RESPONSE: Although not common, parrots especially in Australasia may excavate in trunks that are in advanced stages of decay, without there having been a previous cavity, see e.g., Courtney, J. 2010. An Observation of Nesting Behaviour in Marshall's Fig-Parrot 'Cyclopsitta diophthalma marshalli'. Australian Field Ornithology, Vol. 27, No. 4, Dec 149-152.

.....So, unless you have a better argumentation for this association at a global scale, I suggest to delete this part.

RESPONSE: We have, in this new version, followed up on your suggestion and removed this statement that the response of woodpeckers can be indicative of that of forest birds in general.

This is not what Cockle et al. (2011) found. According to the map in the Figure 2a, you can see that the pattern that you mention (excavated cavities as main source for non-excavator birds) is only strong in North America, very weak in Europe (only 1 out of 4 studies showed that pattern), and opposite in South America and Australia, this is, non-excavator birds mainly used decay formed cavities (there is no study in Africa). After Cockle et al. (2011), Ruggera et al. (2016) quoted “Therefore, in the PF woodpeckers do not seem to play a key role in provisioning cavities to the nest-web, a pattern also shown for other South American and some European forests, and contrary to the findings in North America (Cockle et al., 2011a,b; Pereira et al., 2009; Wesolowski, 2007).” Please, be careful when citing other authors, and put strictly what they have found.

RESPONSE: We agree that we may have overstated the importance of excavators for the provision of cavities, and have altered our language accordingly (e.g., stating that ‘cavity formation by excavation is an importance source of nest holes for species in many regions of the world’). However, Cockle et al. has shown that woodpeckers are an important provider of cavities in farmlands and altered forests in Argentina. Thus, even in South America, woodpeckers may provide a larger role in the provisioning of tree cavities than currently recognized.

- Please, see the previous comment on this pattern. Actually, it seems that woodpeckers play a considerably smaller role than that you stated, as cavity providers in many non-disturbed ecosystems. If my previous comment is correct, how is this prediction held?

RESPONSE: We think this new version of our manuscript does a better job at highlighting the two mechanisms by which excavators are indicators: due to their associations with particular forest elements and their role in facilitating the presence of other species through the provision of nest cavities. We therefore think this prediction is still valid, but we did alter the text to recognize this point:

“Nevertheless, we might predict that these birds could be reliable indicators of avian diversity and richness not only at stand or landscape-levels (8, 17), but also across large regions at a global scale, not only for the cavities that excavators supply but also due to excavators’ associations with forest elements that are indicative of heterogeneous forests with a high degree of naturalness.”

I see necessary to call to this group with something that recalls that are cavity users, because both forest specialist species and forest birds in general include also non-excavator bird species. An option can be ‘secondary cavity-nesters’

RESPONSE: We have followed this useful suggestion and have changed non-excavator to ‘secondary cavity-nester’ throughout the text.

Here or in Methods: it’s necessary to clarify if these categories are mutually exclusive. For example, some secondary cavity-nesters are forest specialists, and of course, are also forest birds in general. These species are considered in the three categories or only in the ‘non-excavator’ category? How this nested data set would eventually influence your results?

RESPONSE: We added a sentence in the Methods to explain that some of these numbers are indeed nested, with for example a forest specialist also being counted as a forest bird. We can see how it could be interesting to know (for example) details on the relationship between excavators and forest specialists that do not nest in cavities, which we do not provide at this point. One reason for this is that sample size for many grid cells would become too low for our models to converge, particularly when we would count forest specialists that do not breed in cavities as a separate group (e.g., there are relatively few forest specialists in the Nearctic that are not also cavity-nesters).

Another reason is that we aimed to know whether excavators make useful indicators and management surrogates. For this we wanted to know how indicative excavators are of larger and more inclusive suites of forest-associated birds, not of very specific subgroups (e.g., forest-associated species that do not breed in cavities and that do not exclusively use forest habitat).

What about the other two bird groups?

RESPONSE: We have altered this text to make clear we predict that spatial variation in forest characteristics, management regimes, and bird communities will lead to differences in relationships between excavators and forest birds.

- I suggest to deeply restructure the introduction: 1) a first paragraph that talks in general about the relationship of excavators with other birds (fusion of the first two paragraphs); 2) Exclusive paragraphs for each excavator relationship with each of the other 3 groups of birds. In each paragraph express what is known and what is the relationship predicted by the authors, and how they think that these relationships may vary in relation to the biogeographic regions considered. Another option is that the geographical variations of the relationships are expressed together in a separate paragraph. On the other hand, I think that it must necessarily be emphasized and made clear, what is the novelty provided by this study, in comparison with other studies cited in the same introduction and that address comparisons very similar to those of this study (eg Drever et al. 2008, Mikusinski et al 2001, Virkkala 2006).

RESPONSE: We thank both Reviewer 1 and 2 for this suggestion for restructuring, and have adopted these suggestions. We have given more attention to the novelty of our study in the very first paragraph of our new version (Larger scale, testing for general patterns across all major biogeographic areas), and have further regrouped and organized the paragraphs in the introduction.

- I find necessary to clearly define the categories of bird groups you use. I tried to do this by copy-and-paste some information you wrote below. However, I don't know if I did it correct, and the definition of 'excavator' group is missing. Please, check it and correct it. Yet, something is not clear to me, Can a given bird species be part of more than one group? If the answer is yes, how that influences on results?

RESPONSE: We have added some text to clarify this in the methods and provided clear definitions of all bird groups used (see also our response to an earlier query above.

- I find necessary to clarify some 'ecological' issues (i.e. not statistical) in methods. 1) Did you include the whole area of each biogeographic zone? or 2) only the forest areas in each one of them? If (1): How do you think the different amount of area/forest area of each biogeographic zone can influence your results? How non-forest excavators (as some woodpeckers) can influence your results? or did you exclude them? Another question: given the consideration of Palearctic and Oriental regions, what does the AUSTRALASIA region includes? what part of Asia?

RESPONSE: We thank the reviewer for this question. We have added text to explain this in the Methods: "We did not restrict our analyses to forested regions, wanting to avoid making subjective decisions on the classification of a grid cell to a specific vegetation/habitat type. Yet, to make sure we did look for possible relationships between these forest-associated species in regions they actually occur, we did remove cells with zero excavators/woodpeckers and cells where the numbers of secondary cavity-nesters, forest specialists, or all forest birds were lower than that of excavators/woodpeckers from further analyses. "

That said, we are not sure how the amount of area/forest area would influence our results, but we have aimed to circumvent that issue as much as possible by randomly selecting (1000 times) 1000 grid

cells per region, thereby comparing similar surface (though not necessarily forest, but see above) areas per region.

Australasia encompasses the following regions, following our adaptation of the classification of van der Hoek et al. 2017. Diversity and Distributions, which is in turn based on that of the Handbook of the Birds of the World (Del Hoyo et al.): American Samoa, Australia, Bismarck Archipelago, Admiralty Islands, Cook Islands, Easter Island, Federated States Micronesia, Fiji, French Polynesia, Guam and the Marianas, Johnston Island, Kiribati, Macquarie Island, Marshall Islands, Nauru, New Caledonia, New Guinea, New Zealand, Niue, Norfolk Island, Papua New Guinea, Pitcairn Islands, Samoa, Solomon Archipelago, Tokelau, Tonga, Tuvalu, Vanuatu, Wake Island, Wallis and Futuna. Practically, we can say that most excavators in this region are from (Papua) New Guinea and surrounding islands.

- What did you specifically model? which were the variables?

RESPONSE: The gls models are more clearly explained now: “ This implied the use of gls models with excavator richness as a predictor variable, the richness of one of the categories of forest-associated birds as a response variable, and a variable exponential spatial covariance structure...”

- I am not an specialist in this kind of analyses, and I could not fully understand it. Given that many readers of this kind of papers are not specialists either, I find very important to better explain this methodological part. What are the predicted and what the observed data? What does tell us the regression slopes and what the mean spearman rho? Nevertheless, at this point, I have the impression that you modeled the richness of non-excavators and forest birds, relative the presence of excavators (explanatory variable), i.e. in a causal manner (= gls models). However, in the introduction this relationship was mostly raised in a correlational manner, this is, that both excavators and the other forest bird groups show similar responses to forest structure and disturbances. Exceptions to this correlational relationship are, for example, the FEW cases (in terms of global areas) in which excavators are essential for nonexcavators by providing new cavities or food resources (sap and insects in excavator perforations). In short, you are analysing a relationship that you mostly posted as correlational, with regression models. Please, if my reasoning is incorrect, just ignore it, but explain better the methodology.

RESPONSE: We have added some clarification (see e.g., the answer above) that may help understand these methods better.

That said, it is correct that a gls model is basically an ordinary linear regression model with some different assumptions (or some assumptions not met; in this case related to spatial autocorrelation). However, whereas the reasoning behind a correlation vs. a regression differ, the statistics are, in this particular case, the same. A correlation can indeed encompass any relationship between two variables, without assuming cause and effect, whereas a linear regression can only be that: a linear relationship where we assume that X predicts Y. When the correlation is linear though, this becomes one and the same in terms of output (though not interpretation, which we therefore have been careful with, see our Discussion). In other words, when in this particular case both tests address the ‘null hypothesis that the two variables are not linearly related. If run on the same data, a correlation

test and slope test provide the same test statistic and p-value.' (see e.g., <http://sites.utexas.edu/sos/guided/inferential/numeric/bivariate/cor/>).

So why then this regression and not a correlation, if they provide the same outcome but correlations are easier to interpret? Largely because we would be unable to incorporate a measure to address spatial autocorrelation into a correlation test.

- given that in Fig. 1 you detailed ellipses for each of the six zoogeographic region. I'd say that you can leave only the figure 2 and nothing would be lost

RESPONSE: We have followed this suggestion and removed the first of the two figures, leaving only the multi-paneled one. That said, we opted to add one map that shows global distributions of excavators relative to secondary cavity-nesters, a map which we deem to be illustrative of the spatial variation and clustering (e.g., between zoogeographic regions) that we discuss in our manuscript.

- It seems like sometimes your reasoning is associative/correlative (e.g. both excavators and other groups of forest birds have similar responses to disturbances), other times it is explanatory (e.g. the excavator richness determines the richness of other non excavator cavity-users by providing new cavities), and here it is predictive: the addition of 1 excavator species implies the addition of X forest bird species, ignoring any other type of causes, such as phylogenetic, biogeographical, etc. You must decide which of these visions you are going to explore, or clearly establish that all of them will be analysed, with their respective methodologies and theoretical support.

RESPONSE: Although regressions provide both explanatory and predictive outcomes we understand that our language here is confusing, and our statements may be misinterpreted as meaning one can simply predict from a given number of excavators how many other forest birds there are at that location (which is, of course, not that simple as you rightly point out). Yet, whether the tested relationships are associative (similar habitat needs etc.) or causative (birds use the cavities created by excavators), the fact of the matter is that they are linear. But, in some regions, this relationship is stronger than in others. We should have focused on this and not the steepness of the slopes; and we have therefore removed these parts from the Results and Discussion sections.

That said, we explore whether there are relationships between excavators and other forest birds (the GLS models themselves), but also aimed to see if this could then be used to determine to what extent excavator richness is indicative of that of other species (the matching with the test data set and derivation of Spearman rho's). We hope that this approach is clearer from our revised manuscript.

- Commented [MOU25]: What exactly does the Rho (=strength) mean in this context? low values of Rho mean that the relationships are not very reliable? for example in australasian region. And more important, what would be the joint interpretation of the slopes, the percentage of models with p greater than 0.5 and the rho?

RESPONSE: Rho would here give us an idea how 'perfect' the linear relationships are (with rho = 1 being perfect with all grid cells falling on one line), whereas the slope (which we now removed from further discussion (see above) but present in the Results section to be complete in our presentation of model statistics) would tell us how many other forest birds (other than excavators) there are likely to be in a given location (grid cell) relative to the number of excavators (this does thus present an idea of

‘what richness’ a certain number of excavators would possibly indicate). The % of models with a $p < 0.05$ gives us an idea whether these relationships are significant or not, given we had to iterate over many (1000) models, this would be a measure of reliability of the relationships given the combination of both the correlation coefficient and the sample size (which here is always 1000).

Low values of rho do not necessarily mean that that the relationships are not very reliable, but rather that they would not be strong enough to be able to ‘use richness of excavators’ to predict how many other forest birds are in a given location (here grid cell). Thus low values of rho would, in this case, show that excavators are rather poor indicators of other forest birds, even though the relationships might be real and significant.

- Commented [MOU27]: I would like to read what are those differences to take into account from a practical point of view and Commented [MOU28]: The comment on nest-webs here, in the first paragraph of Discussion, is out of context. Besides, variation among nestwebs... in relation to what? Are you trying to relate the association of excavators and other forest birds with variations in nestweb characteristics? it's far from clear, and you are leaving out a key aspect: the diversity, persistence and characteristics of trees with cavities

RESPONSE: We have made some substantial changes to our text and hope to now provide a clearer line of argumentation.

- However, in several other nestwebs (for example in South America [Cockle et al. 2012, Ruggera et al. 2016] and India [Manikandan & Balasubramanian 2018]), most non excavator birds mainly use decay-formed cavities in living trees. Only 10-25% of interactions (including excavator interactions) are made with snags. So, the evidence to date (even in the Neotropics, where your results showed the strongest association between excavators and non-excavators), would not be explained by the number of snags or the amount of woodpecker-excavated cavities. Also, you need to better explain why forest specialist birds would benefit from the presence of ‘certain levels (how much?) of availability of decaying and dead wood’

RESPONSE: We have made substantial changes to the text in the Discussion, incorporating suggestions by both reviewers, which means that we now provide a clearer insight in the differences in the importance of decay vs. excavated cavities in the different regions. With that, we anticipate to have resolved these issues as well.

- differences in relation to what of the nestwebs? you need to be more specific. You need to better explain this: how speciation and other macro-scale processes across the globe influence on the results you found. Explanations generally are biased to nestwebs. However, you have other bird groups that not necessarily are cavity-nesters. Also, you don't necessarily have to invoke networks (of which there are relatively few reports worldwide) to talk about cavity-nesters (of which there are comparatively many more reports around the world)

RESPONSE: We thank the reviewer for this comment. We have made substantial changes to our text, reflecting better that both direct facilitation through nest-webs and indirect associations play a role in shaping the observed relationships. As a result of these changes, we have also paid more attention to the role of differences in community composition etc. in shaping these relationships.

- Australasia region doesn't lack Piciformes. As you say a few words later, SE Asia has a high diversity of Piciformes. So, be careful when you explain Australasia results as a unique region (see comment also about AUSTRALASIA in methods). In fact, I would like to find a better explanation of the results of the Australasian region, since it had the most contrasting results with the rest of the regions.

RESPONSE: There is a difference between the Oriental and Australasian zoogeographic regions that may have gone unnoticed to the reviewer. Most of Indonesia and other SE Asian countries falls within the Oriental region, with the Wallace line separating this region from the Australasian region (e.g., Papua/New Guinea) where some excavators exist but Piciformes are notably absent (see also e.g., Mikusiński, G. (2006, January). Woodpeckers: distribution, conservation, and research in a global perspective. In *Annales Zoologici Fennici* (pp. 86-95). Finnish Zoological and Botanical Publishing Board.). That said, we do agree that the results we found for the Australasian region deserved more attention, and have thus extended on this in the text.

- what differences? you have not established what are the differences along the neotropical region in terms of climatic variables, tree species composition, tree availability and their formation agents. Also, include in this comparison the work by Ruggera et al. 2016, how their results fit in the explanation your trying to develop?

RESPONSE: In our revised version you will find references to Ruggera et al. as well as more comprehensive explanation of the differences found across and within regions.

- then, if non-excavators do not 'need' excavators as cavity-forming agents, why did they show the greatest association in richness in the Neotropics?

RESPONSE: We anticipate that we have now explained this better, focusing on the fact that indirect associations (shared forest elements etc.) are more important than direct facilitation (cavities) in explaining these relationships.

- Commented [MOU40]: Altamirano et al. 2017 wrote: 'Among SCNs, ten species (40%) were obligate and 15 (60%) were non obligate cavity nesters. Non obligate cavity nesters included six (24%) facultative, five (20%) marginal, and four (16%) incidental SCN'. If we include also PCN, facultative cavity nesters are 20%. Commented [MOU41]: how are they influenced? Again, I lost the point. You began talking about 'spatial variation in the relationships between excavators and other forest birds within zoogeographic regions', but then you focused (again) only on excavators and non-excavators, especially in the Neotropics.

RESPONSE: We have changed these parts of text substantially, hopefully addressing these issues and providing more clarity.

- I believe that this is a central point, that it should be better developed in the introduction, and that the methodology should be chosen according to that. It is not the same to analyze something in a correlative way, that in an explanatory or predictive way

RESPONSE: Yes, we agree. We have made an effort to provide more attention to these two different mechanisms (direct facilitation and indirect association) by which relationships between excavators and other forest birds may be formed.

Appendix C**ROYAL SOCIETY
OPEN SCIENCE****Global and universal relationships between tree-cavity
excavators and forest bird richness**

Journal:	Royal Society Open Science
Manuscript ID	RSOS-192177
Article Type:	Research
Date Submitted by the Author:	13-Dec-2019
Complete List of Authors:	van der Hoek, Yntze; The Dian Fossey Gorilla Fund International, Karisoke Research Center Gaona, Gabriel; Justus Liebig Universität Giessen Graduiertenzentrum Ciach, Michal; University of Agriculture in Krakow Martin, Kathy; University of British Columbia,
Subject:	ecology < BIOLOGY
Keywords:	Facilitator species, Indicator species, Species interactions, Picidae, Management surrogates, Woodpeckers
Subject Category:	Biology (whole organism)

Author-supplied statements

Relevant information will appear here if provided.

Ethics

Does your article include research that required ethical approval or permits?:

This article does not present research with ethical considerations

Statement (if applicable):

CUST_IF_YES_ETHICS :No data available.

Data

It is a condition of publication that data, code and materials supporting your paper are made publicly available. Does your paper present new data?:

Yes

Statement (if applicable):

Links to the data and R scripts to reproduce analyses presented in Global and universal relationships between tree-cavity excavators and forest bird richness can be found in Appendix S1.

Conflict of interest

I/We declare we have no competing interests

Statement (if applicable):

CUST_STATE_CONFLICT :No data available.

Authors' contributions

This paper has multiple authors and our individual contributions were as below

Statement (if applicable):

YH and GVG carried out the data analysis, and led the design of the study and writing of the manuscript; MC and KM critically revised the manuscript. All authors contribute to the writing of the final manuscript, gave final approval for publication and agree to be held accountable for the work performed therein.

Title: Global and  relationships between tree-cavity excavators and forest bird richness

Running title: Excavator and forest bird relationships

Yntze van der Hoek^{1,2*}, Gabriel V. Gaona¹, Michał Ciach³ and Kathy Martin^{4,5}

¹ Universidad Regional Amazónica Ikiam, Vía Muyuna, Kilómetro 7, Tena, Ecuador.

² The Dian Fossey Gorilla Fund International, Musanze, Rwanda.

³ Department of Forest Biodiversity, Institute of Forest Ecology and Silviculture, Faculty of
Forestry, University of Agriculture, al. 29 Listopada 46, 31-425 Kraków, Poland.

⁴ Department of Forest and Conservation Sciences, University of British Columbia, 2424 Main
Mall, Vancouver, British Columbia V6T 1Z4 Canada.

⁵ Environment and Climate Change Canada, 5421 Robertson Road, R.R. 1, Delta, British
Columbia V4K 3N2, Canada.

*Corresponding author: yntzevanderhoek@gmail.com

Abstract

Global monitoring of biodiversity and ecosystem change can be aided by the effective use of indicators. Tree-cavity excavators, the majority of which are woodpeckers (Picidae), are known to be useful indicators of the health or naturalness of forest ecosystems and by the diversity of forest birds. They are indicators of the latter due to their associations with particular forest elements and because of their role in facilitating other species through the provision of nest cavities. Here, we investigated whether these positive correlations between excavators and other forest birds are also found at global scales. We used global distribution maps to extract richness estimates of tree-cavity nesting and forest-associated birds, which we grouped by zoogeographic regions. We then created generalized least squares (gls) models to assess the relationships between these groups of birds. We show that richness of tree-cavity excavating birds correlates positively with that of cavity-nesters and forest birds at global scales, but with variation across zoogeographic regions. As many excavators are relatively easy to detect, play keystone roles at local scales, and are effective management targets, we propose that excavators are useful for biodiversity monitoring across multiple spatial scales and geographic regions, especially in the tropics.

Keywords

Facilitator species, Indicator species, Management surrogates, Picidae, Species interactions, Woodpeckers

Introduction

Forests worldwide are facing increasing anthropogenic pressures, with both a rapid decline in the
surface area of natural forests and a decrease in the naturalness of forests as a result [1, 2]. In
turn, these reductions of high-quality forest habitats have led to a loss of forest-associated
biodiversity [3-5]. To monitor these changes in forest ecosystems and their denizens, we often
look at specialized animals that require the availability of specific habitat structures and
processes across long temporal and large spatial scales [6], such as birds that respond not only to
a loss in overall forest cover but also to a change in forest health, quality, and integrity [7-10].
Excavators such as woodpeckers, barbets, nuthatches, trogons and certain parrot species—among
which woodpeckers are the most numerous group [11]—may be especially effective indicators
due to their associations with particular forest elements and their role in facilitating the presence
of other species through the provision of nest cavities. Indeed, there is empirical evidence that
their presence can be indicative of the state of both the forest (e.g., the presence of large trees
[12], heterogeneous forest structure [13], or a high level of naturalness [14-18]) as well as the
richness and abundance of other forest-associated species [19-21]. However, we may question
whether excavators are universally effective indicators, at least of biodiversity, in all geographic
regions or forest types. For example, excavators are better predictors of richness and abundance
of other forest resident birds in deciduous versus coniferous forests of certain parts of hemiboreal
Europe [15].

If relationships between excavators and other forest-associated biota are universally
positive in nature, and common not only at stand- and landscape-scales but hold across large
geographic regions, then excavators could form a unique group of indicators that may be
effective across multiple locations and spatial scales [22]. To find species that are effective as
both indicator and management surrogates across multiple regions would be especially useful in
the largely understudied tropics where comprehensive biodiversity assessments are costly and
logistically challenging [23]. The first signs are promising as the overwhelming majority of tree-
cavity excavators are forest or woodland birds and general patterns of woodpecker richness
correlate positively with the amount of forest cover at the global scale [24]. Moreover, many
excavators are regarded as habitat specialists and tend to have highly specific habitat
requirements, ~~requirements~~ which are usually only met in forests with high degrees of
naturalness and low levels of disturbances (e.g., logging). In fact, restoration efforts have
effectively made use of woodpeckers to indicate that restoration has successfully reached a
certain level of naturalness [25, 26]. This group of birds often utilizes large, live or decaying
trees, or coarse woody debris, for nesting, roosting, and feeding, and the availability of dead
wood is in particular found to be important [27-29]. This role of excavators as indicators of the
naturalness of ecosystems, whereby their presence correlates strongly with other aspects of
biodiversity, is enhanced by the fact that the majority of species are forest residents, which
arguably makes them more responsive to local changes in habitat quality (see e.g., [30]).

Excavators are not only reliable indicators of other forest birds by means of their mutual
associations with forest characteristics, but correlations between both groups of birds are also
partially determined by species interactions. Taken together, these ‘nest-webs’ [31] of interacting
species are structured by the availability and acquisition of tree-cavities formed by two major
processes: ‘~~natural~~ formation’, in which fungi decompose wood over an extended period, or
animal excavation [32, 33]. Since most excavators—at least the majority of woodpeckers—
excavate a new nest cavity each year, this mode of cavity creation steadily provides a large
number of potential sheltered roosting and nesting sites for secondary-users: both for vertebrates

[33] and invertebrates [34, 35]. As a result, cavity formation by excavation is an important
source of nest holes for species in many regions of the world, especially North America [33].
However, decay-formed cavities are likely to be more important nest sites in other regions,
especially the tropics, and it is very likely that the relative importance of both processes of cavity
formation—and thus the strength of the relationship between excavator richness and secondary
cavity-nester richness—varies globally [33]. Finally, it is worth noting that excavators play
additional roles (beyond the direct provision of cavities) that may facilitate other species. For
example, as excavators perforate the bark of trees they may expose insects or sap for other
species to forage on [36, 37]. In addition, excavators can aid in the dispersal of fungi, which in
turn enhances processes of wood softening and the formation of decay-formed cavities [38].

**Local studies of cavity-nesting birds and their interactions are largely biased towards**
**temperate zones, and only a few authors have explored global variation in nest-web**
**characteristics or addressed how such variation should be taken into account when considering**
**cavity-nester communities in forest conservation or management (e.g., [33, 39]).** Nevertheless,
we might predict that these birds could be reliable indicators of avian diversity and richness not
only at stand or landscape-levels [9, 19], but also across large regions at a global scale, if not for
the cavities that excavators supply but for the associations of excavators with forest elements that
are indicative of heterogeneous forests with a high degree of naturalness. Therefore, the aim of
the present study is to investigate whether there are consistent patterns in the relationships
between avian tree-cavity excavators (hereafter: excavators) and species richness of non-
excavating tree-cavity nesters (hereafter: secondary cavity-nesters), forest specialist species, and
forest birds in general, across the globe. If previously observed relationships between cavity-
excavators and other forest birds scale up from local forest stands [17] or ecosystems [19] to
zoogeographic or global levels, then this strengthens the potential of tree cavity-excavators as
both indicators and management surrogates.

Given the important role of excavators in forest ecosystems, we predicted that secondary
cavity-nester, forest specialist, and forest bird richness generally will increase with the number of
excavators present in the ecosystem, regardless of the ecosystem or location. However, we also
predicted that the nature (e.g., the strength and slope of correlations) of the relationships between
excavators and richness of other forest birds will vary across zoogeographic regions, reflecting
differences in forest characteristics (e.g., deciduous vs. coniferous forests), forest management
practices (e.g., forests being more or less intensively managed), and bird communities (e.g., the
relative richness of different groups of forest-associated birds), as previously discussed in e.g.,
[33]. In particular, we predicted that correlations between excavators and other birds would be
particularly strong, but with relatively shallow regression slopes, in regions with relatively high
intensities of forest management (vs. unmanaged natural forests) and low overall diversity. In
these predominantly temperate and boreal regions, such as the Nearctic and Palearctic,
relationships between excavators and other forest birds that are driven by mutual associations
with particular forest elements would be enhanced by direct interactions through the provision of
nest cavities, as a lack of available decay-formed cavities may increase the dependence of
secondary-cavity nesters on excavated cavities. We predicted relatively shallow regression
slopes in these regions (Nearctic, Palearctic), as there is a relatively low ratio between the
numbers of excavators and secondary-cavity nesters [11]. For regions that largely span the
tropics, such as the Neotropics, we also predicted positive correlations between excavators and
other forest birds, but as these relationships would be driven mainly by mutual associations with
particular forest elements and less by direct interactions through the provision of nest cavities—

decay-formed cavities being more prevalent here—we predicted these correlations to be weaker.
However, as these regions are more diverse, we also predicted that there would be relatively
higher numbers of forest birds per excavator, resulting in steeper regression slopes.
189 **Material and Methods**

We used published global distribution maps of tree-cavity nesting birds [11] and forest birds [40]
to extract richness estimates per 10×10 km grid cell. Maps for both tree-cavity nesters and forest
birds were based on the same species range maps provided by BirdLife International and
NatureServe [41]. We selected and classified focal bird species following [11, 40] as 1)
excavators (species that are known to often or always excavate their own nesting cavities),
secondary cavity-nesters: birds which are known to nest in tree cavity but which never or rarely
excavate their own cavity, 3) forest specialists: birds that exclusively use forest habitat, and 4)
forest birds: birds that use forest habitat but also at least one other type of habitat. We note that
some species were counted in more than one  gory (e.g., a species that is a ‘forest specialist’
is per definition also counted as a ‘forest bird’). However, excavators were never counted in any
other category.

We used only range polygons where a species’ presence was classified as ‘Extant’ or
‘Probably extant’, and assigned ‘presence’ to all grid cells that overlapped with range polygons.
Next, we created generalized least squares (gls) models to assess the linear relationships between
excavators and other categories of forest-associated birds. To take biogeographical differences
into account, we proceeded to separate trends across the globe, and created gls models for the six
zoogeographic regions separately: Nearctic, Palearctic, Neotropical, Afrotropical, Oriental, and
Australasia. We did not restrict our analyses to forested regions, wanting to avoid making
subjective decisions on the classification of a grid cell to a specific vegetation/habitat type. Yet,
to make sure we did look for possible relationships between these forest-associated species in
regions they actually occur, we removed cells with zero excavators/~~woodpeckers~~ and cells where
the numbers of secondary cavity-nesters, forest specialists, or all forest birds were lower than
that of excavators/~~woodpeckers~~ from further analyses.

Our data were spatially structured (i.e., high and low richness of birds in different
categories (excavators etc.) was spatially clustered) across and within zoogeographic regions, as
demonstrated by the global distribution of the relative number of cavity excavators to non-
excavating cavity-nesting birds (expressed as the ratio of excavator / non-excavating cavity-
nester; Fig. 1). Thus, we followed similar protocols to adjust for spatial autocorrelation as per
Storch et al. [42]. This implied the use of gls models with excavator richness as a predictor
variable, the richness of one of the categories of forest-associated birds as a response variable,
and a variable exponential spatial covariance structure, as these models had the best fit in
preliminary tests, which reduced—rather than eliminated—the influence of spatial
autocorrelation on our models. In these tests, exponential gls models consistently had the lowest
Akaike’s Information Criterion (AIC) value compared to a set of models, including an ordinary
least squares model with no spatial structure, and gls models with Gaussian, linear, rational
quadratics and spherical spatial covariance structures.

Even though gls models work well with large datasets, we found that gls models fitted on
all data failed to converge (using the R package “nlme” [43]). To circumvent this computational
problem, we first grouped grid cells by zoogeographic region. We thereafter ran 1000 models on
bootstrap permutations of 1000 randomly selected grid cells (see [24, 44]). Of these 1000 model
iterations, we subsequently calculated the mean (\pm SD), minimum and maximum coefficients of

the regression slope. As gls models do not provide a measure of the strength of relationships, we proceeded with a different measure of the strength of each correlation: fitting each exponential gls model to a different, non-overlapping, test data set of 1000 randomly selected grid cells. We then proceeded to calculate the Spearman rho coefficient of the correlation between predicted and observed data, for all 1000 models per zoogeographic region. Finally, we calculated the mean Spearman rho of all significant models per zoogeographic region. We repeated all of these analyses for woodpeckers as a subset of the excavators, with the notable exception of Australasia where woodpeckers are absent. All data, R scripts, and initial results needed to reproduce our analyses presented here are deposited online (Appendix S1).

Results

Using gls modelling to assess the global patterns for the relationships between excavator richness and that of secondary cavity-nesters, forest specialists, and all forest birds, we found positive relationships between all pairs of comparisons. However, these relationships differed in relative strength and regression slope depending on the group of birds and zoogeographic region analysed (Fig. 2, Table 1; see Table S1 for an overview of the number of species per group per region).

In terms of variation across zoogeographic regions, the relationships between excavators and all other groups of forest-associated birds were typified by particularly steep regression slopes in the Neotropics, where the presence of one additional excavator species in a grid cell implied an increase of more than two additional secondary cavity-nesters, ten forest bird species, or four forest specialists. In contrast, regression slopes were less steep in predominantly temperate and boreal regions such as the Nearctic and Palearctic, where an increase of one excavator species signified an increase of approximately one secondary cavity-nester species, four to five forest birds, or one forest specialist. More importantly, and similar to variation in the slopes of the regressions, we found differences in the strength (as represented by Spearman's rho) of these relationships, with again stronger relationships in the predominantly tropical Afrotropical, Oriental, and Neotropical regions than in the comparably more temperate/boreal Nearctic and Palearctic regions (Table 1).

Although the relationships between excavators and the other three groups of cavity-nesting and forest-associated birds were generally similar in nature (positive), there were some differences in the relative strength of these relationships across zoogeographic regions. For example, excavator richness was a particularly strong predictor of secondary cavity-nester richness and of all forest birds in the Oriental region ($\rho = 0.88$ and 0.86 , respectively), and slightly less so for forest specialists ($\rho = 0.77$), whereas excavator richness in the Afrotropics showed a strong relationship with all forest birds ($\rho = 0.94$) and with forest specialists ($\rho = 0.88$), but a somewhat weaker association with secondary cavity-nesters ($\rho = 0.79$).

We repeated all analyses for woodpeckers as a subset of excavators and found nearly identical results, with again significant positive relationships between woodpeckers and other groups of cavity-nesting or forest-associated birds in all regions except Australasia, and with steeper regression slopes and stronger relationships in the predominantly tropical Neotropical, Oriental and Afrotropical regions as compared to more shallow slopes and weaker associations in the Nearctic and Palearctic (Table 2). The relative strength of the relationships between woodpeckers and the three other groups of cavity-nesting and forest-associated birds again depended on the zoogeographic region (in similar ways as for entire group of excavators) such that woodpeckers were strong predictors of the richness of other birds as excavators in general,

with the notable exception of Australasia. Nevertheless, despite the complete absence of
woodpeckers from this region, we did find a $\sqrt{r^2}$ (rho = 0.69) but often significant ($p < 0.05$ in
97 percent of model iterations) positive relationship between richness of excavators and forest
specialists.

282 Discussion

We found strong positive relationships between excavators/~~woodpeckers~~ and secondary cavity-
nesters, forest birds and forest specialists at global and zoogeographic scales, although these
relationships differed in both strength and characteristics (regression slopes) across
zoogeographic regions. Our results imply that excavators and woodpeckers (a predominant
subset of excavators) hold potential as indicators and management surrogates across most of the
world, but also that we need to take differences among zoogeographic regions (e.g., in the
composition of local and regional nest-webs with regards to the relative numbers of excavators
and secondary cavity-nesters as seen in Fig. 1) into account if we are to adopt these as surrogates
for conservation or management.

Many excavators, ~~woodpeckers included~~, have strong preferences for specific forest
elements, such as dead and decaying trees or particularly large trees (see e.g., [28]). In turn, these
elements, and the processes that drive their presence and abundance, are of great importance to
habitat formation and create additional niche space to support many groups of organisms [46].
Therefore, we can expect that forests that meet the habitat requirements of excavators (e.g.,
certain levels of availability of decaying and dead wood) provide high-quality habitat for many
other wildlife species. ~~This may explain the higher richness of forest birds in areas occupied by~~
~~high numbers of excavators/woodpeckers~~. For non-excavating species that utilize tree-cavities
for their nests, this relationship may be ~~enhanced~~ by the fact that excavators/woodpeckers create
or facilitate potential nest substrates. In other words, the strong habitat preferences of many
excavators/~~woodpeckers~~, as well as the facilitatory role excavators play in providing nest
substrates, make these birds potentially effective indicators of many forest-associated species
and, by extension, of high-quality forests (i.e., natural, old-growth; [17, 18]).

The steep slopes we found for correlations between excavators/~~woodpeckers~~ and other
forest birds in the predominantly tropical regions (Neotropics, Oriental, Afrotropical) contrast to
more shallow slopes in the temperate/boreal regions (Palearctic, Nearctic). Macro-scale
processes (e.g., as outlined by Gaston [47]), such as speciation events, are likely to account for
some of these differences among zoogeographic regions. First, the overall richness of birds [48],
including cavity-nesting birds [11], and also of trees as potential substrates for cavities [49],
differs both among and within zoogeographic regions, likely inducing differences in tree-cavity-
nesting assemblages [33]. However, variation in species composition and associated variation in
functional richness (i.e., the variety of functional traits varies across spatial and environmental
gradients, as has been shown, for example, for dietary guilds [44]), is likely to contribute most to
spatial variation in nest-web assemblages across global scales. Specifically, with regards to nest-
webs, species with the ability to excavate cavities are not randomly distributed across the world.
For example, the Australasian region lacks the presence of Piciformes, the Order to which nearly
75% of excavators belong [11], whereas speciation processes have led to a relatively high
diversity of Piciformes in Southeast Asia [50, 51].

The observed spatial variation in the relationships between excavators and other forest
birds within zoogeographic regions may stem mainly from local differences in habitat and
vegetation characteristics (e.g., tree species composition and climatic variables [52]) as well as

spatial variation in tree cavity availability and their formation agents [53], rather than from
speciation processes. Such differences would, for example, explain how in the Neotropics,
cavity-nesting assemblages studied in the temperate mountain forests of Chile [54] had a 1:6
ratio of excavators (4 species) to secondary cavity-nesters (25 species) whereas a subtropical
Atlantic moist forest [32] supported relatively higher numbers of excavators with a 1:3 ratio of
excavators (9 species) to secondary cavity-nesters (25 species). An important factor to consider
here is the potential for regional variation in cavity substrate availability (e.g., large trees) and
the supply of decay-formed cavities due to differences in the onset and rate of decay processes as
well as forest management (e.g., [55-57]). For example, retention of large trees can, in many
forests, provide nesting and roosting opportunities for additional species as both opportunities for
excavation and processes of decay become more prevalent.

In addition, we can hypothesize that at least some variation at regional scales stems from
differences in the propensities for re-use of excavated cavities by secondary cavity-nesters. For
example, there are signs that non-excavating cavity-nesters may experience higher predation risk
in excavated versus decay-formed cavities, which may induce a preference for decay-formed
cavities among secondary users if decay-formed cavities are in sufficient supply (e.g., a pattern
found in some of the few unmanaged temperate forests in the European parts of the Palearctic;
[58, 59]). In turn, the availability of decay-formed cavities may correspond to the
aforementioned historical or current forest management practices as well as spatial patterns in
climatic factors. For example, proportions of decay-formed cavities used are particularly high in
parts of the world with high amounts of precipitation and more active regimes of fungal growth
(e.g., most of the wet tropics), with both being lower in anthropogenically-disturbed landscapes
than in primary forests [53]. In turn, high availability of decay-formed cavities might allow for a
relatively high richness of secondary cavity-nesters, with fewer species dependent on excavators
for the supply of nest-cavities. For example, even woodpeckers, commonly considered primary
excavators, occasionally use decay-formed cavities in Neotropical temperate rainforests where
tree decay is the key driver of nest-web structure [54].

Finally, we find that relatively higher numbers of secondary cavity-nesters, or cavity-
nesters in general, may occur in regions where many species are habitat generalists that utilize
other nest substrates than tree-cavities (e.g., 40% of cavity nesters in Neotropical temperate
forests are facultative users of tree cavities for nesting [54]), as compared to regions where most
cavity nesters are obligate tree-cavity users (e.g., over 85% of cavity nesters in temperate
Nearctic forests [59]). Finally, it is worth noting that cavity availability and use are also likely to
be influenced by cavity persistence (longevity) characteristic that can have considerable spatial
variation due to factors such as land-use patterns (e.g., logging), climatic factors (e.g., rainfall),
and cavity characteristics (e.g., whether they are located in live or dead trees [60-62]).

We detected relative differences in strengths among the relationships between
excavators/woodpeckers and secondary cavity-nesters, forest birds, and forest specialists across
zoogeographic regions. These differences were most likely to stem from the same variation
among species and ecosystem characteristics discussed above (e.g., the distribution and relative
importance of cavity-forming agents), and indicate that richness of excavators/woodpeckers may
be a better indicator of richness of other cavity and forest-associated species in one region than in
another. In addition, the relative strength of relationships between excavators and other forest
birds might reflect whether excavators are indicative of the richness of other species due to
shared associations with particular forest elements (indirect association), due to their direct
facilitation of nest cavities (direct facilitation), or whether both factors play a role. For example,

we found that relationships between excavator and forest bird or forest specialist richness in the Afrotropics are relatively strong, whereas the relationship between excavators and secondary cavity-nesters is comparatively weaker. This might indicate that, in the Afrotropics, excavators are especially strong predictors of other groups of forest-associated birds through their mutual habitat requirements, rather than through their direct facilitation of tree-cavities. In contrast, excavators are relatively stronger predictors of secondary cavity-nesters in the Oriental region, which might lead to hypotheses that excavators play a more direct, nest-web structuring role through the provision of cavities in the Oriental region.

Some secondary cavity-nesters might use cavities created by multiple excavator species, whereas others predominantly use cavities formed by processes of degradation, damage or insect activity [31, 32, 39]. In addition, certain excavators might be especially abundant or provide cavities that are very commonly reused by several species, whereas others are scarce and leave only a few short-lived cavities that would not be generally available for use by secondary cavity-nesters. Given such complexities among nest-web interactions, it is important to emphasize that we present general global patterns in the composition of tree-cavity-using assemblages (i.e., communities of linked species that often interact within networks). We lack evidence for direct causal links between the presence of one species group with another, as well as direct evidence of species interactions at the zoogeographic regional scale.

Around the world, we found relationships of a universal nature (strong and positive) between excavator and forest bird richness. ~~such, we deem excavators, of which woodpeckers are an important subset,~~ to be useful indicator species for forest bird diversity at multiple spatial scales. They form a small and fairly easy to identify (even by citizen scientists) subset of a larger group of species (forest birds) often studied to understand the effects of anthropogenic disturbances. Studies of excavators ~~(or woodpeckers specifically)~~, especially in high biodiversity tropical regions, might provide quick measures of forest quality and biodiversity, which may in turn guide forest management decisions (e.g., improving the retention of large trees). A comprehensive study of all forest birds in some regions like the Amazon or Congo basin would require considerable local expertise and training, which is often difficult to achieve due to social and monetary limitations. Thus, we conclude, excavators have excellent potential as study subjects for conservation monitoring and planning initiatives, in comparative research, and to guide the establishment of region-wide forest conservation strategies, especially in the largely understudied tropical regions of the world.

Acknowledgments

We would like to thank Lisa Manne, and Kristina Cockle and two anonymous reviewers for their critical remarks on drafts of this article.

References

- Allan JR, Venter O, Maxwell S, Bertzky B, Jones K, Shi Y, Watson JE. 2017 Recent increases in human pressure and forest loss threaten many Natural World Heritage Sites. *Biological conservation*. **206**, 47-55.
- Hansen MC, Stehman SV, Potapov PV. 2010 Quantification of global gross forest cover loss. *Proceedings of the National Academy of Sciences*. **107**, 8650-8655.
- Barlow J, Lennox GD, Ferreira J, Berenguer E, Lees AC, Mac Nally R, Thomson JR, de Barros Ferraz SF, Louzada J, Oliveira VHF. 2016 Anthropogenic disturbance in tropical forests can double biodiversity loss from deforestation. *Nature*. **535**, 144.

Gaston KJ, Blackburn TM, Goldewijk KK. 2003 Habitat conversion and global avian biodiversity loss.
*Proceedings of the Royal Society of London. Series B: Biological Sciences.* **270**, 1293-1300.
- Betts MG, Wolf C, Pfeifer M, Banks-Leite C, Arroyo-Rodríguez V, Ribeiro DB, Barlow J, Eigenbrod F,
Faria D, Fletcher RJ. 2019 Extinction filters mediate the global effects of habitat fragmentation on
animals. *Science.* **366**, 1236-1239.
- Carignan V, Villard M-A. 2002 Selecting indicator species to monitor ecological integrity: a review.
*Environmental monitoring and assessment.* **78**, 45-61.
- Canterbury GE, Martin TE, Petit DR, Petit LJ, Bradford DF. 2000 Bird communities and habitat as
ecological indicators of forest condition in regional monitoring. *Conservation Biology.* **14**, 544-558.
- Venier LA, Pearce JL. 2004 Birds as indicators of sustainable forest management. *The Forestry*
*Chronicle.* **80**, 61-66.
- Martin K, Ibarra JT, Drever M. 2015 Avian surrogates in terrestrial ecosystems: theory and practice. In
*Indicators and surrogates of biodiversity and environmental change* (eds D Lindenmayer, P Barton, J
Pierson), pp 33-44. Melbourne & London: CSIRO Publishing & CRC Press.
- Ibarra JT, Martin M, Cockle KL, Martin K. 2017 Maintaining ecosystem resilience: functional
responses of tree cavity nesters to logging in temperate forests of the Americas. *Scientific reports.* **7**,
4467.
- van der Hoek Y, Gaona GV, Martin K. 2017 The diversity, distribution and conservation status of the
tree-cavity-nesting birds of the world. *Diversity and Distributions.* **23**, 1120-1131.
- McClelland BR, McClelland PT. 1999 Pileated woodpecker nest and roost trees in Montana: links with
old-growth and forest" health". *Wildlife Society Bulletin.* 846-857.
- Stachura-Skierczyńska K, Kosiński Z. 2016 Do factors describing forest naturalness predict the
occurrence and abundance of middle spotted woodpecker in different forest landscapes? *Ecological*
*Indicators.* **60**, 832-844.
- Marchetti M. 2005 *Monitoring and indicators of forest biodiversity in Europe: from ideas to*
*operationality.* Joensuu, Finland: European Forest Institute.
- Roberge J-M, Angelstam P. 2006 Indicator species among resident forest birds—a cross-regional
evaluation in northern Europe. *Biological Conservation.* **130**, 134-147.
- Virkkala R. 2006 Why study woodpeckers? The significance of woodpeckers in forest ecosystems.
*Annales Zoologici Fennici.* 82-85.
- Drever MC, Aitken KE, Norris AR, Martin K. 2008 Woodpeckers as reliable indicators of bird richness,
forest health and harvest. *Biological conservation.* **141**, 624-634.
- Drever MC, Martin K. 2010 Response of woodpeckers to changes in forest health and harvest:
Implications for conservation of avian biodiversity. *Forest Ecology and Management.* **259**, 958-966.
- Mikusiński G, Gromadzki M, Chylarecki P. 2001 Woodpeckers as indicators of forest bird diversity.
*Conservation Biology.* **15**, 208-217.
- Angelstam PK, Bütler R, Lazdinis M, Mikusiński G, Roberge J-M. 2003. Habitat thresholds for focal
species at multiple scales and forest biodiversity conservation—dead wood as an example. *Annales*
*Zoologici Fennici.* 473-482.
- Nilsson SG, Hedin J, Niklasson M. 2001 Biodiversity and its assessment in boreal and nemoral forests.
*Scandinavian Journal of Forest Research.* **16**, 10-26.
- Hess GR, Bartel RA, Leidner AK, Rosenfeld KM, Rubino MJ, Snider SB, Ricketts TH. 2006 Effectiveness
of biodiversity indicators varies with extent, grain, and region. *Biological Conservation.* **132**, 448-457.
- Collen B, Ram M, Zamin T, McRae L. 2008 The tropical biodiversity data gap: addressing disparity in
global monitoring. *Tropical Conservation Science.* **1**, 75-88.
- Ilsøe SK, Kissling WD, Fjeldsø J, Sandel B, Svenning JC. 2017 Global variation in woodpecker species
richness shaped by tree availability. *Journal of Biogeography.* **44**, 1824-1835.

Bell D, Hjältén J, Nilsson C, Jørgensen D, Johansson T. 2015 Forest restoration to attract a putative
umbrella species, the white-backed woodpecker, benefited saproxylic beetles. *Ecosphere*. **6**, 1-14.
Gatica-Saavedra P, Echeverría C, Nelson CR. 2017 Ecological indicators for assessing ecological
success of forest restoration: a world review. *Restoration ecology*. **25**, 850-857.
Hoyt JS, Hannon SJ. 2002 Habitat associations of black-backed and three-toed woodpeckers in the
boreal forest of Alberta. *Canadian Journal of Forest Research*. **32**, 1881-1888.
Tikkanen O-P, Martikainen P, Hyvärinen E, Junninen K, Kouki J. 2006. Red-listed boreal forest species
of Finland: associations with forest structure, tree species, and decaying wood. *Annales Zoologici*
*Fennici*. 373-383.
Bütler R, Angelstam P, Ekelund P, Schlaepfer R. 2004 Dead wood threshold values for the three-toed
woodpecker presence in boreal and sub-Alpine forest. *Biological Conservation*. **119**, 305-318.
McLaren MA, Thompson ID, Baker JA. 1998 Selection of vertebrate wildlife indicators for monitoring
sustainable forest management in Ontario. *The Forestry Chronicle*. **74**, 241-248.
Martin K, Eadie JM. 1999 Nest webs: A community-wide approach to the management and
conservation of cavity-nesting forest birds. *Forest Ecology and Management*. **115**, 243-257.
Cockle KL, Martin K, Robledo G. 2012 Linking fungi, trees, and hole-using birds in a Neotropical tree-
cavity network: Pathways of cavity production and implications for conservation. *Forest Ecology and*
*Management*. **264**, 210-219.
Cockle KL, Martin K, Wesolowski T. 2011 Woodpeckers, decay, and the future of cavity-nesting
vertebrate communities worldwide. *Frontiers in Ecology and the Environment*. **9**, 377-382.
Tylianakis JM, Klein AM, Lozada T, Tschardt T. 2006 Spatial scale of observation affects α , β and γ
diversity of cavity-nesting bees and wasps across a tropical land-use gradient. *Journal of Biogeography*.
**33**, 1295-1304.
Powell S, Costa AN, Lopes CT, Vasconcelos HL. 2011 Canopy connectivity and the availability of
diverse nesting resources affect species coexistence in arboreal ants. *Journal of Animal Ecology*. **80**, 352-
360.
Bull EL, Jackson JA., 1995 Pileated Woodpecker (*Dryocopus pileatus*). In: *The birds of North America*
(eds A Poole, F Gill). Washington DC: The Academy of Natural Sciences & The American Ornithologists'
Union.
Montellano MGN, Blendinger PG, Macchi L. 2013 Sap consumption by the White-fronted
Woodpecker and its role in avian assemblage structure in dry forests. *The Condor*. **115**, 93-101.
Jusino MA, Lindner DL, Banik MT, Rose KR, Walters JR. 2016 Experimental evidence of a symbiosis
between red-cockaded woodpeckers and fungi. *Proceedings of the Royal Society B: Biological Sciences*.
**283**, 20160106.
Ruggera RA, Schaaf AA, Vivanco CG, Politi N, Rivera LO. 2016 Exploring nest webs in more detail to
improve forest management. *Forest Ecology and Management*. **372**, 93-100.
Betts MG, Wolf C, Ripple WJ, Phalan B, Millers KA, Duarte A, Butchart SH, Levi T. 2017 Global forest
loss disproportionately erodes biodiversity in intact landscapes. *Nature*. **547**, 441-444.
BirdLife International and NatureServe 2015 *Bird species distribution maps of the world*. Cambridge
and Arlington: BirdLife International and NatureServe.
Storch D, Davies RG, Zajíček S, Orme CDL, Olson V, Thomas GH, Ding TS, Rasmussen PC, Ridgely RS,
Bennett PM. 2006 Energy, range dynamics and global species richness patterns: reconciling mid-domain
effects and environmental determinants of avian diversity. *Ecology Letters*. **9**, 1308-1320.
Pinheiro J, Bates D, DebRoy S, Sarkar D, Heisterkamp S, Van Willigen B. 2017 Package 'nlme'. *Linear*
*and Nonlinear Mixed Effects Models, version 3-1*. Available from [https://cran.r-](https://cran.r-project.org/web/packages/nlme/nlme.pdf)
[project.org/web/packages/nlme/nlme.pdf](https://cran.r-project.org/web/packages/nlme/nlme.pdf) (accessed January 15, 2018).

Kissling WD, Sekercioglu CH, Jetz W. 2012 Bird dietary guild richness across latitudes, environments
and biogeographic regions. *Global Ecology and Biogeography*. **21**, 328-340.
- Hunter M, Westgate M, Barton P, Calhoun A, Pierson J, Tulloch A, Beger M, Branquinho C, Caro T,
Gross J. 2016 Two roles for ecological surrogacy: Indicator surrogates and management surrogates.
*Ecological Indicators*. **63**, 121-125.
- Stokland JN, Siitonen J, Jonsson BG. 2012 *Biodiversity in dead wood*. Cambridge, UK: Cambridge
University Press.
- Gaston KJ. 2000 Global patterns in biodiversity. *Nature*. **405**, 220.
- Blackburn TM, Gaston KJ. 1996 Spatial patterns in the species richness of birds in the New World.
*Ecography*. **19**, 369-376.
- Currie DJ, Paquin V. 1987 Large-scale biogeographical patterns of species richness of trees. *Nature*.
**329**, 326-327.
- Benz BW, Robbins MB, Peterson AT. 2006 Evolutionary history of woodpeckers and allies (Aves:
Picidae): placing key taxa on the phylogenetic tree. *Molecular phylogenetics and evolution*. **40**, 389-399.
- Mikusiński G. 2006 Woodpeckers: distribution, conservation, and research in a global perspective.
*Annales Zoologici Fennici*. 86-95.
- Huston MA. 1999 Local processes and regional patterns: appropriate scales for understanding
variation in the diversity of plants and animals. *Oikos*. **86**, 393-401.
- Remm J, Löhmus A. 2011 Tree cavities in forests—the broad distribution pattern of a keystone
structure for biodiversity. *Forest Ecology and Management*. **262**, 579-585.
- Altamirano TA, Ibarra JT, Martin K, Bonacic C. 2017 The conservation value of tree decay processes as
a key driver structuring tree cavity nest webs in South American temperate rainforests. *Biodiversity and
Conservation*. **26**, 2453-2472.
- Andersson J, Domingo Gómez E, Michon S, Roberge J-M. 2018 Tree cavity densities and
characteristics in managed and unmanaged Swedish boreal forest. *Scandinavian journal of forest
research*. **33**, 233-244.
- Goodburn JM, Lorimer CG. 1998 Cavity trees and coarse woody debris in old-growth and managed
northern hardwood forests in Wisconsin and Michigan. *Canadian Journal of Forest Research*. **28**, 427-
438.
- Gutzat F, Dormann CF. 2018 Decaying trees improve nesting opportunities for cavity-nesting birds in
temperate and boreal forests: A meta-analysis and implications for retention forestry. *Ecology and
evolution*. **8**, 8616-8626.
- Wesolowski T, Tomialojc L. 2005 Nest sites, nest depredation, and productivity of avian broods in a
primeval temperate forest: do the generalisations hold? *Journal of Avian Biology*. **36**, 361-367.
- Wesolowski T, Martin K. 2018 Tree Holes and Hole-Nesting Birds in European and North American
Forests. In *Ecology and Conservation of Forest Birds* (eds G Mikusinski, J-M Roberge, RJ Fuller), pp 79-
134. Cambridge, UK: Cambridge University Press.
- Edworthy AB, Wiebe KL, Martin K. 2012 Survival analysis of a critical resource for cavity-nesting
communities: patterns of tree cavity longevity. *Ecological Applications*. **22**, 1733-1742.
- Cockle KL, Martin K, Bodrati A. 2017 Persistence and loss of tree cavities used by birds in the
subtropical Atlantic Forest. *Forest ecology and management*. **384**, 200-207.
- Wesolowski T. 2011 “Lifespan” of woodpecker-made holes in a primeval temperate forest: A thirty
550 year study. *Forest ecology and management*. **262**, 1846-1852.

Table 1. Results of generalized least square models of possible correlations between (1) excavator and secondary cavity-nester richness, (2) excavator and forest bird richness, (3) excavator and forest specialist richness. Each result reflects the mean outcome of 1000 models build with 1000 randomly selected grid cell data points that include an exponential spatial covariance structure. The mean Spearman rho was calculated over the fit of each of 1000 models to a test data set.

Zoogeographic region	Correlation	Mean Slope (Min. – Max.; SD)	% of models with $P < 0.05$	Mean Spearman rho
Nearctic	1	0.8 (0.4 - 1.3; 0.1)	60	0.73
	2	4.3 (3.3 - 5.9; 0.4)	63	0.84
	3	1.4 (1.2 - 1.6; 0.1)	71	0.71
Palearctic	1	1.1 (0.7 - 1.4; 0.1)	84	0.78
	2	4.9 (4.0 - 6.7; 0.4)	73	0.85
	3	1.1 (0.8 - 1.7; 0.1)	83	0.79
Oriental	1	0.7 (0.6 - 0.9; 0.0)	100	0.88
	2	5.7 (4.9 - 6.6; 0.3)	100	0.86
	3	1.6 (1.3 - 1.9; 0.1)	100	0.77
Neotropical	1	2.4 (1.9 - 3.0; 0.2)	83	0.95
	2	10.8 (9.1 - 14.0; 0.7)	72	0.97
	3	4.2 (3.4 - 5.2; 0.2)	72	0.96
Afrotropical	1	1.1 (0.7 - 1.9; 0.2)	76	0.79
	2	6.8 (5.0 - 9.8; 0.9)	73	0.94
	3	0.9 (0.6 - 1.5; 0.1)	79	0.88
Australasian	1	1.8 (1.4 - 2.2; 0.1)	88	-0.14
	2	7.5 (2.3 - 11.2; 1.3)	100	0.31
	3	1.0 (-0.1 - 2.0; 0.3)	97	0.69

Table 2. Results of generalized least square models of possible correlations between (1) woodpecker and secondary cavity-nester richness, (2) woodpecker and forest bird richness, and (3) woodpecker and forest specialist richness. Each result reflects the mean outcome of 1000 models build with 1000 randomly selected grid cell data points that include an exponential spatial covariance structure. The mean Spearman rho was calculated over the fit of each of 1000 models to a test data set.

Zoogeographic region	Correlation	Mean Slope (Min. – Max.; SD)	% of models with $P < 0.05$	Mean Spearman rho
Nearctic	1	1.0 (0.5 - 1.6; 0.2)	61	0.70
	2	5.3 (3.7 - 8.0; 0.7)	67	0.79
	3	1.7 (1.1 - 2.2; 0.1)	70	0.63
Palearctic	1	1.2 (0.9 - 2.3; 0.1)	81	0.79
	2	5.6 (4.3 - 8.1; 0.6)	73	0.83
	3	1.3 (0.9 - 2.0; 0.2)	82	0.73
Oriental	1	1.0 (0.8 - 1.3; 0.1)	100	0.88
	2	7.9 (6.6 - 9.8; 0.5)	100	0.88
	3	2.2 (1.7 - 2.6; 0.1)	100	0.80
Neotropical	1	3.3 (2.4 - 4.9; 0.4)	68	0.94
	2	14.3 (10.8 - 20.5; 1.6)	67	0.93
	3	5.2 (3.9 - 6.6; 0.5)	73	0.88
Afrotropical	1	1.6 (0.9 - 3.5; 0.4)	72	0.74
	2	9.9 (6.4 - 19.6; 1.9)	69	0.86
	3	1.3 (0.7 - 3.2; 0.3)	80	0.81
Australasian	1	1.9 (-0.1 - 4.4; 0.6)	81	-0.33
	2	8.6 (1.2 - 21.8; 2.4)	75	-0.30
	3	3.9 (-0.2 - 9.0; 1.2)	96	-0.08

**600 Figure legends**

601

**602 Figure 1. Global map of the relative richness of tree cavity excavators versus secondary**
**603 cavity-nesting birds (expressed as the ratio excavator / secondary cavity-nester) in 10x10**
**604 km grid cells.** Low values (light colors) indicate relatively low numbers of excavators as
compared to secondary cavity-nesters (or, inversely, relatively high numbers of secondary
~~cavity-nesters~~) while high values (dark colors) indicate a relatively higher proportion of
excavators to secondary cavity-nesters.
607

608

**609 Figure 2. Scatter plots of the correlation between the richness (number of species) of**
**610 excavators and all forest birds (upper row), all forest specialists (middle row), and all non-**
**611 excavating cavity-nesters (bottom row) in six zoogeographic regions.** Points represent species
richness values in 10x10 km grid cells, and the solid line represents the fit of a linear regression
model, created using all grid cell values.

Figure 1.

Figure 2.

Figure 1. Global map of the relative richness of tree cavity excavators versus secondary cavity-nesting birds (expressed as the ratio excavator / secondary cavity-nester) in 10x10 km grid cells. Low values (light colors) indicate relatively low numbers of excavators as compared to secondary cavity-nesters (or, inversely, relatively high numbers of secondary cavity-nesters) while high values (dark colors) indicate a relatively higher proportion of excavators to secondary cavity-nesters.

Figure 2. Scatter plots of the correlation between the richness (number of species) of excavators and all forest birds (upper row), all forest specialists (middle row), and all non-excavating cavity-nesters (bottom row) in six zoogeographic regions. Points represent species richness values in 10x10 km grid cells, and the solid line represents the fit of a linear regression model, created using all grid cell values.

167x119mm (300 x 300 DPI)

Authors Response letter to editor:

I am recommending a "reject/resub" decision because our "revision" turnaround time
may be too short for the authors' needs in revising, and because the reviewers have
raised extensive concerns, despite being encouraging overall about the manuscript.

They see issues with how the main problem of the study is set out, the methods, the
evaluation, and the conclusions. They also note a geographic bias that may entail
ecological biases in the relationships of excavators to their ecosystems.

If you elect to re-submit, please detail your responses to reviewers' comments and
make clear how you have altered your re-submitted manuscript. Best wishes with your
revisions.

Reviewers' Comments to Author:

Reviewer: 1

General comment:

The manuscript entitled "Global and universal relationships between tree-cavity
excavators and forest bird richness." reports on an interesting exploratory study linking
woodpeckers/other tree cavity excavators, secondary cavity users and other forest tree
species at the global and bioregional levels. The results are useful and may inspire new
studies exploring these relationships more in-depth.

**RESPONSE: We thank the reviewer for their supportive comments.**

My main criticism concerns the quality of reasoning and how and to what extent
different issues are introduced and discussed. Proposing the use of certain group of
species as indicators or other type of indicator species shall rest on solid grounds as
well as need of such indicators should be clearly put forward. I simply miss the
delineation of the conservation/management problem why we need indicators. There
are many articles published on the worldwide decline of primary/intact forests, their
characteristics and, in my opinion, you do not use the opportunity to discuss the
potential role of woodpeckers/other tree cavity excavators (with very intimate linkages to
natural forest characteristics) to support your case. Also, even if you mention their
usefulness in management, there is lacking information on the intensively managed
forests as areas that lose both important characteristics linked to old-growth forests and
forests with high level of naturalness and thereby lose most specialized forest species
often being woodpeckers.

**RESPONSE: We thank the reviewer for pointing out these ways to strengthen our**
**text. We have added arguments and citations to strengthen these points**
**throughout the Introduction (e.g., lines 96-98).**

Also use of woodpeckers/other excavators as indicators of successful forest restoration
could be mentionaed. There is large body of literature on both above topics and I
suggest that you should add paragraphs pertaining on these issues in both Introduction
and Discussion sections. To limit the length of the article, you may get rid of several
unnecessary repetitions. More suggestions below:

Detailed comments:

Line 99: Several species use smaller trees or snags too. Possibly add "often" or
"usually".

**RESPONSE: We agree with the reviewer that there are many exceptions, for**
**which we followed the suggestion to add 'often' to soften the generalization.**

Line 109: Woodpeckers (cavity excavators) are most often resident species what could
additionally strengthen your reasoning on strong linkages to forest environments and
their quality.

**RESPONSE: We thank the reviewer for this valid point, and have incorporated a**
**sentence to illustrate this in the second paragraph of our introduction.**

Lines 127-133: Please consult Wesolowski and Martin (2018) and extend the discussion
on mechanism a bit more. I am lacking a clear statement on two possible, non-
excluding each other mechanisms of expected relationships as very nicely stated way
down in Discussion (Lines 312-314).

**RESPONSE: We thank the reviewer for this valid point. We agree that this is a key**
**element of our manuscript that should be clarified much earlier in the text, and**
**have a such included lines to address this topic in the very beginning of the**
**Introduction.**

RESULTS: Could you add a summary table with numbers of species in different groups
in different bioregions? It would provide some nice background for further discussions of
results. You have those numbers already!

**RESPONSE: We understand that this might make it much easier to follow some of**
**the reasoning, plus it would benefit authors who would like to build on our**
**analyses. We have therefore followed your advice and added a supplementary**
**table (Table S1) summarizing these data.**

Lines 224-231: I do not understand how you can model the relationship between
woodpeckers and the three other groups in Australasia, where, as you state below,
there is no woodpeckers or even other strong excavators are largely lacking.

**RESPONSE: Thank you for pointing this out, this is indeed a mistake. We have**
**removed the 'woodpecker part' for Australasia and have added text to make this**
**clear in methods as well ('We repeated all of these analyses for woodpeckers as a**

**subset of the excavators, with the notable exception of Australasia where**
**woodpeckers are absent.’)**

DISCUSSION: a bit confusing use of words “excavators/woodpeckers” (Line 244) and
“excavators” (Line 275). Please check if necessary to use both of them!

**RESPONSE: We agree that this is a bit confusing and have simplified this, by**
**simply using excavator/woodpecker when an argument or explanation concerns**
**both groups equally and ‘excavator’ or ‘woodpecker’ by itself if it is particular to**
**only one of these groups (e.g., when evidence does not extend beyond**
**woodpeckers or when discussing specific results (e.g., ratios)).**

Lines 239-243: Worthy to mention that forests in some bioregions supply decay-formed
cavities in high numbers (particularly non-managed forests) and some other not (e.g.
boreal forest in Europe – see Andersson et al. (2018): Andersson J., Domingo Gomez
E., Michon S., Roberge J.-M. (2018) Tree cavity densities and characteristics in
managed and unmanaged Swedish boreal forest. Scandinavian Journal of Forest
Research, 33 (3), pp. 233-244. It would add the management angle in discussion about
the role of woodpeckers/excavators for other species.

**RESPONSE: We have added more text on this (esp. lines 304-309) including the**
**reference suggested. We absolutely agree that this is a key element of the story,**
**yet we aimed not to add too much to an already rather lengthy manuscript.**

Lines 272-273: Please refer to studies on the global species richness of woodpeckers
e.g. Mikusinski, G. 2006. Woodpeckers (Picidae) - distribution, conservation and
research in a global perspective. Annales Zoologici Fennici 43: 86–95.

**RESPONSE: We have added this reference.**

Lines 289-294: Refer also to the “primeval” Bialowieza Forest case in temperate zone
being in stark contrast to managed forests (i.e. higher predation risk and subsequently
preference for decay-formed cavities among secondary users. See: Wesolowski and
Martin (2018) and references therein.

**RESPONSE: We have added a section on this, including additional references.**

Lines 299-304: Please refer to papers by Wesolowski here: Wesolowski T.
2011. "Lifespan" of woodpecker-made holes in a primeval temperate forest: A thirty year
study. Forest Ecology and Management, 262 (9) , pp. 1846-1852.

**RESPONSE: We added this reference.**

Reviewer 2:

What is the difference between ‘global’ and ‘universal’. There is nothing in the introduction, nor
in the discussion, justifying the usage of these two words. I suggest to delete ‘universal’

**RESPONSE: We thank the reviewer for this remark, it made it clear to us that we needed**
**to clarify these terms. When we use the term ‘global’ we imply that these relationships can**
**be found in nearly every region of the world, with ‘universal’ we mean that they are of**
**similar nature (strong and positive).**

**We clarify these terms in both the Introduction and the Discussion (e.g., by adding that**
**‘Around the world, we can find relationships of a universal nature (strong and positive)**
**between excavator and forest bird richness.’).**

I see this paragraph as mostly unnecessary, with some of this information that can be fused
with the second paragraph

**RESPONSE: We understand that this start of the Introduction could be stronger, highlighting both the**
**problem (need for indicators) and the opportunity (excavators/woodpeckers could make great**
**indicators). We have reworked the introduction to accommodate these changes.**

**The reviewer mentions multiple inconsistencies in citations/references, we have now adopted the**
**‘Open Biology/Royal Society Open Science’ style and all issues should be resolved.**

I am not aware of this. I’ve seen some parrot species burrowing their own cavities in termitaria and
cliffs, or ‘arranging’ cavities already done by woodpeckers or fungi. But I have not seen/heard about
parrots excavating ‘brand new’ cavities in trees. Please, could you add a reference for that?

**RESPONSE: Although not common, parrots especially in Australasia may excavate in trunks that are in**
**advanced stages of decay, without there having been a previous cavity, see e.g., Courtney, J. 2010. An**
**Observation of Nesting Behaviour in Marshall's Fig-Parrot 'Cyclopsitta diophthalma marshalli'.**
**Australian Field Ornithology, Vol. 27, No. 4, Dec 149-152.**

39So, unless you have a better argumentation for this association at a global scale, I suggest to delete
this part.

**RESPONSE: We have, in this new version, followed up on your suggestion and removed this statement**
**that the response of woodpeckers can be indicative of that of forest birds in general.**

This is not what Cockle et al. (2011) found. According to the map in the Figure 2a, you can see that the
pattern that you mention (excavated cavities as main source for non-excavator birds) is only strong in
North America, very weak in Europe (only 1 out of 4 studies showed that pattern), and opposite in South
America and Australia, this is, non-excavator birds mainly used decay formed cavities (there is no study
in Africa). After Cockle et al. (2011), Ruggera et al. (2016) quoted “Therefore, in the PF woodpeckers do
not seem to play a key role in provisioning cavities to the nest-web, a pattern also shown for other
South American and some European forests, and contrary to the findings in North America (Cockle et al.,
2011a,b; Pereira et al., 2009; Wesolowski, 2007).” Please, be careful when citing other authors, and put
strictly what they have found.

**RESPONSE: We agree that we may have overstated the importance of excavators for the provision of**
**cavities, and have altered our language accordingly (e.g., stating that ‘cavity formation by excavation**
**is an importance source of nest holes for species in many regions of the world’). However, Cockle et**
**al. has shown that woodpeckers are an important provider of cavities in farmlands and altered forests**
**in Argentina. Thus, even in South America, woodpeckers may provide a larger role in the provisioning**
**of tree cavities than currently recognized.**

 - Please, see the previous comment on this pattern. Actually, it seems that woodpeckers play a

12 considerably smaller role than that you stated, as cavity providers in many non-disturbed

ecosystems. If my previous comment is correct, how is this prediction held?

**RESPONSE: We think this new version of our manuscript does a better job at highlighting the two**
**mechanisms by which excavators are indicators: due to their associations with particular forest**
**elements and their role in facilitating the presence of other species through the provision of nest**
**cavities. We therefore think this prediction is still valid, but we did alter the text to recognize this**
**point:**

**“Nevertheless, we might predict that these birds could be reliable indicators of avian diversity and**
**richness not only at stand or landscape-levels (8, 17), but also across large regions at a global scale,**
**not only for the cavities that excavators supply but also due to excavators’ associations with forest**
**elements that are indicative of heterogeneous forests with a high degree of naturalness.”**

I see necessary to call to this group with something that recalls that are cavity users, because both forest
specialist species and forest birds in general include also non-excavator bird species. An option can be
‘secondary cavity-nesters’

**RESPONSE: We have followed this useful suggestion and have changed non-excavator to ‘secondary**
**cavity-nester’ throughout the text.**

Here or in Methods: it’s necessary to clarify if these categories are mutually exclusive. For example,
some secondary cavity-nesters are forest specialists, and of course, are also forest birds in general.
These species are considered in the three categories or only in the ‘non-excavator’ category? How this
nested data set would eventually influence your results?

**RESPONSE: We added a sentence in the Methods to explain that some of these numbers are indeed**
**nested, with for example a forest specialist also being counted as a forest bird. We can see how it**
**could be interesting to know (for example) details on the relationship between excavators and forest**
**specialists that do not nest in cavities, which we do not provide at this point. One reason for this is**
**that sample size for many grid cells would become too low for our models to converge, particularly**
**when we would count forest specialists that do not breed in cavities as a separate group (e.g., there**
**are relatively few forest specialists in the Nearctic that are not also cavity-nesters).**

**Another reason is that we aimed to know whether excavators make useful indicators and**
**management surrogates. For this we wanted to know how indicative excavators are of larger and**
**more inclusive suites of forest-associated birds, not of very specific subgroups (e.g., forest-associated**
**species that do not breed in cavities and that do not exclusively use forest habitat).**

What about the other two bird groups?

**RESPONSE: We have altered this text to make clear we predict that spatial variation in forest**
**characteristics, management regimes, and bird communities will lead to differences in relationships**
**between excavators and forest birds.**

- • I suggest to deeply restructure the introduction: 1) a first paragraph that talks in general about
the relationship of excavators with other birds (fusion of the first two paragraphs); 2) Exclusive
paragraphs for each excavator relationship with each of the other 3 groups of birds. In each
paragraph express what is known and what is the relationship predicted by the authors, and
how they think that these relationships may vary in relation to the biogeographic regions
considered. Another option is that the geographical variations of the relationships are expressed
together in a separate paragraph. On the other hand, I think that it must necessarily be
emphasized and made clear, what is the novelty provided by this study, in comparison with
other studies cited in the same introduction and that address comparisons very similar to those
of this study (eg Drever et al. 2008, Mikusinski et al 2001, Virkkala 2006).

**RESPONSE: We thank both Reviewer 1 and 2 for this suggestion for restructuring, and have adopted**
**these suggestions. We have given more attention to the novelty of our study in the very first**
**paragraph of our new version (Larger scale, testing for general patterns across all major biogeographic**
**areas), and have further regrouped and organized the paragraphs in the introduction.**

- • I find necessary to clearly define the categories of bird groups you use. I tried to do this by copy-
and-paste some information you wrote below. However, I don't know if I did it correct, and the
definition of 'excavator' group is missing. Please, check it and correct it. Yet, something is not
clear to me, Can a given bird species be part of more than one group? If the answer is yes, how
that influences on results?

**RESPONSE: We have added some text to clarify this in the methods and provided clear definitions of**
**all bird groups used (see also our response to an earlier query above.**

- • I find necessary to clarify some 'ecological' issues (i.e. not statistical) in methods. 1) Did you
include the whole area of each biogeographic zone? or 2) only the forest areas in each one of
them? If (1): How do you think the different amount of area/forest area of each biogeographic
zone can influence your results? How non-forest excavators (as some woodpeckers) can
influence your results? or did you exclude them? Another question: given the consideration of
Palearctic and Oriental regions, what does the AUSTRALASIA region includes? what part of Asia?

**RESPONSE: We thank the reviewer for this question. We have added text to explain this in the**
**Methods: "We did not restrict our analyses to forested regions, wanting to avoid making subjective**
**decisions on the classification of a grid cell to a specific vegetation/habitat type. Yet, to make sure we**
**did look for possible relationships between these forest-associated species in regions they actually**
**occur, we did remove cells with zero excavators/woodpeckers and cells where the numbers of**
**secondary cavity-nesters, forest specialists, or all forest birds were lower than that of**
**excavators/woodpeckers from further analyses. "**

**That said, we are not sure how the amount of area/forest area would influence our results, but we**
**have aimed to circumvent that issue as much as possible by randomly selecting (1000 times) 1000 grid**

cells per region, thereby comparing similar surface (though not necessarily forest, but see above)
areas per region.

**Australasia encompasses the following regions, following our adaptation of the classification of van**
**der Hoek et al. 2017. Diversity and Distributions, which is in turn based on that of the Handbook of**
**the Birds of the World (Del Hoyo et al.): American Samoa, Australia, Bismarck Archipelago, Admiralty**
**Islands, Cook Islands, Easter Island, Federated States Micronesia, Fiji, French Polynesia, Guam and the**
**Marianas, Johnston Island, Kiribati, Macquarie Island, Marshall Islands, Nauru, New Caledonia, New**
**Guinea, New Zealand, Niue, Norfolk Island, Papua New Guinea, Pitcairn Islands, Samoa, Solomon**
**Archipelago, Tokelau, Tonga, Tuvalu, Vanuatu, Wake Island, Wallis and Futuna. Practically, we can say**
**that most excavators in this region are from (Papua) New Guinea and surrounding islands.**

- • What did you specifically model? which were the variables?

**RESPONSE: The gls models are more clearly explained now: “ This implied the use of gls models with**
**excavator richness as a predictor variable, the richness of one of the categories of forest-associated**
**birds as a response variable, and a variable exponential spatial covariance structure...”**

- • I am not an specialist in this kind of analyses, and I could not fully understand it. Given that
many readers of this kind of papers are not specialists either, I find very important to better
explain this methodological part. What are the predicted and what the observed data? What
does tell us the regression slopes and what the mean spearman rho? Nevertheless, at this point,
I have the impression that you modeled the richness of non-excavators and forest birds, relative
the presence of excavators (explanatory variable), i.e. in a causal manner (= gls models).
However, in the introduction this relationship was mostly raised in a correlational manner, this
is, that both excavators and the other forest bird groups show similar responses to forest
structure and disturbances. Exceptions to this correlational relationship are, for example, the
FEW cases (in terms of global areas) in which excavators are essential for nonexcavators by
providing new cavities or food resources (sap and insects in excavator perforations). In short,
you are analysing a relationship that you mostly posted as correlational, with regression models.
Please, if my reasoning is incorrect, just ignore it, but explain better the methodology.

**RESPONSE: We have added some clarification (see e.g., the answer above) that may help understand**
**these methods better.**

**That said, it is correct that a gls model is basically an ordinary linear regression model with some**
**different assumptions (or some assumptions not met; in this case related to spatial autocorrelation).**
**However, whereas the reasoning behind a correlation vs. a regression differ, the statistics are, in this**
**particular case, the same. A correlation can indeed encompass any relationship between two**
**variables, without assuming cause and effect, whereas a linear regression can only be that: a linear**
**relationship where we assume that X predicts Y. When the correlation is linear though, this becomes**
**one and the same in terms of output (though not interpretation, which we therefore have been**
**careful with, see our Discussion). In other words, when in this particular case both tests address the**
**‘null hypothesis that the two variables are not linearly related. If run on the same data, a correlation**

test and slope test provide the same test statistic and p-value.' (see e.g., <http://sites.utexas.edu/sos/guided/inferential/numeric/bivariate/cor/>).

So why then this regression and not a correlation, if they provide the same outcome but correlations are easier to interpret? Largely because we would be unable to incorporate a measure to address spatial autocorrelation into a correlation test.

- given that in Fig. 1 you detailed ellipses for each of the six zoogeographic region. I'd say that you can leave only the figure 2 and nothing would be lost

RESPONSE: We have followed this suggestion and removed the first of the two figures, leaving only the multi-paneled one. That said, we opted to add one map that shows global distributions of excavators relative to secondary cavity-nesters, a map which we deem to be illustrative of the spatial variation and clustering (e.g., between zoogeographic regions) that we discuss in our manuscript.

- It seems like sometimes your reasoning is associative/correlative (e.g. both excavators and other groups of forest birds have similar responses to disturbances), other times it is explanatory (e.g. the excavator richness determines the richness of other non excavator cavity-users by providing new cavities), and here it is predictive: the addition of 1 excavator species implies the addition of X forest bird species, ignoring any other type of causes, such as phylogenetic, biogeographical, etc. You must decide which of these visions you are going to explore, or clearly establish that all of them will be analysed, with their respective methodologies and theoretical support.

RESPONSE: Although regressions provide both explanatory and predictive outcomes we understand that our language here is confusing, and our statements may be misinterpreted as meaning one can simply predict from a given number of excavators how many other forest birds there are at that location (which is, of course, not that simple as you rightly point out). Yet, whether the tested relationships are associative (similar habitat needs etc.) or causative (birds use the cavities created by excavators), the fact of the matter is that they are linear. But, in some regions, this relationship is stronger than in others. We should have focused on this and not the steepness of the slopes; and we have therefore removed these parts from the Results and Discussion sections.

That said, we explore whether there are relationships between excavators and other forest birds (the GLS models themselves), but also aimed to see if this could then be used to determine to what extent excavator richness is indicative of that of other species (the matching with the test data set and derivation of Spearman rho's). We hope that this approach is clearer from our revised manuscript.

- Commented [MOU25]: What exactly does the Rho (=strength) mean in this context? low values of Rho mean that the relationships are not very reliable? for example in australasian region. And more important, what would be the joint interpretation of the slopes, the percentage of models with p greater than 0.5 and the rho?

RESPONSE: Rho would here give us an idea how 'perfect' the linear relationships are (with rho = 1 being perfect with all grid cells falling on one line), whereas the slope (which we now removed from further discussion (see above) but present in the Results section to be complete in our presentation of model statistics) would tell us how many other forest birds (other than excavators) there are likely to be in a given location (grid cell) relative to the number of excavators (this does thus present an idea of

**'what richness' a certain number of excavators would possibly indicate). The % of models with a $p < 0.05$ gives us an idea whether these relationships are significant or not, given we had to iterate over**
**many (1000) models, this would be a measure of reliability of the relationships given the combination**
**of both the correlation coefficient and the sample size (which here is always 1000).**

**Low values of rho do not necessarily mean that that the relationships are not very reliable, but rather**
**that they would not be strong enough to be able to 'use richness of excavators' to predict how many**
**other forest birds are in a given location (here grid cell). Thus low values of rho would, in this case,**
**show that excavators are rather poor indicators of other forest birds, even though the relationships**
**might be real and significant.**

- • Commented [MOU27]: I would like to read what are those differences to take into account from
a practical point of view and Commented [MOU28]: The comment on nest-webs here, in the
first paragraph of Discussion, is out of context. Besides, variation among nestwebs... in relation
to what? Are you trying to relate the association of excavators and other forest birds with
variations in nestweb characteristics? it's far from clear, and you are leaving out a key aspect:
the diversity, persistence and characteristics of trees with cavities

**RESPONSE: We have made some substantial changes to our text and hope to now provide a clearer**
**line of argumentation.**

- • However, in several other nestwebs (for example in South America [Cockle et al. 2012, Ruggera
et al. 2016] and India [Manikandan & Balasubramanian 2018]), most non excavator birds mainly
use decay-formed cavities in living trees. Only 10-25% of interactions (including excavator
interactions) are made with snags. So, the evidence to date (even in the Neotropics, where your
results showed the strongest association between excavators and non-excavators), would not
be explained by the number of snags or the amount of woodpecker-excavated cavities. Also,
you need to better explain why forest specialist birds would benefit from the presence of
'certain levels (how much?) of availability of decaying and dead wood'

**RESPONSE: We have made substantial changes to the text in the Discussion, incorporating suggestions**
**by both reviewers, which means that we now provide a clearer insight in the differences in the**
**importance of decay vs. excavated cavities in the different regions. With that, we anticipate to have**
**resolved these issues as well.**

- • differences in relation to what of the nestwebs? you need to be more specific. You need to
better explain this: how speciation and other macro-scale processes across the globe influence
on the results you found. Explanations generally are biased to nestwebs. However, you have
other bird groups that not necessarily are cavity-nesters. Also, you don't necessarily have to
invoke networks (of which there are relatively few reports worldwide) to talk about
cavitynesters (of which there are comparatively many more reports around the world)

**RESPONSE: We thank the reviewer for this comment. We have made substantial changes to our text,**
**reflecting better that both direct facilitation through nest-webs and indirect associations play a role in**
**shaping the observed relationships. As a result of these changes, we have also paid more attention to**
**the role of differences in community composition etc. in shaping these relationships.**

- Australasia region doesn't lack Piciformes. As you say a few words later, SE Asia has a high diversity of Piciformes. So, be careful when you explain Australasia results as a unique region (see comment also about AUSTRALASIA in methods). In fact, I would like to find a better explanation of the results of the Australasian region, since it had the most contrasting results with the rest of the regions.

RESPONSE: There is a difference between the Oriental and Australasian zoogeographic regions that may have gone unnoticed to the reviewer. Most of Indonesia and other SE Asian countries falls within the Oriental region, with the Wallace line separating this region from the Australasian region (e.g., Papua/New Guinea) where some excavators exist but Piciformes are notably absent (see also e.g., Mikusiński, G. (2006, January). Woodpeckers: distribution, conservation, and research in a global perspective. In *Annales Zoologici Fennici* (pp. 86-95). Finnish Zoological and Botanical Publishing Board.). That said, we do agree that the results we found for the Australasian region deserved more attention, and have thus extended on this in the text.

- what differences? you have not established what are the differences along the neotropical region in terms of climatic variables, tree species composition, tree availability and their formation agents. Also, include in this comparison the work by Ruggera et al. 2016, how their results fit in the explanation your trying to develop?

RESPONSE: In our revised version you will find references to Ruggera et al. as well as more comprehensive explanation of the differences found across and within regions.

- then, if non-excavators do not 'need' excavators as cavity-forming agents, why did they show the greatest association in richness in the Neotropics?

RESPONSE: We anticipate that we have now explained this better, focusing on the fact that indirect associations (shared forest elements etc.) are more important than direct facilitation (cavities) in explaining these relationships.

- Commented [MOU40]: Altamirano et al. 2017 wrote: 'Among SCNs, ten species (40%) were obligate and 15 (60%) were non obligate cavity nesters. Non obligate cavity nesters included six (24%) facultative, five (20%) marginal, and four (16%) incidental SCN'. If we include also PCN, facultative cavity nesters are 20%. Commented [MOU41]: how are they influenced? Again, I lost the point. You began talking about 'spatial variation in the relationships between excavators and other forest birds within zoogeographic regions', but then you focused (again) only on excavators and non-excavators, especially in the Neotropics.

RESPONSE: We have changed these parts of text substantially, hopefully addressing these issues and providing more clarity.

- I believe that this is a central point, that it should be better developed in the introduction, and that the methodology should be chosen according to that. It is not the same to analyze something in a correlative way, that in an explanatory or predictive way

RESPONSE: Yes, we agree. We have made an effort to provide more attention to these two different mechanisms (direct facilitation and indirect association) by which relationships between excavators and other forest birds may be formed.

Appendix D

We thank the Editor and reviewers for the chance to submit a revised version of our manuscript, and for the many constructive comments we received. As per request from the editor and reviewers, we have redone some analyses (specifically we now present non-overlapping 'response groups' of forest birds), use ecological language rather than the statistical aspects of the correlations (slopes etc.), and discuss in more detail the results as they pertain to evidence for the two main drivers of possible relationships between excavators and forest birds.

We provide more detailed responses below, and hope you will find this renewed version sufficiently improved to merit publication.

Ps. We would like to point out that most changes we made can also be found in the track change version of our new manuscript, though the final version without track changes has a few more grammar, writing, and outline changes and should thus be considered as the final version.

Response to the reviewers are preceded by an *

- 1) work on the predictions, making them more precise, with more ecological language, and that revolve around the two main causes that the authors propose as the main drivers determining the relationships of excavators and forest birds: the sharing of habitat requirements, and the provision of excavated cavities. How these causes spatially vary based on geographic, climatic and historical characteristics in each region, will determine the predictions that will be tested.

*We thank the reviewer for the comments and have made substantial changes to the Introduction (especially the prediction part). We have reduced our attention to secondary topics, such the analyses of the relationships that include woodpeckers (which are now in supplementary materials) and statistical elements (such as the steepness of slopes), and instead focus more on the two drivers mentioned in this comment. Where possible, we aimed to also strengthen our discussion of differences across regions, though we are admittedly limited in our knowledge of the more tropical regions (e.g., the Afrotropics), for which there is but little relevant literature available.

- 2) rethink the overlapping of the bird categories, assessing which is the best option to determine which of the aforementioned factors (i.e. sharing of habitat requirements and cavity provisions), are the most likely drivers of the proposed relationships. As I comment in the text, the definition of the categories is inconsistent with the methodology used. And, if I understood well the methodology, I also have the impression that the overlapping of categories used may be influencing the results, inflating the relationships between excavators and forest birds, since in this last category are also included many SCNs and forest specialists. I have seen your response to a related comment I made in the previous version. I guess that a good possibility is that the only categories that overlap are 2 (SCN) and 3 (forest specialists). That way, you can see relationships that supposedly may be due more exclusively to provision of cavities (relationships of excavator with 2 would be stronger than with 3), or more related to sharing habitat requirements (relationships of excavator with 3 would be stronger than with 2). Also, leaving category 4 without overlapping can lead to different results and interpretations I think could be more interesting.

*We agree that this decision, to use nested categories, might limit our ability to separate relationships that are more shaped by the provision of cavities from those relationships that rely solely on mutually shared preferences for certain habitat elements etc. We therefore opted to re-run all the analyses for groups that do not have any overlaps. For example, in this revised manuscript version, the group of forest specialists does not include any secondary cavity-nesters, etc.

The results of these new analyses are largely in line with those we obtained previously, but there are some changes that we have more extensively discussed in Results and Discussion. Notably though, the relationships between excavators and forest birds that do not nest in cavities are now particularly strong (in most cases, stronger than those between excavators and secondary cavity-nesters). In other words, we [still] see that excavators are mainly indicators of other birds probably due to their mutual association with habitat elements etc., rather than the provision of cavities.

3) The discussion needs a lot of work. I would suggest that you focus on the differences between the zoogeographic regions, which is the main objective of the paper and what has been evaluated. The differences that may exist within each region should be minimized to the maximum, and only mentioned when strictly necessary. The discussion still has a great bias to try to explain the relationships between excavators and SCN, and there is a considerably smaller space in trying to explain the relationships with forest birds that do not nest in cavities. Finally, relationships in the Australasia region, which showed a different pattern from the rest (even in sign), were not explained, and these results are practically ignored when proposed, from the title, and the first and last paragraph of the discussion, that relationships are common to all zoogeographic regions.

*We did a major rewrite of the discussion to reflect our revised strong and consistent results. We agree that the differences between the zoogeographic regions are important, though we personally think another topic is the main objective of this paper: that excavators can be indicators of other birds at global scales. Yet, we see that we need to minimize our attention to within regions variation, and have done so, and understand that we need to emphasize how the Australasian region differs from others (and attempt to explain why). To that end, and as per your request, we removed some of the analyses that concern Australasia, notably those on woodpeckers which are not found in the region (but which we erroneously, and inaccurately, included in some of the tables and text).

Overall, we re-worked the Discussion, also taking into account the comments below and in the PDF, to provide a simpler but clearer story.

Many other comments have been made in the attached file

Reviewer: 1

Comments to the Author(s)

Line 130: add a bracket at the end of the sentence

*done

Lines 164-187: I opt for a very short explanation what do you mean by "universal" used in the title of the paper already here.

*We have removed the word 'universal' from the title following other reviewer comments, but have nevertheless added some text in this paragraph to illustrate that we expect these relationships to be of similar nature across the world.

Lines 268-277: In contrast with added information about not analysing woodpeckers in Australasia (lines 236-238), you report it here and in Table 2. It is confusing to me.

*This text has been altered, and there is no reporting of results for Australia left in this part of the manuscript.

Line 293: I have a hard time finding the paper by Tikkanen et al. 2006 as supporting your claim here. Possibly cite several paper based on single-species woodpecker studies or use some of those:

JM Roberge, P Angelstam, MA Villard. 2008. Specialised woodpeckers and naturalness in hemiboreal forests—deriving quantitative targets for conservation planning. *Biological conservation*, 2008

or

Angelstam, P., & Mikusiński, G. 1994: Woodpecker assemblages in natural and managed boreal and hemiboreal forest - a review. *Annales Zoologici Fennici* 31: 157-172.

*We think the suggestions are very valid references here, but we think Tikkanen et al. 2006's reference is a perfect fit here, and is actually very easy to find (Google 'Red-listed boreal forest species of Finland: associations with forest structure, tree species, and decaying wood. ' provides it right away). That said, we do see that especially Roberge and Angelstam would be a good fit here as well, and have included that reference accordingly.

Lines 392-394: I suggest adding further argument here i.e. that woodpeckers are highly responsive to playbacks (both calls and particularly drumming) that, if applied in right season are very effective survey tool and cover relatively large areas. See e.g. Kumar, R. and Singh, P. (2010), Determining woodpecker diversity in the sub-Himalayan forests of northern India using call playbacks. *Journal of Field Ornithology*, 81: 215-222. doi:10.1111/j.1557-9263.2009.00267.x or Jeremy A. Baumgardt, Joel D. Sauder, and Kerry L. Nicholson (2014) Occupancy Modeling of Woodpeckers: Maximizing Detections for Multiple Species With Multiple Spatial Scales. *Journal of Fish and Wildlife Management*: December 2014, Vol. 5, No. 2, pp. 198-207.

*We again thank the reviewer for this valuable recommendation. We have added such text at this point as well as the Baumgardt et al. 2014 reference.

Table 2: remove data concerning the woodpeckers (see my comment above)

*We have changed Table 2 to be a new supplementary table (Table S2).

There is one important paper dealing with woodpeckers at the global scale that is relevant to your study but omitted namely: Vergara-Tabares, D. L., M.Lammertink, E. G.Verga, A.Schaaf, and J.Nori (2018). Gone with the forest: Assessing global woodpecker conservation from land use patterns. *Diversity and Distributions* 24:640–651.

Please consider mentioning the results of this paper in the Introduction or Discussion. It is particularly interesting from the management perspective and the human influence on forest qualities.

*We were aware of this paper but previously did not see a particular text location where this reference would fit well. But, in this revised version, we found that this reference fits very well in the Discussion and strengthens our story.

*A few specific responses to comments made to the PDF (other changes can be found in the document version with track changes):

- After reading your response to my comment on the title, I must kindly say that I still disagree with the use of the word 'universal'. First, because the meaning of the word itself is more a synonym for global, than what you want to express; and second, because what you found is not universal (in the sense that you want to give it), the Australasia region does not have the same relationships, either in strength or in sign, as the other regions.

*We have removed the word universal from the title following this recommendation.

- Even when I understand your point here, when you say 'natural', in my opinion this is a long-standing misuse of the word. Bird-excavated cavities are also 'natural'. I suggest changing those terms, maybe write directly 'fungal formation'?

*We agree with this observation and have altered the text to say 'fungal formation'.

- I don't see a clear and direct link between the first two sentences of the paragraph. Moreover, as I mentioned in the previous version, nest-webs involve only one of the subsets of birds that you are trying to associate with excavators. I suggest deleting the first sentence, and slightly modifying the second to be the beginning of the paragraph

*We have made considerable changes in the text in this section.

- Taking into account these arguments, how do you explain that in the Neotropical region, where the relationship between excavators and SCN found is the strongest association between excavators and SCN in the world, SCN do not rely heavily on excavated cavities?

*We refocused on the relatively more important role of indirect association, rather than direct facilitation, in shaping the relationships between excavators and other forest birds. As a result, we largely changed the text in the Discussion, including providing explanations related to the relative 'natural state' of forests in the Neotropics as compared to other regions, the larger number of niches available in the Neotropics, etc. We kindly refer to the altered Discussion for all details.

Appendix E**ROYAL SOCIETY
OPEN SCIENCE****Global relationships between tree-cavity excavators and
forest bird richness**

Journal:	Royal Society Open Science
Manuscript ID	RSOS-192177.R1
Article Type:	Research
Date Submitted by the Author:	31-Jan-2020
Complete List of Authors:	van der Hoek, Yntze; The Dian Fossey Gorilla Fund International, Karisoke Research Center Gaona, Gabriel; Justus Liebig Universität Giessen Graduiertenzentrum Ciach, Michal; University of Agriculture in Krakow Martin, Kathy; University of British Columbia,
Subject:	ecology < BIOLOGY
Keywords:	Facilitator species, Indicator species, Species interactions, Picidae, Management surrogates, Secondary cavity-nesting birds
Subject Category:	Ecology, Conservation, and Global Change Biology

Author-supplied statements

Relevant information will appear here if provided.

Ethics

Does your article include research that required ethical approval or permits?:

This article does not present research with ethical considerations

Statement (if applicable):

CUST_IF_YES_ETHICS :No data available.

Data

It is a condition of publication that data, code and materials supporting your paper are made publicly available. Does your paper present new data?:

Yes

Statement (if applicable):

Links to the data and R scripts to reproduce analyses presented in Global and universal relationships between tree-cavity excavators and forest bird richness can be found in Appendix S1.

Conflict of interest

I/We declare we have no competing interests

Statement (if applicable):

CUST_STATE_CONFLICT :No data available.

Authors' contributions

This paper has multiple authors and our individual contributions were as below

Statement (if applicable):

YH and GVG carried out the data analysis, and led the design of the study and writing of the manuscript; MC and KM critically revised the manuscript. All authors contribute to the writing of the final manuscript, gave final approval for publication and agree to be held accountable for the work performed therein.

We thank the Editor and reviewers for the chance to submit a revised version of our manuscript,
and for the many constructive comments we received. As per request from the editor and
reviewers, we have redone some analyses (specifically we now present non-overlapping
'response groups' of forest birds), use ecological language rather than the statistical aspects of
the correlations (slopes etc.), and discuss in more detail the results as they pertain to evidence
for the two main drivers of possible relationships between excavators and forest birds.

We provide more detailed responses below, and hope you will find this renewed version
sufficiently improved to merit publication.

Ps. We would like to point out that most changes we made can also be found in the track
change version of our new manuscript, though the final version without track changes has a few
more grammar, writing, and outline changes and should thus be considered as the final version.

Response to the reviewers are preceded by an *

- 1) work on the predictions, making them more precise, with more ecological language, and
that revolve around the two main causes that the authors propose as the main drivers
determining the relationships of excavators and forest birds: the sharing of habitat
requirements, and the provision of excavated cavities. How these causes spatially vary
based on geographic, climatic and historical characteristics in each region, will
determine the predictions that will be tested.

*We thank the reviewer for the comments and have made substantial changes to the
Introduction (especially the prediction part). We have reduced our attention to secondary
topics, such the analyses of the relationships that include woodpeckers (which are now
in supplementary materials) and statistical elements (such as the steepness of slopes),
and instead focus more on the two drivers mentioned in this comment. Where possible,
we aimed to also strengthen our discussion of differences across regions, though we are
admittedly limited in our knowledge of the more tropical regions (e.g., the Afrotropics),
for which there is but little relevant literature available.

- 2) rethink the overlapping of the bird categories, assessing which is the best option to
determine which of the aforementioned factors (i.e. sharing of habitat requirements and
cavity provisions), are the most likely drivers of the proposed relationships. As I
comment in the text, the definition of the categories is inconsistent with the methodology
used. And, if I understood well the methodology, I also have the impression that the
overlapping of categories used may be influencing the results, inflating the relationships
between excavators and forest birds, since in this last category are also included many
SCNs and forest specialists. I have seen your response to a related comment I made in
the previous version. I guess that a good possibility is that the only categories that
overlap are 2 (SCN) and 3 (forest specialists). That way, you can see relationships that
supposedly may be due more exclusively to provision of cavities (relationships of
excavator with 2 would be stronger than with 3), or more related to sharing habitat
requirements (relationships of excavator with 3 would be stronger than with 2). Also,
leaving category 4 without overlapping can lead to different results and interpretations I
think could be more interesting.

*We agree that this decision, to use nested categories, might limit our ability to separate
relationships that are more shaped by the provision of cavities from those relationships that
rely solely on mutually shared preferences for certain habitat elements etc. We therefore
opted to re-run all the analyses for groups that do not have any overlaps. For example, in
this revised manuscript version, the group of forest specialists does not include any
secondary cavity-nesters, etc.

The results of these new analyses are largely in line with those we obtained previously, but
there are some changes that we have more extensively discussed in Results and
Discussion. Notably though, the relationships between excavators and forest birds that do
not nest in cavities are now particularly strong (in most cases, stronger than those between
excavators and secondary cavity-nesters). In other words, we [still] see that excavators are
mainly indicators of other birds probably due to their mutual association with habitat
elements etc., rather than the provision of cavities.

3) The discussion needs a lot of work. I would suggest that you focus on the differences
between the zoogeographic regions, which is the main objective of the paper and what has
been evaluated. The differences that may exist within each region should be minimized to
the maximum, and only mentioned when strictly necessary. The discussion still has a great
bias to try to explain the relationships between excavators and SCN, and there is a
considerably smaller space in trying to explain the relationships with forest birds that do not
nest in cavities. Finally, relationships in the Australasia region, which showed a different
pattern from the rest (even in sign), were not explained, and these results are practically
ignored when proposed, from the title, and the first and last paragraph of the discussion, that
relationships are common to all zoogeographic regions.

*We did a major rewrite of the discussion to reflect our revised strong and consistent results.
We agree that the differences between the zoogeographic regions are important, though we
personally think another topic is the main objective of this paper: that excavators can be
indicators of other birds at global scales. Yet, we see that we need to minimize our attention
to within regions variation, and have done so, and understand that we need to emphasize
how the Australasian region differs from others (and attempt to explain why). To that end,
and as per your request, we removed some of the analyses that concern Australasia,
notably those on woodpeckers which are not found in the region (but which we erroneously,
and inaccurately, included in some of the tables and text).

Overall, we re-worked the Discussion, also taking into account the comments below and in
the PDF, to provide a simpler but clearer story.

Many other comments have been made in the attached file

Reviewer: 1

Comments to the Author(s)

Line 130: add a bracket at the end of the sentence

*done

Lines 164-187: I opt for a very short explanation what do you mean by "universal" used in
the title of the paper already here.

*We have removed the word 'universal' from the title following other reviewer comments, but
have nevertheless added some text in this paragraph to illustrate that we expect these
relationships to be of similar nature across the world.

Lines 268-277: In contrast with added information about not analysing woodpeckers in
Australasia (lines 236-238), you report it here and in Table 2. It is confusing to me.

*This text has been altered, and there is no reporting of results for Australia left in this part of
the manuscript.

Line 293: I have a hard time finding the paper by Tikkanen et al. 2006 as supporting your
claim here. Possibly cite several paper based on single-species woodpecker studies or use
some of those:

JM Roberge, P Angelstam, MA Villard. 2008. Specialised woodpeckers and naturalness in
hemiboreal forests—deriving quantitative targets for conservation planning. *Biological*
*conservation*, 2008

or

Angelstam, P., & Mikusiński, G. 1994: Woodpecker assemblages in natural and managed
boreal and hemiboreal forest - a review. *Annales Zoologici Fennici* 31: 157-172.

*We think the suggestions are very valid references here, but we think Tikkanen et al.
2006's reference is a perfect fit here, and is actually very easy to find (Google 'Red-listed
boreal forest species of Finland: associations with forest structure, tree species, and decaying wood.
' provides it right away). That said, we do see that especially Roberge and Angelstam would be a
good fit here as well, and have included that reference accordingly.

Lines 392-394: I suggest adding further argument here i.e. that woodpeckers are highly
responsive to playbacks (both calls and particularly drumming) that, if applied in right
season are very effective survey tool and cover relatively large areas. See e.g. Kumar, R.
and Singh, P. (2010), Determining woodpecker diversity in the sub-Himalayan forests of
northern India using call playbacks. *Journal of Field Ornithology*, 81: 215-222.
doi:10.1111/j.1557-9263.2009.00267.x or Jeremy A. Baumgardt, Joel D. Sauder, and Kerry
45 L. Nicholson (2014) Occupancy Modeling of Woodpeckers: Maximizing Detections for
Multiple Species With Multiple Spatial Scales. *Journal of Fish and Wildlife Management*:
December 2014, Vol. 5, No. 2, pp. 198-207.

*We again thank the reviewer for this valuable recommendation. We have added such text
at this point as well as the Baumgardt et al. 2014 reference.

Table 2: remove data concerning the woodpeckers (see my comment above)

*We have changed Table 2 to be a new supplementary table (Table S2).

There is one important paper dealing with woodpeckers at the global scale that is relevant to
your study but omitted namely: Vergara-Tabares, D. L., M.Lammertink, E. G.Verga,
7 A.Schaaf, and J.Nori (2018). Gone with the forest: Assessing global woodpecker
conservation from land use patterns. Diversity and Distributions 24:640–651.

Please consider mentioning the results of this paper in the Introduction or Discussion. It is
particularly interesting from the management perspective and the human influence on forest
qualities.

*We were aware of this paper but previously did not see a particular text location where this reference
would fit well. But, in this revised version, we found that this reference fits very well in the Discussion
and strengthens our story.

*A few specific responses to comments made to the PDF (other changes can be found in the document
version with track changes):

- • After reading your response to my comment on the title, I must kindly say that I still disagree
with the use of the word 'universal'. First, because the meaning of the word itself is more a
synonym for global, than what you want to express; and second, because what you found is not
universal (in the sense that you want to give it), the Australasia region does not have the same
relationships, either in strength or in sign, as the other regions.

*We have removed the word universal from the title following this recommendation.

- • Even when I understand your point here, when you say 'natural', in my opinion this is a long-
standing misuse of the word. Bird-excavated cavities are also 'natural'. I suggest changing those
terms, maybe write directly 'fungal formation'?

*We agree with this observation and have altered the text to say 'fungal formation'.

- • I don't see a clear and direct link between the first two sentences of the paragraph. Moreover,
as I mentioned in the previous version, nest-webs involve only one of the subsets of birds that
you are trying to associate with excavators. I suggest deleting the first sentence, and slightly
modifying the second to be the beginning of the paragraph

*We have made considerable changes in the text in this section.

- • Taking into account these arguments, how do you explain that in the Neotropical region, where
the relationship between excavators and SCN found is the strongest association between
excavators and SCN in the world, SCN do not rely heavily on excavated cavities?

*We refocused on the relatively more important role of indirect association, rather than direct
facilitation, in shaping the relationships between excavators and other forest birds. As a result, we
largely changed the text in the Discussion, including providing explanations related to the relative
'natural state' of forests in the Neotropics as compared to other regions, the larger number of
niches available in the Neotropics, etc. We kindly refer to the altered Discussion for all details.

| 1 Title: Global ~~and universal~~ relationships between tree-cavity excavators and forest bird richness

Running title: Excavator and forest bird relationships

Yntze van der Hoek^{1,2*}, Gabriel V. Gaona¹, Michał Ciach³ and Kathy Martin^{4,5}

¹ Universidad Regional Amazónica Ikiam, Vía Muyuna, Kilómetro 7, Tena, Ecuador.

² The Dian Fossey Gorilla Fund International, Musanze, Rwanda.

³ Department of Forest Biodiversity, Institute of Forest Ecology and Silviculture, Faculty of
Forestry, University of Agriculture, al. 29 Listopada 46, 31-425 Kraków, Poland.

⁴ Department of Forest and Conservation Sciences, University of British Columbia, 2424 Main
Mall, Vancouver, British Columbia V6T 1Z4 Canada.

⁵ Environment and Climate Change Canada, 5421 Robertson Road, R.R. 1, Delta, British
Columbia V4K 3N2, Canada.

*Corresponding author: yntzevanderhoek@gmail.com

Abstract

Global monitoring of biodiversity and ecosystem change can be aided by the effective use of indicators. Tree-cavity excavators, the majority of which are woodpeckers (Picidae), are known to be useful indicators of the health or naturalness of forest ecosystems and by the diversity of forest birds. They are indicators of the latter due to their associations with particular forest elements and because of their role in facilitating other species through the provision of nest cavities. Here, we investigated whether these positive correlations between excavators and other forest birds are also found at global scales. We used global distribution maps to extract richness estimates of tree-cavity nesting and forest-associated birds, which we grouped by zoogeographic regions. We then created generalized least squares (gls) models to assess the relationships between these groups of birds. We show that richness of tree-cavity excavating birds correlates positively with that of cavity-nesters and forest birds at global scales, but with variation across zoogeographic regions. As many excavators are relatively easy to detect, play keystone roles at local scales, and are effective management targets, we propose that excavators are useful for biodiversity monitoring across multiple spatial scales and geographic regions, especially in the tropics.

Keywords

Facilitator species, Indicator species, Management surrogates, Picidae, Species interactions, Woodpeckers

Introduction

Forests worldwide are facing increasing anthropogenic pressures, with both a rapid decline in the
surface area of natural forests and a decrease in the naturalness of forests as a result [1, 2]. In
turn, these reductions of high-quality forest habitats have led to a loss of forest-associated
biodiversity [3-5]. To monitor these changes in forest ecosystems and their denizens, we often
look at specialized animals that require the availability of specific habitat structures and
processes across long temporal and large spatial scales [6], such as birds that respond not only to
a loss in overall forest cover but also to a change in forest health, quality, and integrity [7-10].
Excavators such as woodpeckers, barbets, nuthatches, trogons and certain parrot species—among
which woodpeckers are the most numerous group [11]—may be especially effective indicators
due to their associations with particular forest elements and their role in facilitating the presence
of other species through the provision of nest cavities. Indeed, there is empirical evidence that
their presence can be indicative of the state of both the forest (e.g., the presence of large trees
[12], heterogeneous forest structure [13], or a high level of naturalness [14-18]) as well as the
richness and abundance of other forest-associated species [19-21]. However, we may question
whether excavators are universally effective indicators, at least of biodiversity, in all geographic
regions or forest types. For example, excavators are better predictors of richness and abundance
of other forest resident birds in deciduous versus coniferous forests of certain parts of hemiboreal
Europe [15].

If relationships between excavators and other forest-associated biota are universally
positive in nature, and common not only at stand- and landscape-scales but hold across large
geographic regions, then excavators could form a unique group of indicators that may be
effective across multiple locations and spatial scales [22]. To find species that are effective as
both indicator and management surrogates across multiple regions would be especially useful in
the largely understudied tropics where comprehensive biodiversity assessments are costly and
logistically challenging [23]. The first signs are promising as the overwhelming majority of tree-
cavity excavators are forest or woodland birds and general patterns of woodpecker richness
correlate positively with the amount of forest cover at the global scale [24]. Moreover, many
excavators are regarded as habitat specialists and tend to have highly specific habitat
requirements, ~~requirements~~ which are usually only met in forests with high degrees of
naturalness and low levels of disturbances (e.g., logging). In fact, restoration efforts have
effectively made use of woodpeckers to indicate that restoration has successfully reached a
certain level of naturalness [25, 26]. This group of birds often utilizes large, live or decaying
trees, or coarse woody debris, for nesting, roosting, and feeding, and the availability of dead
wood is in particular found to be important [27-29]. This role of excavators as indicators of the
naturalness of ecosystems, whereby their presence correlates strongly with other aspects of
biodiversity, is enhanced by the fact that the majority of species are forest residents, which
arguably makes them more responsive to local changes in habitat quality (see e.g., [30]).

Excavators are not only reliable indicators of other forest birds by means of their mutual
associations with forest characteristics, but correlations between both groups of birds are also
partially determined by species interactions. Taken together, these ‘nest-webs’ [31] of interacting
species are structured by the availability and acquisition of tree-cavities formed by two major
processes: ‘~~funga~~natural formation’, in which fungi decompose wood over an extended period,
or animal excavation [32, 33]. Since most excavators—at least the majority of woodpeckers—
excavate a new nest cavity each year, this mode of cavity creation steadily provides a large
number of potential sheltered roosting and nesting sites ~~available to~~for secondary-users: both for

vertebrates [33] and invertebrates [34, 35]. As a result, cavity formation by excavation is an
important, and often more rapid, -source of nest holes for species in many regions of the world,
especially North America [33]. However, decay-formed cavities have shown to beare likely to be
more important nest sites in other regions, especially the tropics, and it is very likely that the
relative importance of both processes of cavity formation—and thus the strength of the
relationship between excavator richness and secondary cavity-nester richness—varies globally
[33]. ~~Finally, it is worth noting that excavators play additional roles (beyond the direct provision~~
~~of cavities) that may facilitate other species. For example, as excavators perforate the bark of~~
~~trees they may expose insects or sap for other species to forage on [36, 37].~~ In addition,
excavators can aid in the dispersal of fungi, which in turn enhances processes of wood softening
and the formation of decay-formed cavities [368]. Finally, it is worth noting that excavators play
additional roles (beyond the direct provision of cavities) that may facilitate other species that are
not necessarily cavity-nesters. For example, as excavators perforate the bark of trees they may
expose insects or sap for other species to forage on [368].

Local studies of cavity-nesting birds and their interactions are largely biased towards
temperate zones, and only a few authors have explored global variation in nest-web
characteristics or addressed how such variation should be taken into account when considering
cavity-nester communities in forest conservation or management (e.g., [33, 39]). Nevertheless,
we might predict that these birds Excavators could are arguably be reliable indicators of forest-
birdavian diversity and richness not only at stand or landscape-levels [9, 19], but it is unclear if
this, but also across large regions at a global scale, if not for the cavities that excavators supply
but for the associations of excavators with forest elements that are indicative of heterogeneous
forests with a high degree of naturalness. scales up across larger regions or holds across the
globe given, for example, indications that the interactions between cavity-nesting birds may vary
spatially (e.g., [11]) or the notion that certain dominant groups of excavators (woodpeckers in
particular) vary in richness across zoogeographic regions [11]. Therefore, the aim of the present
study is to investigate whether there are consistent patterns in the relationships between avian
tree-cavity excavators (hereafter: excavators) and species richness of non-excavating tree-cavity
nesters (hereafter: secondary cavity-nesters), forest specialist species, and forest birds in general,
across the globe. If previously observed relationships between cavity-excavators and other forest
birds scale up from local forest stands [17] or ecosystems [19] to zoogeographic or global levels,
then this strengthens the potential of tree cavity-excavators as both indicators and management
surrogates.

Given the important, and potential indicator, role of excavators in forest ecosystems
across the world, we predicted a global tendency for that secondary cavity-nester, forest
specialist, forest generalist and overall for bird (specialists + generalists) richness generally
wilto increase with-in correlation the number of excavators present in the ecosystem, regardless
of the ecosystem or location. However, we also predicted that the nature (e.g., the strength and
slope of correlations) of the relationships between excavators and richness of other forest birds
will vary across zoogeographic regions, reflecting differences in forest characteristics (e.g.,
deciduous vs. coniferous forests), forest management practices (e.g., forests being more or less
intensively managed), and bird communities (e.g., the relative richness of different groups of
forest-associated birds), as previously discussed in e.g., [33]. In particular, we predicted that
correlations between excavators and other birds secondary cavity-nesters would be particularly
strong, but with relatively shallow regression slopes, in regions with relatively high intensities of
forest management (vs. unmanaged natural forests) and low overall avian diversity and slow

processes of decay-related tree-cavity formation. In such regions these predominantly temperate
 and boreal regions, particularly such as the Nearctic and Palearctic, relationships between
 excavators and other forest birds and secondary cavity-nesters would be that are driven by mutual
 associations with particular forest elements but would be enhanced by direct interactions through
 the provision of nest cavities, as a lack of available decay-formed cavities may increase the
 dependence of secondary-cavity nesters on excavated cavities. We predicted relatively shallow
 regression slopes in these regions (Nearctic, Palearctic), as there is a relatively low ratio between
 the numbers of excavators and secondary cavity-nesters [11]. In contrast, for regions that
 largely span the tropics, such as the Neotropics, Afrotropics, and Oriental regions, we also
 predicted predict weaker, though still positive, correlations between excavators and secondary
 cavity-nesters other forest birds, but as these relationships would be driven mainly by mutual
 associations with particular forest elements and less by direct interactions through the provision
 of nest cavities—decay-formed cavities being more prevalent here. As a result, we also predict
 that relationships between excavators and secondary cavity-nesters stronger than those
 between excavators and non-cavity-nesting forest birds (specialists or generalists) in the Nearctic
 and Palearctic, but that there is no such clear pattern in of differences in strength of correlations
 for the tropical regions. For Australasia, where the largest group of excavators (woodpeckers) is
 notably absent, we predict relationships between the present excavators, often weaker excavators
 that require softer or rotting wood (see [11] for species), and other groups of birds to be
 relatively weak and to be solely based on certain shared habitat preferences. Finally, for all
 regions, we predict that relationships between excavators and secondary cavity-nesters and
 between excavators and forest specialists stronger than those between excavators and forest
 generalists, as the latter group includes species that are unlikely to have either direct interactions
 or mutually shared habitat preferences with excavators. In fact, many generalist forest birds may
 have traits (e.g., ground-nesting behaviour and a preference for early-successional forests with
 small trees) that can lead to habitat requirements opposite from those required by
 excavators—we predicted these correlations to be weaker. However, as these regions are
 more diverse, we also predicted that there would be relatively higher numbers of forest birds per
 excavator, resulting in steeper regression slopes.

**Material and Methods**

We used published global distribution maps of tree-cavity nesting birds [11] and forest birds
 [410] to extract richness estimates per 10×10 km grid cell (Table 1). Maps for both tree-cavity
 nesters and forest birds were based on the same species range maps provided by BirdLife
 International and NatureServe [421]. We selected and classified focal bird species following [11,
 410] as 1) excavators (species that are known to often or always excavate their own nesting
 cavities), 2) secondary cavity-nesters: birds which are known to nest in tree cavity but which
 never or rarely excavate their own cavity, 3) forest specialists: birds that exclusively use forest
 habitat, and 4) forest birds: birds that use forest habitat but also at least one other type of habitat.
 We note that some species were counted in more than one category (e.g., a species that is a
 ‘forest specialist’ is per definition also counted as a ‘forest bird’). However, excavators were
 never counted in any other category.

We used only range polygons where a species’ presence was classified as ‘Extant’ or
 ‘Probably extant’, and assigned ‘presence’ to all grid cells that overlapped with range polygons.
 Next, we created generalized least squares (gls) models to assess the linear relationships between
 excavators and other categories of forest-associated birds. To take biogeographical differences

into account, we proceeded to separate trends across the globe, and created gls models for the six
zoogeographic regions separately: Nearctic, Palearctic, Neotropical, Afrotropical, Oriental, and
Australasia. We did not restrict our analyses to forested regions, wanting to avoid making
subjective decisions on the classification of a grid cell to a specific vegetation/habitat type. Yet,
to make sure we did look for possible relationships between these forest-associated species in
regions they actually occur, we removed cells with zero excavators/~~woodpeckers~~ and cells where
the numbers of secondary cavity-nesters, forest specialists, or all forest birds were lower than
that of excavators/~~woodpeckers~~ from further analyses.

Our data were spatially structured (i.e., high and low richness of birds in different
categories (excavators etc.) was spatially clustered) across and within zoogeographic regions, as
demonstrated by the global distribution of the relative number of cavity excavators to non-
excavating cavity-nesting birds (expressed as the ratio of excavator / non-excavating cavity-
nester; Fig. 1). Thus, we followed similar protocols to adjust for spatial autocorrelation as per
Storch et al. [432]. This implied the use of gls models with excavator richness as a predictor
variable, the richness of one of the categories of forest-associated birds as a response variable,
and a variable exponential spatial covariance structure, as these models had the best fit in
preliminary tests, which reduced—rather than eliminated—the influence of spatial
autocorrelation on our models. In these tests, exponential gls models consistently had the lowest
Akaike’s Information Criterion (AIC) value compared to a set of models, including an ordinary
least squares model with no spatial structure, and gls models with Gaussian, linear, rational
quadratics and spherical spatial covariance structures.

Even though gls models work well with large datasets, we found that gls models fitted on
all data failed to converge (using the R package “nlme” [443]). To circumvent this computational
problem, we first grouped grid cells by zoogeographic region. We thereafter ran 1000 models on
bootstrap permutations of 1000 randomly selected grid cells (see [24, 454]). Of these 1000 model
iterations, we subsequently calculated the mean (\pm SD), minimum and maximum coefficients of
the regression slope. As gls models do not provide a measure of the strength of relationships, we
proceeded with a different measure of the strength of each correlation: fitting each exponential
gls model to a different, non-overlapping, test data set of 1000 randomly selected grid cells. We
then proceeded to calculate the Spearman rho coefficient of the correlation between predicted
and observed data, for all 1000 models per zoogeographic region. Finally, we calculated the
mean Spearman rho of all significant models per zoogeographic region. We repeated all of these
analyses for woodpeckers as a subset of the excavators, with the notable exception of analyses
for Australasia where woodpeckers are absent. All data, R scripts, and initial results needed to
reproduce our analyses presented here are deposited online (Appendix S1).

Results

Using gls modelling to assess the global patterns for the relationships between excavator richness
and that of secondary cavity-nesters, forest specialists, and all forest birds, we found positive
relationships between all pairs of comparisons. However, these relationships differed in relative
strength and characteristics (e.g., steepness of regression slope)-depending on the group of birds
and zoogeographic region analysed (Fig. 2, Table 21; see Table S1 for an overview of the
number of species per group per region)).

In terms of variation across zoogeographic regions, the relationships between excavators
and all other groups of forest-associated birds were typified by particularly steep regression
slopes in the Neotropics, where the presence of one additional excavator species in a grid cell

implied an increase of more than two additional secondary cavity-nesters, ten forest bird species,
 or four forest specialists. In contrast, regression slopes were less steep in predominantly
 temperate and boreal regions such as the Nearctic and Palearctic, where an increase of one
 excavator species signified an increase of approximately one secondary cavity-nester species,
 four to five forest birds, or one forest specialist. More importantly, and similar to variation in the
 slopes of the regressions, we found differences in the strength (as represented by Spearman's
 rho) of these relationships, with ~~aga~~ We found in stronger relationships (higher Spearman rho) in
 the predominantly tropical Afrotropical, Oriental, and Neotropical regions than in the
 comparably more temperate/boreal Nearctic and Palearctic regions, and particularly weak
 relationships in Australasia (Table 1).

Although ~~t~~The relationships between excavators and the other three groups of cavity-
 nesting and forest-associated birds were generally similar in nature (positive), without clear
 evidence for our prediction that the relationship between excavators and ~~it~~ cavity-nesters
 would be stronger than that between excavators and larger groups of forest-associated birds that
 include non-cavity-nesters (forest specialists or all forest birds). Instead, we found that there
 were some differences in the relative strength of these relationships (between excavators and the
 three groups of forest-associated birds) ~~d~~ed largely across zoogeographic regions. For
 example, excavator richness was a particularly strong predictor of secondary cavity-nester
 richness and of all forest birds in the ~~o~~ntal region ($\rho = 0.88$ and 0.86 , respectively), and
 slightly less so for forest specialists ($\rho = 0.727$), whereas excavator richness in the Afrotropics
 showed a strong relationship with forest specialists ($\rho = 0.83$), forest generalists ($\rho = 0.91$)
 and all forest birds taken together ($\rho = 0.934$) and with forest specialists ($\rho = 0.88$), but a
 somewhat weaker association with secondary cavity-nesters ($\rho = 0.79$). Finally, we note that
 excavators in ~~Nevertheless, despite the complete absence of woodpeckers from Australasia~~
 (which include no woodpeckers) showed a moderately strong ~~this region, we did find a weak~~ (ρ
 $= 0.7369$) but ~~often~~ and significant ($p < 0.05$ in 10097 percent of model iterations) positive
 relationship between richness of excavators and with forest specialists, and none with secondary
 cavity-nesters or more generalist forest ~~s~~.

We repeated all analyses for woodpeckers as a subset of excavators and found nearly
 identical results, ~~which is why we opted to list results of these analyses in supplementary Table~~
 ~~S2 and limit our further discussion of these analyses.~~ with again significant positive relationships
 between woodpeckers and other groups of cavity-nesting or forest-associated birds in all regions
 but Australasia, and with steeper regression slopes and stronger relationships in the
 predominantly tropical Neotropical, Oriental and Afrotropical regions as compared to more
 shallow slopes and weaker associations in the Nearctic and Palearctic (Table 2). The relative
 strength of the relationships between woodpeckers and the three other groups of cavity-nesting
 and forest-associated birds again depended on the zoogeographic region (in similar ways as for
 entire group of excavators) such that woodpeckers were strong predictors of the richness of other
 birds as excavators in general, with the notable exception of Australasia. ~~Nevertheless, despite~~
 ~~the complete absence of woodpeckers from this region, we did find a weak~~ ($\rho = 0.69$) but
 ~~often significant~~ ($p < 0.05$ in 97 percent of model iterations) positive relationship between
 richness of excavators and forest specialists.

Discussion

We found strong positive relationships between excavators/woodpeckers and secondary cavity-
 nesters, forest specialists, forest generalists and forest birds in general and forest specialists at

323 global and zoogeographic scales, although these relationships differed in both strength and
324 characteristics (regression slopes) across zoogeographic regions. Moreover, we found that it
depended on the region whether relationships between excavators and secondary cavity-nesters
were stronger or weaker than those between excavators and non-cavity-nesting forest birds. This
indicates that both direct facilitation of nest-cavities and indirect associations with forest
elements and shared habitat preferences are likely to play a role in shaping these relationships,
but that the relative importance of both depends on the region under consideration. Our results
imply that excavators and woodpeckers (a predominant subset of excavators) hold potential as
indicators and management surrogates across most of the world [465], with the possible
exception of the Australasian region, but also that we need to take differences among
zoogeographic regions (e.g., in the composition of local and regional nest-webs with regards to
~~the relative numbers of excavators and secondary cavity-nesters as seen in Fig. 1)~~ into account if
we are to adopt these as surrogates for conservation or management.

~~Many excavators, woodpeckers included, have strong preferences for specific forest~~
~~elements, such as dead and decaying trees or particularly large trees (see e.g., [28]). In turn, these~~
~~elements, and the processes that drive their presence and abundance, are of great importance to~~
~~habitat formation and create additional niche space to support many groups of organisms [46].~~
~~Therefore, we can expect that forests that meet the habitat requirements of excavators (e.g.,~~
~~certain levels of availability of decaying and dead wood) provide high-quality habitat for many~~
~~other wildlife species. This may explain the higher richness of forest birds in areas occupied by~~
~~high numbers of excavators/woodpeckers. For non-excavating species that utilize tree cavities~~
~~for their nests, this relationship may be enhanced by the fact that excavators/woodpeckers create~~
~~or facilitate potential nest substrates [31]. In other words, the strong habitat preferences of many~~
~~excavators/woodpeckers, as well as the facilitatory role excavators play in providing nest~~
~~substrates, make these birds potentially effective indicators of many forest-associated species~~
~~and, by extension, of high-quality forests (i.e., natural, old-growth; [17, 18].~~

We did not find support for the prediction that relationships between excavators and
secondary-cavity nesters are relatively stronger than those between excavators and secondary
cavity-nesters in temperate and boreal regions (Arctic and Palearctic), which would reflect the
additional importance of cavity facilitation in these regions, nor that the reverse was true in all
predominantly tropical regions. In general, we would predict excavated cavities to be less
important for secondary-cavity nesters in the tropics as high rates of precipitation and
temperature lead to relatively rapid formation of cavities through fungal activity [54] and lower
persistence (i.e., longevity) of excavated cavities [33,55]. This, in turn, should lead to relatively
higher use of decay-formed cavities by secondary cavity-nesters in the tropics [33, 55], and less
dependence on excavators for the supply of nest-cavities. For example, even woodpeckers,
occasionally use decay-formed cavities in Neotropical temperate rainforests where tree decay is
the key driver of nest-web structure [55]. Indeed, we see that the relationship between excavators
and secondary cavity-nesters is not stronger (Neotropics) or even weaker (Afrotropics) than
those between excavators and non-cavity-nesting forest birds (specialists or generalists), in two
of the more tropical regions. But we also found that the relationship between excavators and
secondary cavity-nester was relatively strong in the Oriental region, stronger in fact than those
between excavators and any other group of forest birds. This may be the result of the specific
characteristics of the forests or avifaunal communities in this region, but is also likely to be
influenced by relatively high logging rates in the Oriental region. Logging reduces the
availability of large dead trees, which in turn diminishes substrate availability for excavators as

well as the opportunity for cavities to form via decay [1], potentially making secondary cavity-nesters more dependent on excavated cavities. In contrast to our predictions, relationships between excavators and secondary cavity-nesters and between excavators and forest specialists were not necessarily stronger (though this was the case in the Neotropical region) than those between excavators and forest generalists. We do not have a clear explanation for this finding, though we argue that forest specialists are not necessarily closer linked than forest generalists to forest elements (e.g., snags) also preferred by excavators and that both forest generalists and excavators might share landscapes where forest specialists are hard to find, for example open landscapes with scattered trees [2]. Finally, we found that regardless of the relationship, all relationships between excavators and other bird groups were stronger in the more tropical (especially the Afrotropical and Neotropical) regions, with the exception of Australasia (see below), than in the temperate and boreal regions. Arguably, excavators are particularly strong indicators of other bird groups in these tropical regions as these regions still contain vast tracts of unlogged and unmanaged forest, a topic for further research.

Relationships between excavators and groups of forest-associated birds were less evident in Australasia, which is a region that differs largely from all others in both avifauna (e.g., there are no woodpeckers here but there is a relatively high richness of cavity-nesters [11]) and forest structure and dynamics (e.g., cavity densities in Australasian forests are among the highest globally [54]). Specifically, we found a moderately strong relationship between excavators and non-cavity-nesting forest specialists in Australasia, but no such relationships between excavators and secondary-cavity nesters or forest birds with broader habitat requirements (i.e. forest generalists). This, we reason, indicates that the excavators in this region (which notably excludes woodpeckers) show a preference for forest conditions (e.g., the provision of trees that are in advanced stages of decay, given that many of these excavators are weaker excavators that require softer or rotting wood; see [11] for species) that are shared by many forest specialists. In contrast, secondary cavity-nesters in Australasia may depend on elements not formed, or particularly required, by excavators (e.g., large trees of a certain age [3] that contain many dead branches and have accumulated multiple decay-formed cavities [4]).

[2] We detected relative differences in strengths among the relationships between excavators/woodpeckers and secondary cavity-nesters, forest birds, and forest specialists across zoogeographic regions. These differences were most likely to stem from the same variation among species and ecosystem characteristics discussed above (e.g., the distribution and relative importance of cavity-forming agents), and indicate that richness of excavators/woodpeckers may be a better indicator of richness of other cavity and forest-associated species in one region than in another. In addition, the relative strength of relationships between excavators and other forest birds might reflect whether excavators are indicative of the richness of other species due to shared associations with particular forest elements (indirect association), due to their direct facilitation of nest cavities (direct facilitation), or whether both factors play a role. For example, we found that relationships between excavator and forest bird or forest specialist richness in the Afrotropics are relatively strong, whereas the relationship between excavators and secondary cavity-nesters is comparatively weaker. This might indicate that, in the Afrotropics, excavators are especially strong predictors of other groups of forest-associated birds through their mutual habitat requirements, rather than through their direct facilitation of tree-cavities. In contrast, excavators are relatively stronger predictors of secondary cavity-nesters in the Oriental region, which might lead to hypotheses that excavators play a more direct, nest-web structuring role through the provision of cavities in the Oriental region.

Beyond variation in the strength of relationships between regions we also found that
 regressions differed in slopes, both across regions and across the different relationships tested.
 For example, the steep slopes we found for correlations between excavators/woodpeckers and
 other non-cavity-nesting forest birds in the predominantly tropical Neotropical regions
 (Neotropics, Oriental, Afrotropical) contrast with the more shallow slopes in the temperate/boreal
 Palaearctic and Nearctic regions (Palearctic, Neartic). And, within the Neotropical region, the
 relationship between excavators and secondary cavity-nesters showed shallower slopes than that
 between excavators and non-cavity-nesting forest birds. These differences mainly in slopes
 mainly reflect differences in richness between regions and between group of forest birds, with
 relatively steeper slopes representing a relatively larger increase in richness of a particular group
 of birds per increase in the number of excavators. In turn, we may look at macro-scale
 processes (e.g., as outlined by Gaston [487]), such as speciation events, are likely to account for
 some of these differences among zoogeographic regions. First, that drive these differences in
 overall richness of birds [498], including cavity-nesting birds [11], and other possible factors
 (e.g., also of trees as potential substrates for cavities may also differ across regions [49]).
 differs both among and within zoogeographic regions, likely inducing differences in tree-cavity-
 nesting assemblages [33].

However, variation in species composition and associated variation in functional richness
 (i.e., the variety of functional traits varies across spatial and environmental gradients, as has been
 shown, for example, for dietary guilds [44]), is likely to contribute most to spatial variation in
 nest-web assemblages across global scales. Specifically, with regards to nest-webs, species with
 the ability to excavate cavities are not randomly distributed across the world. For example, the
 Australasian region lacks the presence of Piciformes, the Order to which nearly 75% of
 excavators belong [11], whereas speciation processes have led to a relatively high diversity of
 Piciformes in Southeast Asia [50, 51].

The observed spatial variation in the relationships between excavators and other forest
 birds within zoogeographic regions may stem mainly from local differences in habitat and
 vegetation characteristics (e.g., tree species composition and climatic variables [52]) as well as
 spatial variation in tree cavity availability and their formation agents [53], rather than from
 speciation processes. Such differences would, for example, explain how in the Neotropics,
 cavity-nesting assemblages studied in the temperate mountain forests of Chile [54] had a 1:6
 ratio of excavators (4 species) to secondary cavity-nesters (25 species) whereas a subtropical
 Atlantic moist forest [32] supported relatively higher numbers of excavators with a 1:3 ratio of
 excavators (9 species) to secondary cavity-nesters (25 species). An important factor to consider
 here is the potential for regional variation in cavity substrate availability (e.g., large trees) and
 the supply of decay-formed cavities due to differences in the onset and rate of decay processes as
 well as forest management (e.g., [55-57]. For example, retention of large trees can, in many
 forests, provide nesting and roosting opportunities for additional species as both opportunities for
 excavation and processes of decay become more prevalent.

In addition, we can hypothesize that at least some variation at regional scales stems from
 differences in the propensities for re-use of excavated cavities by secondary cavity-nesters. For
 example, there are signs that non-excavating cavity-nesters may experience higher predation risk
 in excavated versus decay-formed cavities, which may induce a preference for decay-formed
 cavities among secondary users if decay-formed cavities are in sufficient supply (e.g., a pattern
 found in some of the few unmanaged temperate forests in the European parts of the Palearctic;

[~~58, 59~~]). In turn, the availability of decay-formed cavities may correspond to the
aforementioned historical or current forest management practices as well as spatial patterns in
climatic factors. For example, proportions of decay-formed cavities used are particularly high in
parts of the world with high amounts of precipitation and more active regimes of fungal growth
(e.g., most of the wet tropics), with both being lower in anthropogenically-disturbed landscapes
than in primary forests [53]. In turn, high availability of decay-formed cavities might allow for a
relatively high richness of secondary cavity-nesters, with fewer species dependent on excavators
for the supply of nest-cavities. For example, even woodpeckers, commonly considered primary
excavators, occasionally use decay-formed cavities in Neotropical temperate rainforests where
tree decay is the key driver of nest-web structure [54].

Finally, we find that relatively higher numbers of secondary cavity-nesters, or cavity-
nesters in general, may occur in regions where many species are habitat generalists that utilize
other nest substrates than tree cavities (e.g., 40% of cavity nesters in Neotropical temperate
forests are facultative users of tree cavities for nesting [54]), as compared to regions where most
cavity nesters are obligate tree-cavity users (e.g., over 85% of cavity nesters in temperate
Neartic forests [59]). Finally, it is worth noting that cavity availability and use are also likely to
be influenced by cavity persistence (longevity), a characteristic that can have considerable spatial
variation due to factors such as land-use patterns (e.g., logging), climatic factors (e.g., rainfall),
and cavity characteristics (e.g., whether they are located in live or dead trees [60–62]):

~~We detected relative differences in strengths among the relationships between~~
~~excavators/woodpeckers and secondary cavity-nesters, forest birds, and forest specialists across~~
~~zoogeographic regions. These differences were most likely to stem from the same variation~~
~~among species and ecosystem characteristics discussed above (e.g., the distribution and relative~~
~~importance of cavity-forming agents), and indicate that richness of excavators/woodpeckers may~~
~~be a better indicator of richness of other cavity and forest-associated species in one region than in~~
~~another. In addition, the relative strength of relationships between excavators and other forest~~
~~birds might reflect whether excavators are indicative of the richness of other species due to~~
~~shared associations with particular forest elements (indirect association), due to their direct~~
~~facilitation of nest cavities (direct facilitation), or whether both factors play a role. For example,~~
~~we found that relationships between excavator and forest bird or forest specialist richness in the~~
~~Afrotropics are relatively strong, whereas the relationship between excavators and secondary~~
~~cavity-nesters is comparatively weaker. This might indicate that, in the Afrotropics, excavators~~
~~are especially strong predictors of other groups of forest-associated birds through their mutual~~
~~habitat requirements, rather than through their direct facilitation of tree cavities. In contrast,~~
~~excavators are relatively stronger predictors of secondary cavity-nesters in the Oriental region,~~
~~which might lead to hypotheses that excavators play a more direct, nest-web structuring role~~
~~through the provision of cavities in the Oriental region.~~

Some secondary cavity-nesters might use cavities created by multiple excavator species,
whereas others predominantly use cavities formed by processes of degradation, damage or insect
activity [31, 32, 39]. In addition, certain excavators might be especially abundant or provide
cavities that are very commonly reused by several species, whereas others are scarce and leave
only a few short-lived cavities that would not be generally available for use by secondary cavity-
nesters. Given such complexities among nest-web interactions, it is important to emphasize that
we present general global patterns in the composition of tree-cavity-using assemblages (i.e.,
communities of linked species that often interact within networks). We lack evidence for direct

causal links between the presence of one species group with another, as well as direct evidence of species interactions at the zoogeographic regional scale.

Around most of the world, we found strong and positive relationships ~~of a universal nature (strong and positive)~~ between excavator and forest bird richness. As such, we deem excavators, ~~of which woodpeckers are an important subset~~, to be useful indicator species for forest bird diversity at multiple-broad spatial scales, as was previously proven at stand- and landscape-scales. They form a small and fairly easy to identify (even by citizen scientists) subset of a larger group of species (forest birds) often studied to understand the effects of anthropogenic disturbances. Moreover, many woodpeckers respond to playbacks, which may therefore become an effective survey tool that allows researchers to fairly easily cover relatively large areas (e.g., [64]). Studies of excavators ~~(or woodpeckers specifically)~~, especially in high biodiversity tropical regions, might provide quick measures of forest quality and biodiversity, which may in turn guide forest management decisions (e.g., improving the retention of large trees). A comprehensive study of all forest birds in some regions like the Amazon or Congo basin (where, notably, there are hotspots of woodpecker richness [65]) would require considerable local expertise and training, which is often difficult to achieve due to social and monetary limitations. Thus, we conclude, excavators have excellent potential as study subjects for conservation monitoring and planning initiatives, in comparative research, and to guide the establishment of region-wide forest conservation strategies, especially in the largely understudied tropical regions of the world.

**Acknowledgments**

We would like to thank Lisa Manne, and Kristina Cockle and two anonymous reviewers for their
critical remarks on drafts of this article.

**Funding**

This research was not externally funded.

**References**

- Allan JR, Venter O, Maxwell S, Bertzky B, Jones K, Shi Y, Watson JE. 2017 Recent increases in human
pressure and forest loss threaten many Natural World Heritage Sites. *Biological conservation*. **206**, 47-
55.
- Hansen MC, Stehman SV, Potapov PV. 2010 Quantification of global gross forest cover loss.
*Proceedings of the National Academy of Sciences*. **107**, 8650-8655.
- Barlow J, Lennox GD, Ferreira J, Berenguer E, Lees AC, Mac Nally R, Thomson JR, de Barros Ferraz SF,
Louzada J, Oliveira VHF. 2016 Anthropogenic disturbance in tropical forests can double biodiversity loss
from deforestation. *Nature*. **535**, 144.
- Gaston KJ, Blackburn TM, Goldewijk KK. 2003 Habitat conversion and global avian biodiversity loss.
*Proceedings of the Royal Society of London. Series B: Biological Sciences*. **270**, 1293-1300.
- Betts MG, Wolf C, Pfeifer M, Banks-Leite C, Arroyo-Rodríguez V, Ribeiro DB, Barlow J, Eigenbrod F,
Faria D, Fletcher RJ. 2019 Extinction filters mediate the global effects of habitat fragmentation on
animals. *Science*. **366**, 1236-1239.
- Carignan V, Villard M-A. 2002 Selecting indicator species to monitor ecological integrity: a review.
*Environmental monitoring and assessment*. **78**, 45-61.
- Canterbury GE, Martin TE, Petit DR, Petit LJ, Bradford DF. 2000 Bird communities and habitat as
ecological indicators of forest condition in regional monitoring. *Conservation Biology*. **14**, 544-558.

Venier LA, Pearce JL. 2004 Birds as indicators of sustainable forest management. *The Forestry*
*Chronicle*. **80**, 61-66.
- Martin K, Ibarra JT, Drever M. 2015 Avian surrogates in terrestrial ecosystems: theory and practice. In
*Indicators and surrogates of biodiversity and environmental change* (eds D Lindenmayer, P Barton, J
Pierson), pp 33-44. Melbourne & London: CSIRO Publishing & CRC Press.
- Ibarra JT, Martin M, Cockle KL, Martin K. 2017 Maintaining ecosystem resilience: functional
responses of tree cavity nesters to logging in temperate forests of the Americas. *Scientific reports*. **7**,
4467.
- van der Hoek Y, Gaona GV, Martin K. 2017 The diversity, distribution and conservation status of the
tree-cavity-nesting birds of the world. *Diversity and Distributions*. **23**, 1120-1131.
- McClelland BR, McClelland PT. 1999 Pileated woodpecker nest and roost trees in Montana: links with
old-growth and forest "health". *Wildlife Society Bulletin*. 846-857.
- Stachura-Skierczyńska K, Kosiński Z. 2016 Do factors describing forest naturalness predict the
occurrence and abundance of middle spotted woodpecker in different forest landscapes? *Ecological*
*Indicators*. **60**, 832-844.
- Marchetti M. 2005 *Monitoring and indicators of forest biodiversity in Europe: from ideas to*
*operationality*. Joensuu, Finland: European Forest Institute.
- Roberge J-M, Angelstam P. 2006 Indicator species among resident forest birds—a cross-regional
evaluation in northern Europe. *Biological Conservation*. **130**, 134-147.
- Virkkala R. 2006 Why study woodpeckers? The significance of woodpeckers in forest ecosystems.
*Annales Zoologici Fennici*. 82-85.
- Drever MC, Aitken KE, Norris AR, Martin K. 2008 Woodpeckers as reliable indicators of bird richness,
forest health and harvest. *Biological conservation*. **141**, 624-634.
- Drever MC, Martin K. 2010 Response of woodpeckers to changes in forest health and harvest:
Implications for conservation of avian biodiversity. *Forest Ecology and Management*. **259**, 958-966.
- Mikusiński G, Gromadzki M, Chylarecki P. 2001 Woodpeckers as indicators of forest bird diversity.
*Conservation Biology*. **15**, 208-217.
- Angelstam PK, Bütler R, Lazdinis M, Mikusiński G, Roberge J-M. 2003. Habitat thresholds for focal
species at multiple scales and forest biodiversity conservation—dead wood as an example. *Annales*
*Zoologici Fennici*. 473-482.
- Nilsson SG, Hedin J, Niklasson M. 2001 Biodiversity and its assessment in boreal and nemoral forests.
*Scandinavian Journal of Forest Research*. **16**, 10-26.
- Hess GR, Bartel RA, Leidner AK, Rosenfeld KM, Rubino MJ, Snider SB, Ricketts TH. 2006 Effectiveness
of biodiversity indicators varies with extent, grain, and region. *Biological Conservation*. **132**, 448-457.
- Collen B, Ram M, Zamin T, McRae L. 2008 The tropical biodiversity data gap: addressing disparity in
global monitoring. *Tropical Conservation Science*. **1**, 75-88.
- Ilsøe SK, Kissling WD, Fjeldså J, Sandel B, Svenning JC. 2017 Global variation in woodpecker species
richness shaped by tree availability. *Journal of Biogeography*. **44**, 1824-1835.
- Bell D, Hjältén J, Nilsson C, Jørgensen D, Johansson T. 2015 Forest restoration to attract a putative
umbrella species, the white-backed woodpecker, benefited saproxylic beetles. *Ecosphere*. **6**, 1-14.
- Gatica-Saavedra P, Echeverría C, Nelson CR. 2017 Ecological indicators for assessing ecological
success of forest restoration: a world review. *Restoration ecology*. **25**, 850-857.
- Hoyt JS, Hannon SJ. 2002 Habitat associations of black-backed and three-toed woodpeckers in the
boreal forest of Alberta. *Canadian Journal of Forest Research*. **32**, 1881-1888.
- Tikkanen O-P, Martikainen P, Hyvärinen E, Junninen K, Kouki J. 2006. Red-listed boreal forest species
of Finland: associations with forest structure, tree species, and decaying wood. *Annales Zoologici*
*Fennici*. 373-383.

- Bütler R, Angelstam P, Ekelund P, Schlaepfer R. 2004 Dead wood threshold values for the three-toed
woodpecker presence in boreal and sub-Alpine forest. *Biological Conservation*. **119**, 305-318.
- McLaren MA, Thompson ID, Baker JA. 1998 Selection of vertebrate wildlife indicators for monitoring
sustainable forest management in Ontario. *The Forestry Chronicle*. **74**, 241-248.
- Martin K, Eadie JM. 1999 Nest webs: A community-wide approach to the management and
conservation of cavity-nesting forest birds. *Forest Ecology and Management*. **115**, 243-257.
- Cockle KL, Martin K, Robledo G. 2012 Linking fungi, trees, and hole-using birds in a Neotropical tree-
cavity network: Pathways of cavity production and implications for conservation. *Forest Ecology and*
*Management*. **264**, 210-219.
- Cockle KL, Martin K, Wesolowski T. 2011 Woodpeckers, decay, and the future of cavity-nesting
vertebrate communities worldwide. *Frontiers in Ecology and the Environment*. **9**, 377-382.
- Tylianakis JM, Klein AM, Lozada T, Tscharrntke T. 2006 Spatial scale of observation affects α , β and γ
diversity of cavity-nesting bees and wasps across a tropical land-use gradient. *Journal of Biogeography*.
**33**, 1295-1304.
- 35 Powell S, Costa AN, Lopes CT, Vasconcelos HL. 2011 Canopy connectivity and the availability of
diverse nesting resources affect species coexistence in arboreal ants. *Journal of Animal Ecology*. **80**, 352-
360.
- 36 Jankowiak R, Ciach M, Bilański P, Linnakoski R. 2019. Diversity of wood-inhabiting fungi in
woodpecker nest cavities in southern Poland. *Acta Mycologica*. **54**, 1126. doi:10.5586/am.1126
- 37 Jusino MA, Lindner DL, Banik MT, Rose KR, Walters JR. 2016 Experimental evidence of a symbiosis
between red-cockaded woodpeckers and fungi. *Proceedings of the Royal Society B: Biological Sciences*.
283, 20160106.
- 38 Bull EL, Jackson JA., 1995 Pileated Woodpecker (*Dryocopus pileatus*). In: *The birds of North America*
(eds A Poole, F Gill). Washington DC: The Academy of Natural Sciences & The American Ornithologists'
Union.
- 39 Montellano MGN, Blendinger PG, Macchi L. 2013 Sap consumption by the White-fronted
Woodpecker and its role in avian assemblage structure in dry forests. *The Condor*. **115**, 93-101.
- 40 Ruggera RA, Schaaf AA, Vivanco CG, Politi N, Rivera LO. 2016 Exploring nest webs in more detail to
improve forest management. *Forest Ecology and Management*. **372**, 93-100.
- 41 Canterbury GE, Martin TE, Petit DR, Petit LJ, Bradford DF. 2000. Bird communities and habitat as
ecological indicators of forest condition in regional monitoring. *Conservation Biology*. **14**(2), 544-558.
- 42 Betts MG, Wolf C, Ripple WJ, Phalan B, Millers KA, Duarte A, Butchart SH, Levi T. 2017 Global forest
loss disproportionately erodes biodiversity in intact landscapes. *Nature*. **547**, 441-444.
- 43 BirdLife International and NatureServe 2015 *Bird species distribution maps of the world*. Cambridge
and Arlington: BirdLife International and NatureServe.
- 44 Storch D, Davies RG, Zajíček S, Orme CDL, Olson V, Thomas GH, Ding TS, Rasmussen PC, Ridgely RS,
Bennett PM. 2006 Energy, range dynamics and global species richness patterns: reconciling mid-domain
effects and environmental determinants of avian diversity. *Ecology Letters*. **9**, 1308-1320.
- 45 Pinheiro J, Bates D, DebRoy S, Sarkar D, Heisterkamp S, Van Willigen B. 2017 Package 'nlme'. *Linear*
and *Nonlinear Mixed Effects Models, version 3-1*. Available from [https://cran.r-](https://cran.r-project.org/web/packages/nlme/nlme.pdf)
project.org/web/packages/nlme/nlme.pdf (accessed January 15, 2018).
- 46 Kissling WD, Sekercioglu CH, Jetz W. 2012 Bird dietary guild richness across latitudes, environments
and biogeographic regions. *Global Ecology and Biogeography*. **21**, 328-340.
- 47 Hunter M, Westgate M, Barton P, Calhoun A, Pierson J, Tulloch A, Beger M, Branquinho C, Caro T,
Gross J. 2016 Two roles for ecological surrogacy: Indicator surrogates and management surrogates.
*Ecological Indicators*. **63**, 121-125.

48 Remm J, Löhmus A. 2011 Tree cavities in forests—the broad distribution pattern of a keystone
structure for biodiversity. *Forest Ecology and Management*. **262**, 579-585.
49 Altamirano TA, Ibarra JT, Martin K, Bonacic C. 2017 The conservation value of tree decay processes as
a key driver structuring tree cavity nest webs in South American temperate rainforests. *Biodiversity and*
Conservation. **26**, 2453-2472.
50 Styring AR, bin Hussin MZ. 2004. Effects of logging on woodpeckers in a Malaysian rain forest: the
relationship between resource availability and woodpecker abundance. *Journal of Tropical Ecology*.
20(5), 495-504.
51 Fischer J, Stott J, Law BS. 2010. The disproportionate value of scattered trees. *Biological*
Conservation. **143**(6), 1564-1567.
52 Lindenmayer DB, Blanchard W, McBurney L, Blair D, Banks S, Likens GE, Franklin JF, Laurance WF,
Stein JA, Gibbons P. 2012. Interacting factors driving a major loss of large trees with cavities in a forest
ecosystem. *PLoS One*. **7**(10).
53 Gibbons P, Lindenmayer DB, Barry SC, Tanton MT. 2002. Hollow selection by vertebrate fauna in
forests of southeastern Australia and implications for forest management. *Biological Conservation*. **103**,
1-12.
54 Gaston KJ. 2000 Global patterns in biodiversity. *Nature*. **405**, 220.
55 Blackburn TM, Gaston KJ. 1996 Spatial patterns in the species richness of birds in the New World.
Ecography. **19**, 369-376.
56 Currie DJ, Paquin V. 1987 Large-scale biogeographical patterns of species richness of trees. *Nature*.
329, 326-327.
57 Baumgardt JA, Sauder JD, Nicholson KL. 2014. Occupancy modeling of woodpeckers: maximizing
detections for multiple species with multiple spatial scales. *Journal of Fish and Wildlife Management*.
5(2), 198-207.
58 Mikusiński G. 2006 Woodpeckers: distribution, conservation, and research in a global perspective.
Annales Zoologici Fennici. 86-95.
59 Vergara-Tabares DL, Lammertink M, Verga EG, Schaaf AA, Nori J. 2018. Gone with the forest:
Assessing global woodpecker conservation from land use patterns. *Diversity and Distributions*. **24**, 640-
51.
35 Powell S, Costa AN, Lopes CT, Vasconcelos HL. 2011 Canopy connectivity and the availability of
diverse nesting resources affect species coexistence in arboreal ants. *Journal of Animal Ecology*. **80**, 352-
360.
38 Jusino MA, Lindner DL, Banik MT, Rose KR, Walters JR. 2016 Experimental evidence of a symbiosis
between red-cockaded woodpeckers and fungi. *Proceedings of the Royal Society B: Biological Sciences*.
283, 20160106.
36 Bull EL, Jackson JA., 1995 Pileated Woodpecker (*Dryocopus pileatus*). In: *The birds of North America*
(eds A Poole, F Gill). Washington DC: The Academy of Natural Sciences & The American Ornithologists'
Union.
37 Montellano MGN, Blendinger PG, Macchi L. 2013 Sap consumption by the White-fronted
Woodpecker and its role in avian assemblage structure in dry forests. *The Condor*. **115**, 93-101.
38 Jusino MA, Lindner DL, Banik MT, Rose KR, Walters JR. 2016 Experimental evidence of a symbiosis
between red-cockaded woodpeckers and fungi. *Proceedings of the Royal Society B: Biological Sciences*.
283, 20160106.
39 Ruggera RA, Schaaf AA, Vivanco CG, Politi N, Rivera LO. 2016 Exploring nest webs in more detail to
improve forest management. *Forest Ecology and Management*. **372**, 93-100.
40 Betts MG, Wolf C, Ripple WJ, Phalan B, Millers KA, Duarte A, Butchart SH, Levi T. 2017 Global forest
loss disproportionately erodes biodiversity in intact landscapes. *Nature*. **547**, 441-444.

- BirdLife International and NatureServe. 2015 *Bird species distribution maps of the world*. Cambridge
and Arlington: BirdLife International and NatureServe.
- Storch D, Davies RG, Zajíček S, Orme CDL, Olson V, Thomas GH, Ding TS, Rasmussen PC, Ridgely RS,
Bennett PM. 2006 Energy, range dynamics and global species richness patterns: reconciling mid-domain
effects and environmental determinants of avian diversity. *Ecology Letters*. **9**, 1308-1320.
- Pinheiro J, Bates D, DebRoy S, Sarkar D, Heisterkamp S, Van Willigen B. 2017 Package 'nlme'. *Linear
and Nonlinear Mixed Effects Models, version 3.1*. Available from [https://cran.r-](https://cran.r-project.org/web/packages/nlme/nlme.pdf)
[project.org/web/packages/nlme/nlme.pdf](https://cran.r-project.org/web/packages/nlme/nlme.pdf) (accessed January 15, 2018).
- Kissling WD, Sekercioglu CH, Jetz W. 2012 Bird dietary guild richness across latitudes, environments
and biogeographic regions. *Global Ecology and Biogeography*. **21**, 328-340.
- Hunter M, Westgate M, Barton P, Calhoun A, Pierson J, Tulloch A, Beger M, Branquinho C, Caro T,
Gross J. 2016 Two roles for ecological surrogacy: Indicator surrogates and management surrogates.
*Ecological Indicators*. **63**, 121-125.
- Stokland JN, Siitonen J, Jonsson BG. 2012 *Biodiversity in dead wood*. Cambridge, UK: Cambridge
University Press.
- Gaston KJ. 2000 Global patterns in biodiversity. *Nature*. **405**, 220.
- Blackburn TM, Gaston KJ. 1996 Spatial patterns in the species richness of birds in the New World.
*Ecography*. **19**, 369-376.
- Currie DJ, Paquin V. 1987 Large-scale biogeographical patterns of species richness of trees. *Nature*.
**329**, 326-327.
- Benz BW, Robbins MB, Peterson AT. 2006 Evolutionary history of woodpeckers and allies (Aves:
Picidae): placing key taxa on the phylogenetic tree. *Molecular phylogenetics and evolution*. **40**, 389-399.
- Mikusiński G. 2006 Woodpeckers: distribution, conservation, and research in a global perspective.
*Annales Zoologici Fennici*. 86-95.
- Huston MA. 1999 Local processes and regional patterns: appropriate scales for understanding
variation in the diversity of plants and animals. *Oikos*. **86**, 393-401.
- Remm J, Löhmus A. 2011 Tree cavities in forests—the broad distribution pattern of a keystone
structure for biodiversity. *Forest Ecology and Management*. **262**, 579-585.
- Altamirano TA, Ibarra JT, Martin K, Bonacic C. 2017 The conservation value of tree decay processes as
a key driver structuring tree cavity nest webs in South American temperate rainforests. *Biodiversity and
Conservation*. **26**, 2453-2472.
- Andersson J, Domingo-Gómez E, Michon S, Roberge J-M. 2018 Tree cavity densities and
characteristics in managed and unmanaged Swedish boreal forest. *Scandinavian journal of forest
research*. **33**, 233-244.
- Goodburn JM, Lorimer CG. 1998 Cavity trees and coarse woody debris in old-growth and managed
northern hardwood forests in Wisconsin and Michigan. *Canadian Journal of Forest Research*. **28**, 427-
438.
- Gutzat F, Dormann CF. 2018 Decaying trees improve nesting opportunities for cavity-nesting birds in
temperate and boreal forests: A meta-analysis and implications for retention forestry. *Ecology and
evolution*. **8**, 8616-8626.
- Wesolowski T, Tomialojc L. 2005 Nest sites, nest depredation, and productivity of avian broods in a
primeval temperate forest: do the generalisations hold? *Journal of Avian Biology*. **36**, 361-367.
- Wesolowski T, Martin K. 2018 Tree Holes and Hole-Nesting Birds in European and North American
Forests. In *Ecology and Conservation of Forest Birds* (eds G Mikusinski, J-M Roberge, RJ Fuller), pp 79-
134. Cambridge, UK: Cambridge University Press.
- Edworthy AB, Wiebe KL, Martin K. 2012 Survival analysis of a critical resource for cavity-nesting
communities: patterns of tree cavity longevity. *Ecological Applications*. **22**, 1733-1742.

Cockle KL, Martin K, Bodrati A. 2017 Persistence and loss of tree cavities used by birds in the subtropical Atlantic Forest. *Forest ecology and management*. **384**, 200–207.
 Wesolowski T. 2011 “Lifespan” of woodpecker-made holes in a primeval temperate forest: A thirty year study. *Forest ecology and management*. **262**, 1846–1852.

Table 1. Species richness for forest-associated bird groups found in each of six zoogeographic regions. In parenthesis the maximum number of species of each group found in a single 10×10 km grid cell in each region.

Zoographic region	Excavators	Secondary cavity-nesters	Forest specialists¹	Forest generalists¹	All forest birds (specialists + generalists)¹
Nearctic	42 (16)	111 (41)	93 (36)	365 (111)	458 (140)
Palaearctic	67 (23)	181 (58)	155 (65)	725 (221)	880 (276)
Oriental	120 (35)	241 (67)	463 (79)	958 (238)	1421 (300)
Neotropical	173(36)	373 (114)	1078 (200)	1764 (277)	2842 (436)
Afrotropical	78 (33)	218 (68)	220 (60)	911 (287)	1131 (331)
Australasia	14 (7)	178 (70)	442 (67)	1014 (190)	1456 (226)

¹Note: these numbers do not include secondary cavity-nesters.

Table 21. Results of generalized least square models of possible correlations between (1) excavator and secondary cavity-nester richness, (2) excavator and forest specialist-bird richness, (3) excavator and forest generalistspecialist richness, (4) excavator and forest bird (specialists and generalists) richness. Each result reflects the mean outcome of 1000 models d with 1000 randomly selected grid cell data points that include an exponential spatial covariance structure. The mean Spearman rho was calculated over the fit of each of 1000 models to a test data set.

Zoogeographic region	Correlation	Mean Slope (Min. – Max.; SD)	% of  models with $p < 0.05$	Mean Spearman rho
Nearctic	1	0.8 (0.4 - 1.3; 0.1)	60	0.73
	2	1.043 (0.733 - 5.913; 0.14)	9763	0.7184
	3	3.114 (2.012 - 1.645; 0.41)	8971	0.7671
	4	4.2 (2.7 - 5.7; 0.5)	94	0.81

Palearctic	1	1.1 (0.7 - 1.4; 0.1)	84	0.78
	2	0.64.9 (0.34.0 - 1.16.7; 0.34)	8373	0.6885
	3	2.91.1 (2.10.8 - 1.74.1; 0.31)	823	0.8079
	4	3.5 (2.6 - 5.4; 0.4)	88	0.80
Oriental	1	0.7 (0.6 - 0.9; 0.0)	100	0.88
	2	1.15.7 (0.94.9 - 1.36.6; 0.13)	100	0.7286
	3	3.51.6 (3.01.3 - 4.01.9; 0.21)	100	0.8677
	4	4.6 (3.9 - 5.5; 0.2)	100	0.83
Neotropical	1	2.4 (1.9 - 3.0; 0.2)	83	0.95
	2	3.510.8 (9.1 - 14.0; 0.27)	732	0.957
	3	5.24.2 (3.93.4 - 6.75.2; 0.52)	772	0.916
	4	8.8 (7.1 - 11.0; 0.6)	84	0.96
Afrotropical	1	1.1 (0.7 - 1.9; 0.2)	76	0.79
	2	0.66.8 (0.35.0 - 1.19.8; 0.19)	783	0.8394
	3	5.80.9 (3.70.6 - 8.11.5; 0.91)	8479	0.9388
	4	6.4 (4.3 - 9.2; 0.9)	87	0.93
Australasian	1	1.8 (1.4 - 2.2; 0.1)	88	-0.14
	2	1.77.5 (0.92.3 - 2.511.2; 0.31.3)	100	0.7331
	3	9.21.0 (7.2-0.1 - 11.22.0; 0.63)	967	0.1969
	4	11.0 (8.3 - 13.5; 0.8)	99	0.50

**Table 2. Results of generalized least square models of possible correlations between (1)**
**woodpecker and secondary cavity-nester richness, (2) woodpecker and forest bird richness,**
**and (3) woodpecker and forest specialist richness.** Each result reflects the mean outcome of
1000 models build with 1000 randomly selected grid cell data points that include an exponential
spatial covariance structure. The mean Spearman rho was calculated over the fit of each of 1000
models to a test data set.

Zoogeographic region	Correlation	Mean Slope (Min.—Max.; SD)	% of models with $P < 0.05$	Mean Spearman rho
Nearetic	1	1.0 (0.5—1.6; 0.2)	61	0.70
	2	5.3 (3.7—8.0; 0.7)	67	0.79
	3	1.7 (1.1—2.2; 0.1)	70	0.63
Palearctic	1	1.2 (0.9—2.3; 0.1)	81	0.79
	2	5.6 (4.3—8.1; 0.6)	73	0.83
	3	1.3 (0.9—2.0; 0.2)	82	0.73
Oriental	1	1.0 (0.8—1.3; 0.1)	100	0.88
	2	7.9 (6.6—9.8; 0.5)	100	0.88
	3	2.2 (1.7—2.6; 0.1)	100	0.80
Neotropical	1	3.3 (2.4—4.9; 0.4)	68	0.94
	2	14.3 (10.8—20.5; 1.6)	67	0.93
	3	5.2 (3.9—6.6; 0.5)	73	0.88
Afrotropical	1	1.6 (0.9—3.5; 0.4)	72	0.74
	2	9.9 (6.4—19.6; 1.9)	69	0.86
	3	1.3 (0.7—3.2; 0.3)	80	0.81
Australasian	1	1.9 (-0.1—4.4; 0.6)	81	-0.33
	2	8.6 (1.2—21.8; 2.4)	75	-0.30
	3	3.9 (-0.2—9.0; 1.2)	96	-0.08

**Figure legends**

**Figure 1. Global map of the relative richness of tree cavity excavators versus secondary**
**cavity-nesting birds (expressed as the ratio excavator / secondary cavity-nester) in 10x10**
**km grid cells.** Low values (light colors) indicate relatively low numbers of excavators as
compared to secondary cavity-nesters ~~(or, inversely, relatively high numbers of secondary~~
~~cavity-nesters)~~ while high values (dark colors) indicate a relatively higher proportion of
excavators to secondary cavity-nesters.

**Figure 2. Scatter plots of the correlations between the richness (number of species) of**
**excavators and secondary cavity-nesters all forest birds (upper row), all-forest specialists**
**(second middle row), forest generalists (third row) and all non-cavity-nesting forest birds**
**(specialists and generalists; excavating cavity-nesters (bottom row) in six zoogeographic**
**regions.** Points represent species richness values in 10x10 km grid cells, and the solid line
represents the fit of a linear regression model, created using all grid cell values. Note that nearly
all secondary cavity-nesters are also forest-associated species, but were not included as forest
specialists or generalists.

Figure 1.

**Figure 2.**

60926

1. Styring, A.R. and M.Z. bin Hussin, *Effects of logging on woodpeckers in a Malaysian rain forest: the relationship between resource availability and woodpecker abundance*. Journal of Tropical Ecology, 2004. **20**(5): p. 495-504.
2. Fischer, J., J. Stott, and B.S. Law, *The disproportionate value of scattered trees*. Biological Conservation, 2010. **143**(6): p. 1564-1567.
3. Lindenmayer, D.B., et al., *Interacting factors driving a major loss of large trees with cavities in a forest ecosystem*. PLoS One, 2012. **7**(10).
4. Gibbons, P., et al., *Hollow selection by vertebrate fauna in forests of southeastern Australia and implications for forest management*. Biological Conservation, 2002. **103**(1): p. 1-12.

[revised manuscript text omitted]

Excavators are not only reliable indicators of other forest birds occurrence by means of
their mutual associations with forest characteristics, but correlations between both groups of
birds are also partially determined by species interactions. Taken together, these ‘nest-webs’ [32]
of interacting species are structured by the availability and acquisition of tree-cavities formed by
two major processes: ‘fungal formation’, in which fungi decompose wood over an extended
period, or animal excavation [33, 34]. Since most excavators—at least the majority of
woodpeckers—excavate a new nest cavity each year, this mode of cavity creation steadily
provides a large number of potential sheltered roosting and nesting sites available to secondary-

139 users: both for vertebrates [34] and invertebrates [35, 36], as well as for maintenance of diverse
fungal assemblages [37]. As a result, cavity formation by excavation is an important source of
nest holes for species in many regions of the world, especially in North America [34]. However,
decay-formed cavities have been shown to be more important nest sites in other regions,
especially in the tropics [33]. Thus, it is very likely that the relative importance of both processes
of cavity formation—and thus the strength of the relationship between excavator richness and
secondary cavity-nester richness—varies globally [34]. In addition, excavators can aid in the
dispersal of fungi, which in turn enhances processes of wood softening and the formation of
decay-formed cavities [38]. It is worth noting that excavators play additional roles (beyond the
direct provision of cavities) that may facilitate other species that are not necessarily cavity-
nesters. For example, as excavators perforate the bark of trees they may expose insects or sap for
other species to forage on [39, 40].

Excavators have been shown to be reliable indicators of forest-bird diversity and richness
at the stand or landscape-levels [9, 19]. It is unclear if this relationship scales up across larger
regions or holds across the globe given indications that interactions between cavity-nesting birds
may vary spatially (e.g., [41]) as well as the notion that certain dominant groups of excavators
(woodpeckers, in particular) vary in richness across zoogeographic regions [11]. The aim of the
present study is to investigate whether there are consistent patterns in the relationships between
avian tree-cavity excavators (hereafter: excavators) and species richness of non-excavating tree-
cavity nesting birds (hereafter: secondary cavity-nesters), or non-cavity nesting forest birds
(forest specialist and generalists species), across the globe. If previously observed relationships
between cavity-excavators and other forest birds scale up from local forest stands [17] or
ecosystems [19] to zoogeographic or global levels, then this strengthens the potential value of
tree cavity-excavators as both indicators and management surrogates.

Given the important potential indicator role of excavators in forest ecosystems across the
world, we predict a global tendency for richness of secondary cavity-nesters, forest specialists,
forest generalists and overall forest birds (specialists + generalists) to increase in correlation with
the number of excavators present in the ecosystem, regardless of the ecosystem or location.
However, we also predict that the nature (e.g., the strength and slope of correlations) of the
relationships between excavators and richness of other forest birds will vary across
zoogeographic regions, reflecting differences in forest characteristics (e.g., deciduous vs.
coniferous forests), forest management practices (e.g., forests being more or less intensively
managed), and bird communities (e.g., the relative richness of different groups of forest-
associated birds), some aspects of which are previously discussed by e.g., Cockle et al. [34]. In
particular, we predict that correlations between excavators and secondary cavity-nesters would
be particularly strong in regions with relatively low overall avian diversity and slow processes of
decay-related tree-cavity formation. In such regions predominantly temperate and boreal regions,
particularly the Nearctic and Palearctic, relationships between excavators and secondary cavity-
nesters would be driven by mutual associations with particular forest elements but enhanced by
direct interactions through the provision of nest cavities, as a lack of available decay-formed
cavities may increase the dependence of secondary-cavity nesters on excavated cavities. In
contrast, for regions that largely span the tropics, such as the Neotropics, Afrotropics, and
Oriental regions, we predict weaker, though still positive, correlations between excavators and
secondary cavity-nesters as these relationships would be driven mainly by mutual associations
with particular forest elements and less by direct interactions through the provision of nest
cavities—decay-formed cavities being more prevalent here. Similarly, we also predict that

relationships between excavators and secondary cavity-nesters are stronger than those between
excavators and non-cavity-nesting forest birds (specialists or generalists) in the Nearctic and
Palearctic, but that there is no such clear pattern of differences in strength of correlations for the
tropical regions. For Australasia, where the largest group of excavators (woodpeckers) is notably
absent, we predict relationships between the present excavators, often weaker excavators that
require softer or rotting wood (see [11] for species), and other groups of birds to be relatively
weak and to be based mainly on certain shared habitat preferences. Finally, for all regions, we
predict that relationships between excavators and secondary cavity-nesters and between
excavators and forest specialists are stronger than those between excavators and forest
generalists, as the latter group includes species that are unlikely to have either direct interactions
or mutually shared habitat preferences with excavators. In fact, many generalist forest birds may
have traits (e.g., ground-nesting behaviour and a preference for early-successional forests with
small trees [41]) that can lead to habitat requirements opposite from those required by
excavators.

**Material and Methods**

We used published global distribution maps of tree-cavity nesting birds [11] and forest birds [42]
to extract richness estimates per 10×10 km grid cell (Table 1). Maps for both tree-cavity nesters
and forest birds were based on the same species range maps provided by BirdLife International
and NatureServe [43]. We selected and classified focal bird species following [11, 42] as 1)
excavators (species that are known to often or always excavate their own nesting cavities), 2)
secondary cavity-nesters: birds which are known to nest in tree cavity but which never or rarely
excavate their own cavity, 3) forest specialists: birds that exclusively use forest habitat, and 4)
forest birds: birds that use forest habitat but also at least one other type of habitat. We note that
some species were counted in more than one category (e.g., a species that is a ‘forest specialist’
is per definition also counted as a ‘forest bird’). However, excavators were never counted in any
other category.

We used only range polygons where a species’ presence was classified as ‘Extant’ or
‘Probably extant’, and assigned ‘presence’ to all grid cells that overlapped with range polygons.
Next, we created generalized least squares (gls) models to assess the linear relationships between
excavators and other categories of forest-associated birds. To take biogeographical differences
into account, we proceeded to separate trends across the globe and created gls models for the six
zoogeographic regions separately: Nearctic, Palearctic, Neotropical, Afrotropical, Oriental, and
Australasia. We did not restrict our analyses to forested regions, wanting to avoid making
subjective decisions on the classification of a grid cell to a specific vegetation/habitat type. Yet,
to ensure we looked for possible relationships between these forest-associated species in regions
they actually occur, we removed cells with zero excavators and cells where the numbers of
secondary cavity-nesters, forest specialists, or all forest birds were lower than that of excavators
from further analyses.

Our data were spatially structured (i.e., high and low richness of birds in different
categories (excavators etc.) was spatially clustered) across and within zoogeographic regions, as
demonstrated by the global distribution of the relative number of cavity excavators to non-
excavating cavity-nesting birds (expressed as the ratio of excavator / non-excavating cavity-
nester; Fig. 1). Thus, we followed similar protocols to adjust for spatial autocorrelation as per
Storch et al. [44]. This implied the use of gls models with excavator richness as a predictor
variable, the richness of one of the categories of forest-associated birds as a response variable,

and a variable exponential spatial covariance structure, as these models had the best fit in
preliminary tests, which reduced—rather than eliminated—the influence of spatial
autocorrelation on our models. In these tests, exponential gls models consistently had the lowest
Akaike’s Information Criterion (AIC) value compared to a set of models, including an ordinary
least squares model with no spatial structure, and gls models with Gaussian, linear, rational
quadratics and spherical spatial covariance structures.

Even though gls models work well with large datasets, we found that gls models fitted on
all data failed to converge (using the R package “nlme” [45]). To circumvent this computational
problem, we first grouped grid cells by zoogeographic region. We thereafter ran 1000 models on
bootstrap permutations of 1000 randomly selected grid cells (see [24, 46]). Of these 1000 model
iterations, we subsequently calculated the mean (\pm SD), minimum and maximum coefficients of
the regression slope. As gls models do not provide a measure of the strength of relationships, we
proceeded with a different measure of the strength of each correlation: fitting each exponential
gls model to a different, non-overlapping, test data set of 1000 randomly selected grid cells. We
then proceeded to calculate the Spearman rho coefficient of the correlation between predicted
and observed data, for all 1000 models per zoogeographic region. Finally, we calculated the
mean Spearman rho of all significant models per zoogeographic region. We repeated all of these
analyses for woodpeckers as a subset of the excavators, with the notable exception of analyses
for Australasia where woodpeckers are absent. All data, R scripts, and initial results needed to
reproduce our analyses presented here are deposited online (Appendix S1).

**Results**

Using gls modelling to assess the global patterns for the relationships between excavator richness
and that of secondary cavity-nesters, forest specialists, and all forest birds, we found positive
relationships between all pairs of comparisons. However, these relationships differed in relative
strength and characteristics (e.g., steepness of regression slope) depending on the group of birds
and zoogeographic region analysed (Fig. 2, Table 2). We found stronger relationships (higher
Spearman rho) in the predominantly tropical Afrotropical, Oriental, and Neotropical regions than
in the comparably more temperate/boreal Nearctic and Palearctic regions, and particularly weak
relationships in Australasia.

The relationships between excavators and the other three groups of cavity-nesting and
forest-associated birds were generally similar in nature (positive), without clear evidence for our
prediction that the relationship between excavators and other cavity-nesters would be stronger
than that between excavators and larger groups of forest-associated birds that include non-cavity-
nesters (forest specialists or all forest birds). Instead, we found that differences in the relative
strength of these relationships (between excavators and the three groups of forest-associated
birds) differed largely across zoogeographic regions. For example, excavator richness was a
particularly strong predictor of secondary cavity-nester richness and of all forest birds in the
Oriental region ($\rho = 0.88$), and slightly less so for forest specialists ($\rho = 0.72$), whereas
excavator richness in the Afrotropics showed a strong relationship with forest specialists ($\rho =$
0.83), forest generalists ($\rho = 0.91$) and all forest birds taken together ($\rho = 0.93$), but a
somewhat weaker association with secondary cavity-nesters ($\rho = 0.79$). Finally, we note that
excavators in Australasia (which include no woodpeckers) showed a moderately strong ($\rho =$
0.73) and significant ($p < 0.05$ in 100 percent of model iterations) positive relationship with
forest specialists, and none with secondary cavity-nesters or more generalist forest birds.

We repeated all analyses for woodpeckers as a subset of excavators and found nearly
identical results. We list results of these analyses in supplementary Table S1.

**Discussion**

We found strong positive relationships between excavators and secondary cavity-nesters, forest
specialists, forest generalists and overall forest birds at all zoogeographic scales, although these
relationships differed in both strength and characteristics (regression slopes) across
zoogeographic regions. Moreover, we found that it depended on the region whether relationships
between excavators and secondary cavity-nesters were stronger or weaker than those between
excavators and non-cavity-nesting forest birds, but that relationships between excavators and
non-cavity-nesting forest birds were particularly strong in most regions (e.g., the relationship
between excavators and all forest birds was the stronger than that between excavators and
secondary cavity-nesters in the Nearctic, Palearctic, Neotropical, and Afrotropical regions). This
indicates that the influence of direct facilitation of nest-cavities is likely less important than the
indirect associations with forest elements and shared habitat preferences in shaping these
relationships, but that the relative importance of both depends on the region under consideration.
Our results imply that excavators and woodpeckers (a predominant subset of excavators) hold
potential as indicator and management surrogates across most of the world [47], with the
possible exception of the Australasian region, but also that we need to take differences among
zoogeographic regions into account (e.g., in the composition of local and regional nest-webs with
regards to the relative numbers of excavators and secondary cavity-nesters as seen in Fig. 1) if
we are to adopt these as surrogates for conservation or management.

We did not find support for the prediction that relationships between excavators and
secondary-cavity nesters are relatively stronger in temperate and boreal regions (Nearctic and
Palearctic), which would reflect the additional importance of cavity facilitation in these regions
as compared to more tropical regions. In general, excavated tree cavities are less likely to be used
by secondary-cavity nesters in the tropics as high rates of precipitation and temperature lead to a
lower persistence (i.e., longevity) of excavated cavities [34] and to the relatively rapid formation
of cavities through fungal activity [48]. This, in turn, leads to relatively higher use of decay-
formed cavities by secondary cavity-nesters in the tropics [34, 49], and less dependence on
excavators for the supply of nest-cavities. For example, even woodpeckers will occasionally use
decay-formed cavities in Neotropical temperate rainforests where tree decay is the key driver of
nest-web structure [49]. Indeed, we see that the relationship between excavators and secondary
cavity-nesters is equally strong (Neotropics) or even weaker (Afrotropics) than those between
excavators and non-cavity-nesting forest birds (specialists or generalists), in two of these mainly
tropical regions. But we also found that the relationship between excavators and secondary
cavity-nester was relatively strong in the Oriental region, stronger in fact than those between
excavators and any other group of forest birds. This may be the result of the specific
characteristics of the forests or avifaunal communities in this region but may also be influenced
by relatively high logging rates in the Oriental region. Logging reduces the availability of large
dead trees, which in turn diminishes substrate availability for excavators as well as the
opportunity for cavities to form via decay [50], potentially making secondary cavity-nesters
more dependent on excavated cavities.

In contrast to our predictions, relationships between excavators and secondary cavity-
nesters and between excavators and forest specialists were not necessarily stronger (though this
was the case in the Neotropical region) than those between excavators and forest generalists. We

do not have a clear explanation for this finding, though we argue that forest generalists and excavators might share landscapes where forest specialists are hard to find, for example in open landscapes with scattered trees [51]. Finally, we found that regardless of the relationship, all relationships between excavators and other bird groups were stronger in the more tropical (especially the Afrotropical and Neotropical) regions, with the exception of Australasia (see below), than in the temperate and boreal regions. Arguably, excavators are particularly strong indicators of other bird groups in these tropical regions as these regions still contain vast tracts of unlogged and unmanaged forests, a topic for further research.

Relationships between excavators and all forest-associated bird groups were less evident in Australasia, which is a region that differs largely from all others in both avifauna (e.g., there are no woodpeckers here but there is a relatively high richness of cavity-nesters [11]) and forest structure and dynamics (e.g., cavity densities in Australasian forests are among the highest globally [47]). Specifically, we found a moderately strong relationship between excavators and non-cavity-nesting forest specialists in Australasia, but no such relationships between excavators and secondary-cavity nesters or forest birds with broader habitat requirements (i.e., forest generalists). This could indicate that the excavators in this region (which notably excludes woodpeckers) show a preference for forest conditions (e.g., the provision of trees that are in advanced stages of decay, given that many of these excavators are weaker excavators that require softer or rotting wood; see [11] for species) that are shared by many forest specialists. In contrast, secondary cavity-nesters in Australasia may depend on elements not formed, or particularly required, by excavators (e.g., large trees of a certain age [52] that contain many dead branches and have accumulated multiple decay-formed cavities [53]).

Beyond variation in the strength of relationships between regions we also found that regressions differed in slopes, both across regions and across the different relationships tested. For example, the steep slopes we found for correlations between excavators and non-cavity-nesting forest birds in the predominantly tropical Neotropical region contrast with more shallow slopes in the temperate/boreal Palearctic and Nearctic regions. And, within the Neotropical region, the relationship between excavators and secondary cavity-nesters showed shallower slopes than that between excavators and non-cavity-nesting forest birds. These differences in slopes mainly reflect differences in species richness across regions and groups of forest birds, with relatively steeper slopes representing a relatively larger increase in richness of a particular group of birds per increase in the number of excavators. In turn, we may look at macro-scale processes (e.g., as outlined by Gaston [54]), such as speciation events, that drive these differences across regions in overall richness of birds [55], including cavity-nesting birds [11], and other possible factors (e.g., trees as potential substrates for cavities may also differ across regions [56]).

In conclusion, we found strong and positive relationships between excavator and forest bird richness around most of the world. As such, we deem excavators to be useful indicator species for forest bird diversity at broad spatial scales, as was previously demonstrated at stand- and landscape-scales. Excavators, especially woodpeckers, form a small and fairly easy to identify (even by citizen scientists) subset of a larger group of species (forest birds) often studied to understand the effects of anthropogenic disturbances. Moreover, many woodpeckers respond to playbacks, which may, therefore, become an effective survey tool that allows researchers to survey relatively large areas fairly readily (e.g., [57]). Studies of excavator presence and abundance, especially in high biodiversity tropical regions, might provide quick measures of forest quality and biodiversity, which may in turn guide forest management decisions (e.g.,

improving the retention of large trees). A comprehensive study of all forest birds in some regions
like the Amazon or Congo Basin (where, notably, there are hotspots of woodpecker richness
[58,59]) would require considerable local expertise and training, which is often difficult to
achieve due to social and monetary limitations. Thus, we conclude, excavators have excellent
potential as study subjects for conservation monitoring and planning initiatives (see also [59]
with regards to woodpecker conservation itself), in comparative research, and to guide the
establishment of region-wide forest conservation strategies, especially in the largely
understudied tropical regions of the world.

**Ethics statement**

No field data were obtained nor human or animal subjects used for this study.

**Data accessibility**

Supporting data as well as R scripts can be found on The GitLab repository

https://gitlab.com/gavg712/CB_cavity_nesters, the details of which are outlined in Appendix S1.

**Competing interests**

We have no competing interests.

**Authors' contributions**

YH and GVG carried out the data analysis, and led the design of the study and writing of the
manuscript; MC and KM critically revised the manuscript. All authors contribute to the writing
of the final manuscript, gave final approval for publication and agree to be held accountable for
the work performed therein.

**Acknowledgments**

We would like to thank Lisa Manne, and Kristina Cockle and two anonymous reviewers for their
critical remarks on drafts of this article.

**Funding**

This research was not externally funded.

**References**

Allan JR, Venter O, Maxwell S, Bertzky B, Jones K, Shi Y, Watson JE. 2017 Recent increases in human
pressure and forest loss threaten many Natural World Heritage Sites. *Biological conservation*. **206**, 47-
55.
Hansen MC, Stehman SV, Potapov PV. 2010 Quantification of global gross forest cover loss.
*Proceedings of the National Academy of Sciences*. **107**, 8650-8655.
Barlow J, Lennox GD, Ferreira J, Berenguer E, Lees AC, Mac Nally R, Thomson JR, de Barros Ferraz SF,
Louzada J, Oliveira VHF. 2016 Anthropogenic disturbance in tropical forests can double biodiversity loss
from deforestation. *Nature*. **535**, 144.
Gaston KJ, Blackburn TM, Goldewijk KK. 2003 Habitat conversion and global avian biodiversity loss.
*Proceedings of the Royal Society of London. Series B: Biological Sciences*. **270**, 1293-1300.
Betts MG, Wolf C, Pfeifer M, Banks-Leite C, Arroyo-Rodríguez V, Ribeiro DB, Barlow J, Eigenbrod F,
Faria D, Fletcher RJ. 2019 Extinction filters mediate the global effects of habitat fragmentation on
animals. *Science*. **366**, 1236-1239.

Carignan V, Villard M-A. 2002 Selecting indicator species to monitor ecological integrity: a review.
*Environmental monitoring and assessment*. **78**, 45-61.
- Canterbury GE, Martin TE, Petit DR, Petit LJ, Bradford DF. 2000 Bird communities and habitat as
ecological indicators of forest condition in regional monitoring. *Conservation Biology*. **14**, 544-558.
- Venier LA, Pearce JL. 2004 Birds as indicators of sustainable forest management. *The Forestry*
*Chronicle*. **80**, 61-66.
- Martin K, Ibarra JT, Drever M. 2015 Avian surrogates in terrestrial ecosystems: theory and practice. In
*Indicators and surrogates of biodiversity and environmental change* (eds D Lindenmayer, P Barton, J
Pierson), pp 33-44. Melbourne & London: CSIRO Publishing & CRC Press.
- Ibarra JT, Martin M, Cockle KL, Martin K. 2017 Maintaining ecosystem resilience: functional
responses of tree cavity nesters to logging in temperate forests of the Americas. *Scientific reports*. **7**,
4467.
- van der Hoek Y, Gaona GV, Martin K. 2017 The diversity, distribution and conservation status of the
tree-cavity-nesting birds of the world. *Diversity and Distributions*. **23**, 1120-1131.
- McClelland BR, McClelland PT. 1999 Pileated woodpecker nest and roost trees in Montana: links with
old-growth and forest "health". *Wildlife Society Bulletin*. 846-857.
- Stachura-Skierczyńska K, Kosiński Z. 2016 Do factors describing forest naturalness predict the
occurrence and abundance of middle spotted woodpecker in different forest landscapes? *Ecological*
*Indicators*. **60**, 832-844.
- Marchetti M. 2005 *Monitoring and indicators of forest biodiversity in Europe: from ideas to*
*operationality*. Joensuu, Finland: European Forest Institute.
- Roberge J-M, Angelstam P. 2006 Indicator species among resident forest birds—a cross-regional
evaluation in northern Europe. *Biological Conservation*. **130**, 134-147.
- Virkkala R. 2006 Why study woodpeckers? The significance of woodpeckers in forest ecosystems.
*Annales Zoologici Fennici*. 82-85.
- Drever MC, Aitken KE, Norris AR, Martin K. 2008 Woodpeckers as reliable indicators of bird richness,
forest health and harvest. *Biological conservation*. **141**, 624-634.
- Drever MC, Martin K. 2010 Response of woodpeckers to changes in forest health and harvest:
Implications for conservation of avian biodiversity. *Forest Ecology and Management*. **259**, 958-966.
- Mikusiński G, Gromadzki M, Chylarecki P. 2001 Woodpeckers as indicators of forest bird diversity.
*Conservation Biology*. **15**, 208-217.
- Angelstam PK, Bütler R, Lazdinis M, Mikusiński G, Roberge J-M. 2003. Habitat thresholds for focal
species at multiple scales and forest biodiversity conservation—dead wood as an example. *Annales*
*Zoologici Fennici*. 473-482.
- Nilsson SG, Hedin J, Niklasson M. 2001 Biodiversity and its assessment in boreal and nemoral forests.
*Scandinavian Journal of Forest Research*. **16**, 10-26.
- Hess GR, Bartel RA, Leidner AK, Rosenfeld KM, Rubino MJ, Snider SB, Ricketts TH. 2006 Effectiveness
of biodiversity indicators varies with extent, grain, and region. *Biological Conservation*. **132**, 448-457.
- Collen B, Ram M, Zamin T, McRae L. 2008 The tropical biodiversity data gap: addressing disparity in
global monitoring. *Tropical Conservation Science*. **1**, 75-88.
- Ilsøe SK, Kissling WD, Fjeldså J, Sandel B, Svenning JC. 2017 Global variation in woodpecker species
richness shaped by tree availability. *Journal of Biogeography*. **44**, 1824-1835.
- Bell D, Hjältén J, Nilsson C, Jørgensen D, Johansson T. 2015 Forest restoration to attract a putative
umbrella species, the white-backed woodpecker, benefited saproxylic beetles. *Ecosphere*. **6**, 1-14.
- Gatica-Saavedra P, Echeverría C, Nelson CR. 2017 Ecological indicators for assessing ecological
success of forest restoration: a world review. *Restoration ecology*. **25**, 850-857.

Hoyt JS, Hannon SJ. 2002 Habitat associations of black-backed and three-toed woodpeckers in the
boreal forest of Alberta. *Canadian Journal of Forest Research*. **32**, 1881-1888.
Roberge, J. M., Angelstam, P., & Villard, M. A. (2008). Specialised woodpeckers and naturalness in
hemiboreal forests—deriving quantitative targets for conservation planning. *Biological conservation*,
141(4), 997-1012.
Tikkanen O-P, Martikainen P, Hyvärinen E, Junninen K, Kouki J. 2006. Red-listed boreal forest species
of Finland: associations with forest structure, tree species, and decaying wood. *Annales Zoologici*
*Fennici*. 373-383.
Bütler R, Angelstam P, Ekelund P, Schlaepfer R. 2004 Dead wood threshold values for the three-toed
woodpecker presence in boreal and sub-Alpine forest. *Biological Conservation*. **119**, 305-318.
McLaren MA, Thompson ID, Baker JA. 1998 Selection of vertebrate wildlife indicators for monitoring
sustainable forest management in Ontario. *The Forestry Chronicle*. **74**, 241-248.
Martin K, Eadie JM. 1999 Nest webs: A community-wide approach to the management and
conservation of cavity-nesting forest birds. *Forest Ecology and Management*. **115**, 243-257.
Cockle KL, Martin K, Robledo G. 2012 Linking fungi, trees, and hole-using birds in a Neotropical tree-
cavity network: Pathways of cavity production and implications for conservation. *Forest Ecology and*
*Management*. **264**, 210-219.
Cockle KL, Martin K, Wesolowski T. 2011 Woodpeckers, decay, and the future of cavity-nesting
vertebrate communities worldwide. *Frontiers in Ecology and the Environment*. **9**, 377-382.
Tyljanakis JM, Klein AM, Lozada T, Tschardt T. 2006 Spatial scale of observation affects α , β and γ
diversity of cavity-nesting bees and wasps across a tropical land-use gradient. *Journal of Biogeography*.
**33**, 1295-1304.
Powell S, Costa AN, Lopes CT, Vasconcelos HL. 2011 Canopy connectivity and the availability of
diverse nesting resources affect species coexistence in arboreal ants. *Journal of Animal Ecology*. **80**, 352-
360.
Jankowiak R, Ciach M, Bilański P, Linnakoski R. 2019. Diversity of wood-inhabiting fungi in
woodpecker nest cavities in southern Poland. *Acta Mycologica*. **54**, 1126. doi:10.5586/am.1126
Jusino MA, Lindner DL, Banik MT, Rose KR, Walters JR. 2016 Experimental evidence of a symbiosis
between red-cockaded woodpeckers and fungi. *Proceedings of the Royal Society B: Biological Sciences*.
**283**, 20160106.
Bull EL, Jackson JA., 1995 Pileated Woodpecker (*Dryocopus pileatus*). In: *The birds of North America*
(eds A Poole, F Gill). Washington DC: The Academy of Natural Sciences & The American Ornithologists'
Union.
Montellano MGN, Blendinger PG, Macchi L. 2013 Sap consumption by the White-fronted
Woodpecker and its role in avian assemblage structure in dry forests. *The Condor*. **115**, 93-101.
Ruggera RA, Schaaf AA, Vivanco CG, Politi N, Rivera LO. 2016 Exploring nest webs in more detail to
improve forest management. *Forest Ecology and Management*. **372**, 93-100.
Betts MG, Wolf C, Ripple WJ, Phalan B, Millers KA, Duarte A, Butchart SH, Levi T. 2017 Global forest
loss disproportionately erodes biodiversity in intact landscapes. *Nature*. **547**, 441-444.
BirdLife International and NatureServe 2015 *Bird species distribution maps of the world*. Cambridge
and Arlington: BirdLife International and NatureServe.
Storch D, Davies RG, Zajíček S, Orme CDL, Olson V, Thomas GH, Ding TS, Rasmussen PC, Ridgely RS,
Bennett PM. 2006 Energy, range dynamics and global species richness patterns: reconciling mid-domain
effects and environmental determinants of avian diversity. *Ecology Letters*. **9**, 1308-1320.

Pinheiro J, Bates D, DebRoy S, Sarkar D, Heisterkamp S, Van Willigen B. 2017 Package 'nlme'. *Linear*
*and Nonlinear Mixed Effects Models, version 3-1*. Available from [https://cran.r-](https://cran.r-project.org/web/packages/nlme/nlme.pdf)
[project.org/web/packages/nlme/nlme.pdf](https://cran.r-project.org/web/packages/nlme/nlme.pdf) (accessed January 15, 2018).
Kissling WD, Sekercioglu CH, Jetz W. 2012 Bird dietary guild richness across latitudes, environments
and biogeographic regions. *Global Ecology and Biogeography*. **21**, 328-340.
Hunter M, Westgate M, Barton P, Calhoun A, Pierson J, Tulloch A, Beger M, Branquinho C, Caro T,
Gross J. 2016 Two roles for ecological surrogacy: Indicator surrogates and management surrogates.
*Ecological Indicators*. **63**, 121-125.
Remm J, Löhmus A. 2011 Tree cavities in forests—the broad distribution pattern of a keystone
structure for biodiversity. *Forest Ecology and Management*. **262**, 579-585.
Altamirano TA, Ibarra JT, Martin K, Bonacic C. 2017 The conservation value of tree decay processes as
a key driver structuring tree cavity nest webs in South American temperate rainforests. *Biodiversity and*
*Conservation*. **26**, 2453-2472.
Styring AR, bin Hussin MZ. 2004. Effects of logging on woodpeckers in a Malaysian rain forest: the
relationship between resource availability and woodpecker abundance. *Journal of Tropical Ecology*.
**20**(5), 495-504.
Fischer J, Stott J, Law BS. 2010. The disproportionate value of scattered trees. *Biological*
*Conservation*. **143**(6), 1564-1567.
Lindenmayer DB, Blanchard W, McBurney L, Blair D, Banks S, Likens GE, Franklin JF, Laurance WF,
Stein JA, Gibbons P. 2012. Interacting factors driving a major loss of large trees with cavities in a forest
ecosystem. *PLoS One*. **7**(10).
Gibbons P, Lindenmayer DB, Barry SC, Tanton MT. 2002. Hollow selection by vertebrate fauna in
forests of southeastern Australia and implications for forest management. *Biological Conservation*. **103**,
1-12.
Gaston KJ. 2000 Global patterns in biodiversity. *Nature*. **405**, 220.
Blackburn TM, Gaston KJ. 1996 Spatial patterns in the species richness of birds in the New World.
*Ecography*. **19**, 369-376.
Currie DJ, Paquin V. 1987 Large-scale biogeographical patterns of species richness of trees. *Nature*.
**329**, 326-327.
Baumgardt JA, Sauder JD, Nicholson KL. 2014. Occupancy modeling of woodpeckers: maximizing
detections for multiple species with multiple spatial scales. *Journal of Fish and Wildlife Management*.
**5**(2), 198-207.
Mikusiński G. 2006 Woodpeckers: distribution, conservation, and research in a global perspective.
*Annales Zoologici Fennici*. 86-95.
Vergara-Tabares DL, Lammertink M, Verga EG, Schaaf AA, Nori J. 2018. Gone with the forest:
Assessing global woodpecker conservation from land use patterns. *Diversity and Distributions*. **24**, 640-
51.

Table 1. Species richness for forest-associated bird groups found in each of six zoogeographic regions. In parenthesis the maximum number of species of each group found in a single 10×10 km grid cell in each region.

Zoographic region	Excavators	Secondary cavity-nesters	Forest specialists ¹	Forest generalists ¹	All forest birds (specialists + generalists) ¹
Nearctic	42 (16)	111 (41)	93 (36)	365 (111)	458 (140)
Palaearctic	67 (23)	181 (58)	155 (65)	725 (221)	880 (276)
Oriental	120 (35)	241 (67)	463 (79)	958 (238)	1421 (300)
Neotropical	173(36)	373 (114)	1078 (200)	1764 (277)	2842 (436)
Afrotropical	78 (33)	218 (68)	220 (60)	911 (287)	1131 (331)
Australasia	14 (7)	178 (70)	442 (67)	1014 (190)	1456 (226)

¹Note: these numbers do not include secondary cavity-nesters.

**Table 2. Results of generalized least square models of possible correlations between (1)**
**excavator and secondary cavity-nester richness, (2) excavator and forest specialist richness,**
**(3) excavator and forest generalist richness, (4) excavator and forest bird (specialists and**
**generalists) richness.** Each result reflects the mean outcome of 1000 models build with 1000
randomly selected grid cell data points that include an exponential spatial covariance structure.
The mean Spearman rho was calculated over the fit of each of 1000 models to a test data set.

Zoogeographic region	Correlation	Mean Slope (Min. – Max.; SD)	% of models with $P < 0.05$	Mean Spearman rho
Nearctic	1	0.8 (0.4 - 1.3; 0.1)	60	0.73
	2	1.0 (0.7 - 1.3; 0.1)	97	0.71
	3	3.1 (2.0 - 4.5; 0.4)	89	0.76
	4	4.2 (2.7 - 5.7; 0.5)	94	0.81
Palearctic	1	1.1 (0.7 - 1.4; 0.1)	84	0.78
	2	0.6 (0.3 - 1.1; 0.3)	83	0.68
	3	2.9 (2.1 - 4.1; 0.3)	82	0.80
	4	3.5 (2.6 - 5.4; 0.4)	88	0.80
Oriental	1	0.7 (0.6 - 0.9; 0.0)	100	0.88
	2	1.1 (0.9 - 1.3; 0.1)	100	0.72
	3	3.5 (3.0 - 4.0; 0.2)	100	0.86
	4	4.6 (3.9 - 5.5; 0.2)	100	0.83
Neotropical	1	2.4 (1.9 - 3.0; 0.2)	83	0.95
	2	3.5 (9.1 - 14.0; 0.2)	73	0.95
	3	5.2 (3.9 - 6.7; 0.5)	77	0.91
	4	8.8 (7.1 - 11.0; 0.6)	84	0.96
Afrotropical	1	1.1 (0.7 - 1.9; 0.2)	76	0.79
	2	0.6 (0.3 - 1.1; 0.1)	78	0.83
	3	5.8 (3.7 - 8.1; 0.9)	84	0.93
	4	6.4 (4.3 - 9.2; 0.9)	87	0.93
Australasian	1	1.8 (1.4 - 2.2; 0.1)	88	-0.14
	2	1.7 (0.9 - 2.5; 0.3)	100	0.73
	3	9.2 (7.2 - 11.2; 0.6)	96	0.19
	4	11.0 (8.3 - 13.5; 0.8)	99	0.50

1
2
3 6034
5 6046 605 **Figure legends**7

**Figure 1. Global map of the relative richness of tree cavity excavators versus secondary**
**cavity-nesting birds (expressed as the ratio excavator / secondary cavity-nester) in 10x10**
**km grid cells.** Low values (light colors) indicate relatively low numbers of excavators as
compared to secondary cavity-nesters while high values (dark colors) indicate a relatively higher
proportion of excavators to secondary cavity-nesters.

**Figure 2. Scatter plots of the correlations between the richness (number of species) of**
**excavators and secondary cavity-nesters all forest birds (upper row), all forest specialists**
**(second row), forest generalists (third row) and all non-cavity-nesting forest birds**
**(specialists and generalists; bottom row) in six zoogeographic regions.** Points represent
species richness values in 10×10 km grid cells, and the solid line represents the fit of a linear
regression model, created using all grid cell values. Note that nearly all secondary cavity-nesters
are also forest-associated species, but were not included as forest specialists or generalists.

Figure 1.

Figure 2.

670

Figure 1. Global map of the relative richness of tree cavity excavators versus secondary cavity-nesting birds (expressed as the ratio excavator / secondary cavity-nester) in 10x10 km grid cells. Low values (light colors) indicate relatively low numbers of excavators as compared to secondary cavity-nesters while high values (dark colors) indicate a relatively higher proportion of excavators to secondary cavity-nesters.

Figure 2. Scatter plots of the correlations between the richness (number of species) of excavators and secondary cavity-nesters all forest birds (upper row), all forest specialists (second row), forest generalists (third row) and all non-cavity-nesting forest birds (specialists and generalists; bottom row) in six zoogeographic regions. Points represent species richness values in 10x10 km grid cells, and the solid line represents the fit of a linear regression model, created using all grid cell values. Note that nearly all secondary cavity-nesters are also forest-associated species, but were not included as forest specialists or generalists.

Appendix F

We thank both Editor and Reviewers for their suggestions and for giving us the chance to submit another version. We would like to point out that unfortunately the previous 'track-changed version' had some editing issues. The 'clean copy' (i.e. without track changes) was much clearer (some small writing issues only appeared after we had removed the track changes, which made it difficult to keep overview). In the clean copy version, which was, unfortunately, not used as a base for the new reviews, there were also some changes on the references (numbering) and a few different sentences.

We apologize to the Reviewers for not having submitted that comprehensive track-changed document. That said, we see that this time around we could still make some valuable improvements. For that, we thank the reviewers for their comprehensive and insightful comments.

You will find a new version with edits attached. A few larger changes are discussed below.

We identify our responses to reviewer comments with an * preceding our response.

Reviewer 1:

Lines 185-190: poor English – check and re-write

*We made changes to this part of the text that will hopefully improve the clarity of our message.

Lines 219-220: provide names of the authors in the text

*We added these references in parentheses rather: “We selected and classified focal bird species (following [11, 42]) as...”

Line 295: was rho really 0.88 in both cases?

*This was a mistake, we only wanted to highlight one relationship (between excavators and secondary cavity-nesters, $\rho=0.88$). We have removed the mention of the other relationship.

Line 367: provide valid reference for this claim i.e. on high logging rates in Oriental Region

*We received the same comment from the other reviewer. We have altered this a bit to focus on the availability of mature forests instead, and have added a reference for that.

Line 373-377: it seems to be against your prediction. Whole sentence is a bit confusing.

*We have rephrased this, and reordered the paragraphs. We hope that this new version provides more clarity.

Line 386: what dynamics do you mean? Fire dynamics?

*We understand that this wording is a bit confusing. Rather, we are interested in discussing the 'processes' of cavity formation, tree decay and persistence, etc., for which reason we highlight the example of the 'process' of decay-formation of tree cavities.

Line 514-516: I would explicitly mention high detectability here

*We have added the words high detectability to this phrase.

Reviewer 2:

-38 and 39?? Please check again cite numbers throughout the manuscript

*These citations were not updated in the track-change version of the manuscript as Endnote did not update numbering until track-changes were accepted. The citations were correctly numbered in the actual, clean, version that was also submitted and that could be found later on in the PDF. In this new submission, both the track-change and clean versions of the manuscript are updated.

I'd suggest to change the name, because secondary-cavity nesters are also 'forest birds', and are not included in this group... can be confusing

*We have retained the name secondary-cavity nesters, but now explain in the methods that forest generalists are 'habitat generalist birds that use forest habitat but also at least one other type of habitat, but that this group in our study excludes all tree cavity-nesters.', following the reviewer's suggestion.

-incorrect. In Australasia, excavators and secondary cavity-nesters had a negative relationship (Fig. 2)

*We understand that this part was confusing. The gls slopes were positive for this relationship, but Spearman rho is negative. However, in general Spearman rho was so low for the relationships between excavators vs. secondary-cavity nesters and excavators vs. forest generalists ($\leq |0.19|$ for both) that we may question whether there are any relationships at all. We added a statement to explain this in the text. Finally, we also point out that due to computational constraints, the figure depicting the relationships (Figure 2) shows lines representing linear regressions rather than gls models, and that these models might have slightly different outcomes. That said, to avoid further confusion, we removed the regression lines for the two relationships in Australasia that had the negligible relationships.

-Again, the use of 'all forest' is confusing. Besides, apparently you have made a correlation between excavators and specialists + generalists. In the responses to comments on the previous version, you have answered "We therefore opted to re-run all the analyzes for groups that do not have any overlaps", which somewhat contradicts this. In any case, although you in the introduction you mention that this relationship will be assessed, it is not explained in methods, and honestly I don't see the sense of evaluating these two groups together, at least with the information provided in the introduction.

*We thank the reviewer for his/her patience in pointing out these inconsistencies. We have altered the text to make clear that we have used non-overlapping groups for three of the relationships (and explained in the methods what these groups are, e.g., "forest generalist birds: habitat generalist birds that use forest habitat but also at least one other habitat type for nesting of habitat, again excluding cavity-nesters."). But we have also explained in more detail why we added tests of a fourth relationship, that between excavators and forest specialists + generalists (grouped). We opted to include this because it strengthens the notion that excavators may be effective indicators of not just the narrow group of specialist species, but of the larger forest bird community in general.

-well... you're saying that .96 is stronger than .95. I'd say that is a minimal difference in which the focus of a discussion doesn't have to be focused.

*In this version we have completely reworked this part of the Discussion, and we no longer make this point. We have followed the reviewer's advice elsewhere, and focus on other discussion points.

-I'd like to see more explanations/discussions on other patterns:

-why relationships between excavators and SCN are mostly weaker than the others relationships across regions? This was one of the relationship more strong that one expect by cavity facilitation

*We explain this is in more detail now. For example, a part of the new text includes:

“This indicates that the influence of direct facilitation of nest-cavities is potentially less important than the indirect associations with forest elements and shared habitat preferences in shaping these relationships, but that the relative importance of both depends on the region under consideration.” We add to that in other paragraphs, explanations such as “we found weaker relationships between excavators and secondary cavity-nesters than between excavators and forest specialists in both the Neotropical and Afrotropical regions, possibly because at least some secondary-cavity nesters do not share a preference for habitats and forest elements with excavators in this region and because a high reliance on decay-formed cavities releases secondary-cavity nesters from dependency on excavators to provide their cavities [34]. “

2) given the known pattern of the greater dependence of SCN in woodpecker-excavated cavities in Nearctic and Palearctic than in tropical regions, why these relationships showed the opposite pattern? (greater rho in tropics than in north temperate and boreal regions)

*We address these patterns in more detail now. For example, a part of the new text includes:

“Thus, the fact that the relationships between excavators and secondary-cavity nesters are stronger in all three largely tropical regions (Oriental, Neotropical, and Afrotropical) as compared to the predominantly temperate/boreal Nearctic and Palearctic may not reflect a larger dependency of secondary cavity-nesters on excavators for cavities in the tropics, but is more likely to stem from overall high species richness as well as from strong indirect associations that both groups of birds have with forest elements and shared habitat preferences.”

-Logging is a highly heterogeneous activity in intensity and extension throughout the world, and of relatively recent impact. Therefore, resorting to this explanation to explain a biogeographic scale pattern, where evolutionary time scales and speciation rates are involved, seems inappropriate. Moreover, it is not clear to me how the changes produced by logging could be influencing the data set used in the analyzes. For example, do the maps you used include information on local extinctions across all world’s forests due exclusively to logging?

*We understand that this might have come across a bit speculative. As we do think that historical deforestation patterns are very different in the Oriental than the Neotropical and Afrotropical regions, due to higher human densities etc., and that this could shape richness patterns, even at larger spatial scales. Therefore, we have altered our discussion to state that there is less mature forest left in this region, a more general statement that, we think, could have influenced the observed patterns.

Appendix G

*We thank the Editor(s) and reviewers for their time and intellectual investment, this has surely led to a much-improved manuscript. We understood the need for further editing of the language, and both the first author as well one of the co-authors (a native English speaker with a long and substantial publication record) have made edits that ensure the manuscript is easier to read and avoid of grammatical or other language mistakes.

We do not have a point-by-point rebuttal at this point, but hope that the text is overall improved, and that there are no more errors such as those previously highlighted by the reviewer in lines 312-319 and 363-368.

*In addition, to the best of our knowledge, we have included all required statements (e.g., on funding etc.). Of course, should we have missed anything, we would be happy to provide further details.

On behalf of the Editors, I am pleased to inform you that your Manuscript RSOS-192177.R2 entitled "Global relationships between tree-cavity excavators and forest bird richness" has been accepted for publication in Royal Society Open Science subject to minor revision in accordance with the referee suggestions. Please find the referees' comments at the end of this email.

The reviewers and Subject Editor have recommended publication, but also suggest some minor revisions to your manuscript. Therefore, I invite you to respond to the comments and revise your manuscript.

- Ethics statement

- Data accessibility

<http://datadryad.org/submit?journalID=RSOS&manu=RSOS-192177.R2>

- Competing interests

- Authors' contributions

All submissions, other than those with a single author, must include an Authors' Contributions section which individually lists the specific contribution of each author. The list of Authors should

meet all of the following criteria; 1) substantial contributions to conception and design, or acquisition of data, or analysis and interpretation of data; 2) drafting the article or revising it critically for important intellectual content; and 3) final approval of the version to be published.

- Acknowledgements

- Funding statement

Because the schedule for publication is very tight, it is a condition of publication that you submit the revised version of your manuscript before 19-Jun-2020. Please note that the revision deadline will expire at 00.00am on this date. If you do not think you will be able to meet this date please let me know immediately.

- 1) A text file of the manuscript (tex, txt, rtf, docx or doc), references, tables (including captions) and figure captions. Do not upload a PDF as your "Main Document".
- 2) A separate electronic file of each figure (EPS or print-quality PDF preferred (either format should be produced directly from original creation package), or original software format)
- 3) Included a 100 word media summary of your paper when requested at submission. Please ensure you have entered correct contact details (email, institution and telephone) in your user account

4) Included the raw data to support the claims made in your paper. You can either include your data as electronic supplementary material or upload to a repository and include the relevant doi within your manuscript

5) All supplementary materials accompanying an accepted article will be treated as in their final form. Note that the Royal Society will neither edit nor typeset supplementary material and it will be hosted as provided. Please ensure that the supplementary material includes the paper details where possible (authors, article title, journal name).

on behalf of Prof Kevin Padian (Subject Editor)
openscience@royalsociety.org

Associate Editor Comments to Author :

The reviewer is broadly satisfied with the scientific content of the paper, though they would prefer that you seek further English language support. Examples of professional services providing such advice may be found at <https://royalsociety.org/journals/authors/benefits/language-editing/>.

Reviewer comments to Author:

Reviewer: 1

Comments to the Author(s)

I am generally pleased with changes included in this version of the manuscript. However, it would be great to carefully read some of the new sentences and improve them linguistically. For example check lines 312-319 and 363-368 (clean version) for missing words.

Journal Name: Royal Society Open Science
Journal Code: RSOS

Online ISSN: 2054-5703

Journal Admin Email: openscience@royalsociety.org

Journal Editor: Andrew Dunn

Journal Editor Email: openscience@royalsociety.org

MS Reference Number: RSOS-192177.R2

Article Status: SUBMITTED

MS Dryad ID: RSOS-192177.R2

MS Title: Global relationships between tree-cavity excavators and forest bird richness

MS Authors: van der Hoek, Yntze; Gaona, Gabriel; Ciach, Michal; Martin, Kathy

Contact Author: Yntze van der Hoek

Contact Author Email: yntzevanderhoek@gmail.com

Contact Author Address 1: NA

Contact Author Address 2: NA

Contact Author Address 3:

Contact Author City: Musanze

Contact Author State: NA

Contact Author Country: Rwanda

Contact Author ZIP/Postal Code: NA

Keywords: Facilitator species, Indicator species, Species interactions, Picidae, Management surrogates, Secondary cavity-nesting birds

Abstract: Global monitoring of biodiversity and ecosystem change can be aided by the effective use of indicators. Tree-cavity excavators, the majority of which are woodpeckers (Picidae), are known to be useful indicators of the health or naturalness of forest ecosystems and the diversity of forest birds. They are indicators of the latter due to their associations with particular forest elements and because of their role in facilitating the occurrence of other species through the provision of nest cavities. Here, we investigated whether these positive correlations between excavators and other forest birds are also found at broad geographical scales. We used global distribution maps to extract richness estimates of tree-cavity nesting and forest-associated birds, which we grouped by zoogeographic regions. We then created generalized least squares models to assess the relationships between these groups of birds. We show that richness of tree-cavity excavating birds correlates positively with that of secondary cavity nesters, and with other forest birds (generalists and specialists), at global scales, but with variation across zoogeographic regions. As many excavators are relatively easy to detect, play keystone roles at local scales, and are effective management targets, we propose that excavators are useful for biodiversity monitoring across multiple spatial scales and geographic regions, especially in the tropics.

EndDryadContent